# Learning non-equilibrium diffusions with Schrödinger bridges: from exactly solvable to simulation-free

**Stephen Y. Zhang***
University of Melbourne

**Michael P. H. Stumpf**
University of Melbourne

## Abstract

We consider the Schrödinger bridge problem which, given ensemble measurements of the initial and final configurations of a stochastic dynamical system and some prior knowledge on the dynamics, aims to reconstruct the "most likely" evolution of the system compatible with the data. Most existing literature assume Brownian reference dynamics, and are implicitly limited to modelling systems driven by the gradient of a potential energy. We depart from this regime and consider reference processes described by a multivariate Ornstein-Uhlenbeck process with generic drift matrix $A \in \mathbb{R}^{d \times d}$. When $A$ is asymmetric, this corresponds to a non-equilibrium system in which non-gradient forces are at play: this is important for applications to biological systems, which naturally exist out-of-equilibrium. In the case of Gaussian marginals, we derive explicit expressions that characterise exactly the solution of both the static and dynamic Schrödinger bridge. For general marginals, we propose MVOU-OTFM, a simulation-free algorithm based on flow and score matching for learning an approximation to the Schrödinger bridge. In application to a range of problems based on synthetic and real single cell data, we demonstrate that MVOU-OTFM achieves higher accuracy compared to competing methods, whilst being significantly faster to train.

## 1   Introduction

We are interested in reconstruction of stochastic dynamics of individuals from static population snapshots. This is a central problem with applications arising across the natural and social sciences, whenever longitudinal tracking of individuals over time is either impossible or impractical [56, 35, 48, 43, 34, 1]. In simple terms, consider a system of indistinguishable particles $x_t \in \mathbb{R}^d$ undergoing some unobserved temporal dynamics. The practitioner observes the system to have distribution $x_0 \sim \rho_0$ at an initial time $t = 0$ and later to be $x_1 \sim \rho_1$ at a final time $t = 1$. The question is then: can we, under some suitable assumptions, reconstruct the *continuous-time* behaviour of the system for the unobserved time interval $0 < t < 1$?

The Schrödinger bridge problem (SBP), by now a centrepiece of the theoretical literature on this topic, places this task on a theoretical footing in terms of a mathematical formulation of this problem as a large deviations principle on the path space [28] and a (stochastic) least action principle intimately linked to optimal transportation theory [12]. Given a *reference process* that encodes prior knowledge on the dynamics, the SBP can be understood as identifying the "most likely" evolution of the system that is compatible with the snapshot observations.

The SBP and related topics have enjoyed a great deal of recent interest from both applied and theoretical perspectives. Many applications arise from biological modelling of cell dynamics [7, 42]. The majority of existing work assumes, either explicitly or implicitly, that the system of interest is *potential driven*, that is, driven by the gradient of a potential energy [26, 13, 52, 5, 51, 57, 6]. In fact,

---

*Correspondence to `syz@syz.id.au`

39th Conference on Neural Information Processing Systems (NeurIPS 2025).

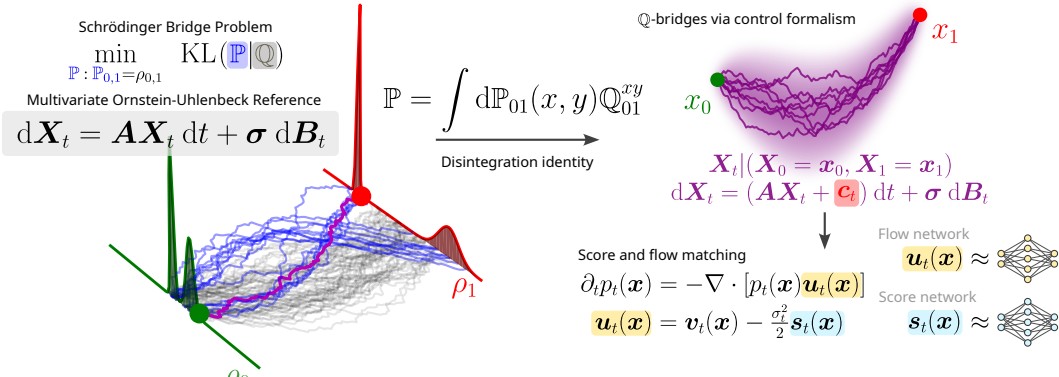

Figure 1: The Schrödinger bridge problem with a multivariate Ornstein-Uhlenbeck reference process (1) can be solved via a generalised entropic transport problem and characterisation of the $\mathbb{Q}$-bridges. For non-Gaussian endpoints, score and flow matching provide a route to building neural approximations without simulation.

most studies do not consider *any* prior drift, corresponding to the setting where the reference process is purely an isotropic Brownian motion. A few studies [5, 45, 15] allow for scalar Ornstein-Uhlenbeck (OU) processes, motivated by applications in the generative modelling domain. Since scalar OU dynamics have unidimensional drift which can always be written as the gradient of a scalar function, all of these are also potential driven systems.

Changing the choice for the reference process provides a route to dealing with systems which are not potential driven, i.e. driven by a drift that is a non-conservative vector field. Abundant motivations for modelling these dynamics arises from biological systems and other kinds of active matter; these exist naturally far from equilibrium [60, 37, 19, 17], exhibiting irreversible dynamics at a non-equilibrium steady state [27]. While a few studies [53, 44, 25] allow for reference processes described by a SDE with a *generic* drift term, they rely on computationally expensive simulation procedures such as numerical integration and suffer from accuracy issues in high dimensions. Here we explore a complementary approach and concentrate our attention on a family of *linear* reference dynamics as a middle ground between physical relevance and analytical tractability. Specifically, we consider reference processes arising in a class of linear drift-diffusion dynamics described by the SDE

$$\mathrm{d}\boldsymbol{X}_t = \boldsymbol{A}(\boldsymbol{X}_t - \boldsymbol{m})\,\mathrm{d}t + \boldsymbol{\sigma}\,\mathrm{d}\boldsymbol{B}_t. \tag{1}$$

These processes are also known as *multivariate Ornstein-Uhlenbeck* (mvOU) processes, often used as a model of non-equilibrium systems in the physics literature [27, 20, 18]. Indeed, for asymmetric drift matrix $\boldsymbol{A}$ the system (1) has a drift that is no longer the gradient of a potential. As an example in the case of isotropic diffusion $\boldsymbol{\sigma} = \sigma\mathbf{I}$, when $\boldsymbol{A}$ is asymmetric and has all eigenvalues with negative real part, a non-equilibrium steady state exists where the dynamics are irreversible and exhibit nonzero probability currents while the population density is unchanged [18, 20]. While allowing a broader range of reference dynamics, the statistics of (1) remain analytically tractable – in particular it is a Markovian Gaussian process with explicit formulae for its mean, covariance, and transition kernel. This provides avenues to efficient solution of the SBP that sidesteps the need for numerical integration, and as we will see, even analytical expressions in the Gaussian case. In particular, the family of SBP problems that we model is strictly larger than the standard (Brownian) SBP, which we recover as a special case when $\boldsymbol{A}, \boldsymbol{m} = \boldsymbol{0}$.

The contributions of our paper are twofold. Leveraging results on mvOU processes, we characterise for Gaussian endpoints the Gaussian Schrödinger Bridge (GSB) with reference dynamics described by (1). To handle general marginals, we develop a *simulation free* training procedure based on score and flow matching [52] to solve the SBP. We conclude by demonstrating our results in a range of synthetic and real data examples. We find that our approach solves the generalised SBP faster and more accurately than comparable simulation-based algorithms, at the cost of assuming the form (1) of the reference process. Our findings highlight the tradeoff between (i) analytically tractable but more restrictive models and (ii) more expressive non-parametric models, which are much less tractable to learn, both in terms of computational cost and statistical complexity.

## 2 Background and related work

### 2.1 Schrödinger bridges

In what follows we provide a concise summary of the SBP. For details we refer readers to [28, 12] for in-depth discussion. We work in $\mathcal{X} \subseteq \mathbb{R}^d$ and denote by $\mathcal{P}(\mathcal{X})$ the space of probability distributions on $\mathcal{X}$. Let $C([0,1], \mathcal{X})$ denote the space of continuous paths $\omega_t : [0,1] \mapsto \mathcal{X}$ valued in $\mathcal{X}$, and informally we will refer to *path measures* as probability measures supported on $C([0,1], \mathcal{X})$. Viewing stochastic processes as random variables valued in $C([0,1], \mathcal{X})$, path measures prescribe their law. Let $\mathbb{Q}$ be a general path measure describing a *reference* Markov stochastic process. For prescribed initial and final marginals $\rho_0, \rho_1 \in \mathcal{P}(\mathcal{X})$, the Schrödinger bridge problem can be written

$$\min_{\mathbb{P} \in \mathcal{P}(C([0,1],\mathcal{X})) \,:\, \mathbb{P}_0 = \rho_0, \mathbb{P}_1 = \rho_1} \mathrm{KL}(\mathbb{P}|\mathbb{Q}), \tag{SBP-dyn}$$

where the minimum is taken over all candidate processes $\mathbb{P}$ absolutely continuous with respect to $\mathbb{Q}$ that are compatible with the observed data at $t = 0, 1$. This *dynamic* form of the SBP, while elegant, is unwieldy for practical purposes owing to the formulation on path space. A well known result [28] connects the dynamical SBP with its static counterpart:

$$\min_{\mathbb{P}_{01} \in \Pi(\rho_0, \rho_1)} \mathrm{KL}(\mathbb{P}_{01}|\mathbb{Q}_{01}). \tag{SBP-static}$$

Furthermore, the static and dynamic SBP are interchangeable via the disintegration identity regarding the optimal law $\mathbb{P}^\star$:

$$\mathbb{P}^\star(\cdot) = \int \mathrm{d}\mathbb{P}_{01}^\star(x_0, x_1) \mathbb{Q}^{(x_0, x_1)}(\cdot), \tag{2}$$

where $\mathbb{Q}^{xy}$ denotes the law of the $\mathbb{Q}$-bridge conditioned at $(0, x_0)$ and $(1, x_1)$. In other words, solution of (SBP-dyn) amounts to solving (SBP-static) for $\mathbb{P}_{01}^\star$ followed by construction of $\mathbb{P}^\star$ as per (2) by taking mixtures of $\mathbb{Q}$-bridges.

The problem (SBP-static) can be reformulated as an entropy-regularised optimal transportation problem [12] where the effective cost matrix is the log-transition kernel under $\mathbb{Q}$. This admits efficient solution via the Sinkhorn-Knopp algorithm [14] in the discrete case. This provides a practical roadmap to constructing dynamical Schrödinger bridges by first solving (SBP-static), then using construction of $\mathbb{Q}$-bridges with (2) to build a solution of (SBP-dyn).

### 2.2 Probability flows and (score, flow)-matching

**Probability flows** Consider a generic drift-diffusion process in $d$ dimensions with drift $\boldsymbol{v}_t(\boldsymbol{x})$ and diffusivity $\boldsymbol{\sigma}_t$, described by an Itô diffusion whose marginal densities evolve following the corresponding Fokker-Planck equation (FPE):

$$\mathrm{d}\boldsymbol{X}_t = \boldsymbol{v}_t(\boldsymbol{X}_t)\,\mathrm{d}t + \boldsymbol{\sigma}_t\,\mathrm{d}\boldsymbol{B}_t, \qquad \partial_t p_t(\boldsymbol{x}) = -\nabla \cdot (p_t(\boldsymbol{x})\boldsymbol{v}_t(\boldsymbol{x})) + \nabla \cdot (\boldsymbol{D}_t \nabla p_t(\boldsymbol{x})), \tag{3}$$

where $\boldsymbol{D}_t = \frac{1}{2}\boldsymbol{\sigma}_t \boldsymbol{\sigma}_t^\top$ is the diffusivity matrix. Equivalently, the FPE can be rewritten in the form of a continuity equation involving a *probability flow* field $\boldsymbol{u}_t$ [30, 3, 2, 52, 31]:

$$\partial_t p_t(\boldsymbol{x}) = -\nabla \cdot [p_t(\boldsymbol{x})\boldsymbol{u}_t(\boldsymbol{x})], \qquad \boldsymbol{u}_t(\boldsymbol{x}) = \boldsymbol{v}_t(\boldsymbol{x}) - \boldsymbol{D}_t \nabla_{\boldsymbol{x}} \log p_t(\boldsymbol{x}), \tag{4}$$

By recognising the form of the continuity equation in (4), it is apparent that the family of *marginal distributions* $\{p_t(\boldsymbol{x})\}_{t \geqslant 0}$ generated by the dynamics specified in (3) are also generated by the *probability flow (PF)-ODE*:

$$\dot{\boldsymbol{X}}_t = \boldsymbol{u}_t(\boldsymbol{X}_t), \ \boldsymbol{X}_0 \sim p_0 \tag{5}$$

The central quantity that allows us to convert between the SDE (3) and the PF-ODE (4) is the gradient of the log-density $\nabla_{\boldsymbol{x}} \log p_t(\boldsymbol{x}) =: \boldsymbol{s}_t(\boldsymbol{x})$, also known as the *score function*. Under mild regularity conditions, knowledge of the score $\boldsymbol{s}(\boldsymbol{x})$ allows for sampling from $p(\boldsymbol{x})$ via Langevin dynamics: the SDE $\mathrm{d}\boldsymbol{X}_t = \frac{1}{2}\boldsymbol{s}(\boldsymbol{x})\mathrm{d}t + \mathrm{d}\boldsymbol{W}_t$ has stationary distribution $p(\boldsymbol{x})$ [46]. While typically one needs to resort to approximations to learn $\boldsymbol{s}$ [55, 46, 47], in setting of the Brownian (and as we will see, the mvOU) bridge, one has closed form expressions for the score and the objective.

**Conditional flow matching** Stated in its original form, let $t \mapsto p_t(\boldsymbol{x})$ be a family of marginals on $t \in [0,1]$ satisfying the continuity equation $\partial_t p_t(\boldsymbol{x}) = -\nabla \cdot (p_t(\boldsymbol{x})\boldsymbol{u}_t(\boldsymbol{x}))$, where $\boldsymbol{u}_t(\boldsymbol{x})$ is a time-dependent vector field. Suppose further that $p_t(\boldsymbol{x})$ admits a representation as a *mixture*

$$p_t(\boldsymbol{x}) = \int \mathrm{d}q(\boldsymbol{z}) p_{t|z}(\boldsymbol{x}|\boldsymbol{z})$$

where $\boldsymbol{z} \sim q(\boldsymbol{z})$ is some latent variable and $t \mapsto p_{t|z}(\boldsymbol{x})$ are called the *conditional probability paths*. We introduce *conditional flow fields* $\boldsymbol{u}_{t|z}$ that generate the conditional probability paths, i.e. $\partial_t p_{t|z}(\boldsymbol{x}) = -\nabla \cdot (p_{t|z}(\boldsymbol{x})\boldsymbol{u}_{t|z}(\boldsymbol{x}|z))$. Then, the insight presented in [30, Theorem 1] is that, in fact,

$$\boldsymbol{u}_t(\boldsymbol{x}) = \mathbb{E}_{\boldsymbol{z} \sim q(\boldsymbol{z})} \left[ \frac{p_{t|z}(\boldsymbol{x}|\boldsymbol{z})}{p_t(\boldsymbol{x})} \boldsymbol{u}_{t|z}(\boldsymbol{x}|\boldsymbol{z}) \right],$$

as can be easily verified by checking the continuity equation. When $p_t(\boldsymbol{x})$ is not tractable but $q(\boldsymbol{z})$, $p_{t|z}(\boldsymbol{x})$ and $\boldsymbol{u}_{t|z}(\boldsymbol{x})$ are, as is the case for the Schrödinger bridge (2), conditional flow matching is useful: Lipman et al. [30, Theorem 2] prove that minimising

$$L_{\mathrm{CFM}}(\theta) = \mathbb{E}_{t \sim U[0,1], \boldsymbol{z} \sim q} \, \mathbb{E}_{\boldsymbol{x} \sim p_{t|z}} \| \boldsymbol{v}_{\theta,t}(\boldsymbol{x}) - \boldsymbol{u}_{t|z}(\boldsymbol{x}) \|^2 \tag{6}$$

is equivalent to regression on the true (marginal) vector field $\boldsymbol{u}_t$.

**Simulation-free Schrödinger bridges** Previous work [52, 39] has exploited the connection (2) between the dynamic and static SBP problems to create solutions to (SBP-dyn) without the need to numerically simulate SDEs. In particular, [52] proposed to utilise score matching and flow matching simultaneously to learn the probability flow and score of the dynamical SBP with a Brownian reference process. Crucially, they exploit availability of closed form expressions for the Brownian bridge and minimise the objective

$$L_{[\mathrm{SF}]^2 M}(\theta, \varphi) = \mathop{\mathbb{E}}_{\substack{t,(x_0,x_1) \sim U[0,1] \otimes \pi \\ \boldsymbol{z} \sim p_{t|(x_0,x_1)}}} \left[ \| \boldsymbol{v}_{\theta,t}(\boldsymbol{z}) - \boldsymbol{u}_{t|(x_0,x_1)}(\boldsymbol{z}) \|^2 + \lambda_t \| \boldsymbol{s}_{\varphi,t}(\boldsymbol{z}) - \boldsymbol{s}_{t|(x_0,x_1)}(\boldsymbol{z}) \|^2 \right], \tag{7}$$

where $\pi = \mathbb{P}_{01}^\star$ is the optimal coupling solving (SBP-static). In the above $\boldsymbol{u}_{t|(x_0,x_1)}, \boldsymbol{s}_{t|(x_0,x_1)}$ are, respectively, the flow and score of the Brownian bridge conditioned on $(x_0, x_1)$, for which there are readily accessible closed form expressions [52, Eq. 8]: $\boldsymbol{u}_t(\boldsymbol{x}) = \frac{1-2t}{2t(1-t)}(\boldsymbol{x} - \overline{\boldsymbol{x}}_t) + (\boldsymbol{x}_1 - \boldsymbol{x}_0)$ and $\boldsymbol{s}_t(\boldsymbol{x}) = (\sigma^2 t(1-t))^{-1}(\overline{\boldsymbol{x}}_t - \boldsymbol{x})$ with $\overline{\boldsymbol{x}}_t = t\boldsymbol{x}_1 + (1-t)\boldsymbol{x}_0$. Motivating the use of the conditional objective (7) in practice, the authors prove [52, Theorem 3.2] that minimising (7) is equivalent to regressing against the SB flow and score. Since $\mathbb{P}_{01}^\star$ can be obtained by solving an static entropic optimal transport problem.

## 2.3 Related work

The generalisation of dynamical optimal transport and the Schrödinger bridge to linear reference dynamics has been studied [11, 9, 10] from the viewpoint of stochastic optimal control. However, as was pointed out by [5], these studies have primarily focused on theoretical aspects of the problem, such as existence and uniqueness. In particular, the result for the Gaussian case [9] is in terms of a system of coupled matrix differential equations and does not lend itself to straightforward computation. More generally, forward-backward SDEs corresponding to a continuous-time iterative proportional fitting scheme [53, 44] have been proposed for general reference processes.

For the Gaussian case, the availability of closed form solutions is by now classical [36, 49], and more recent work provides analytical characterisations of Gaussian *entropy-regularised* transport [5, 22, 32]. [5] provides explicit formulae that characterise the bridge marginals as well as the force (bridge control) for a Brownian reference as well as a class of *scalar* OU references. To the authors' knowledge, however, all explicit formulae for Gaussian Schrödinger bridges assume either Brownian or scalar OU dynamics. The application of flow matching [30, 3] as a simulation-free technique to learn Schrödinger bridges was proposed in [52], but the authors restrict consideration to a Brownian reference process. Recently [33] apply flow matching for stochastic linear control systems, however they are concerned with interpolating distributions and do not study the SBP.

Finally, concurrent work [38] proposes a related flow matching scheme for approximately solving the $\mathbb{Q}$-SBP where the reference process $\mathbb{Q}$ is described by a general, potentially non-linear diffusion. This comes however at the cost of using learned neural approximations for bridges and log-transition densities of $\mathbb{Q}$. Our work treats the complementary case of linear reference dynamics, in which we may avail ourselves of analytical formulae for these quantities.

# 3 Schrödinger bridges for non-equilibrium systems

## 3.1 Multivariate Ornstein-Uhlenbeck bridges

As is evident from (2), the dynamical formulation of the $\mathbb{Q}$-SBP amounts to solution of the *static* $\mathbb{Q}$-SBP (SBP-static) together with characterisation of the $\mathbb{Q}$-bridges, i.e. the reference process $\mathbb{Q}$ conditioned on initial and terminal endpoints. For $\mathbb{Q}$ described by a linear SDE of the form (1), explicit formulae for the $\mathbb{Q}$-bridges are available and we state these in the form of the following theorem.

**Theorem 1** (SDE characterisation of mvOU bridge, adapted from results in [8]). *Consider the $d$-dimensional mvOU process* (1). *Conditioning on* $(0, \boldsymbol{x}_0)$ *and* $(T, \boldsymbol{x}_T)$, *the* bridges *of this process* $\boldsymbol{Y}_t = \boldsymbol{X}_t | \{\boldsymbol{X}_0 = \boldsymbol{x}_0, \boldsymbol{X}_T = \boldsymbol{x}_T\}, 0 \leqslant t \leqslant T$ *are generated by the SDE*

$$\mathrm{d}\boldsymbol{Y}_t = \left(\boldsymbol{A}(\boldsymbol{Y}_t - \boldsymbol{m}) + \boldsymbol{c}_{t|(\boldsymbol{x}_0, \boldsymbol{x}_1)}\right)\mathrm{d}t + \boldsymbol{\sigma}\,\mathrm{d}\boldsymbol{B}_t, \quad \boldsymbol{c}_{t|(\boldsymbol{x}_0, \boldsymbol{x}_1)} = -\boldsymbol{\Lambda}_t^{-1}(\boldsymbol{Y}_t - \boldsymbol{k}_t). \tag{8}$$

*where* $\boldsymbol{\Lambda}_t = \int_0^{T-t} e^{-s\boldsymbol{A}} e^{-s\boldsymbol{A}^\top}\mathrm{d}s$ *and* $\boldsymbol{k}_t = e^{-(T-t)\boldsymbol{A}}(\boldsymbol{x}_T - \boldsymbol{m}) + \boldsymbol{m}$.

We remark that this characterisation can be found implicitly in the results of [8]; however, their results were conceived for a more general setting of time-varying coefficients and hence is stated in terms of the state transition matrix, which does not in general admit an explicit formula. In our case, computation of the control term $\boldsymbol{\Lambda}_t$ relies upon a unidimensional integral as a key quantity and crucially this is independent of the endpoints, meaning it incurs a one-off computational cost. Although the derivation of Theorem 1 follows the same procedure as [8], we provide details in the Appendix since the work of Chen and Georgiou [8] is written for a control audience. Using the SDE characterisation (8) of the $\mathbb{Q}$-bridge and the fact that it remains a Gaussian process, we obtain explicit formulae for the *score* and *flow* fields as well as mean and covariance functions of the $\mathbb{Q}$-bridge. One can furthermore check that the Brownian bridge formulae are recovered as a special case.

**Theorem 2** (Score and flow for multivariate Ornstein-Uhlenbeck bridge). *For the mvOU bridge conditioned on* $(0, \boldsymbol{x}_0), (T, \boldsymbol{x}_T)$, *denote by* $p_{t|(\boldsymbol{x}_0, \boldsymbol{x}_T)}$ *the density at time* $0 < t < T$. *Then, score function and probability flow of the bridge are respectively*

$$\boldsymbol{s}_{t|(\boldsymbol{x}_0, \boldsymbol{x}_T)}(\boldsymbol{x}) = \nabla_{\boldsymbol{x}} \log p_{t|(\boldsymbol{x}_0, \boldsymbol{x}_T)}(\boldsymbol{x}) = \boldsymbol{\Sigma}_{t|(\boldsymbol{x}_0, \boldsymbol{x}_T)}^{-1}\left(\boldsymbol{\mu}_{t|(\boldsymbol{x}_0, \boldsymbol{x}_T)} - \boldsymbol{x}\right) \tag{9}$$

$$\boldsymbol{u}_{t|(\boldsymbol{x}_0, \boldsymbol{x}_T)}(\boldsymbol{x}) = \boldsymbol{A}(\boldsymbol{x} - \boldsymbol{m}) + \boldsymbol{c}_{t|(\boldsymbol{x}_0, \boldsymbol{x}_T)}(\boldsymbol{x}) - \tfrac{1}{2}\boldsymbol{\sigma}\boldsymbol{\sigma}^\top \boldsymbol{s}_{t|(\boldsymbol{x}_0, \boldsymbol{x}_T)}(\boldsymbol{x}). \tag{10}$$

*where* $\boldsymbol{c}_{t|(\boldsymbol{x}_0, \boldsymbol{x}_T)}$ *is the bridge control from Theorem 1 and* $(\boldsymbol{\mu}_{t|(\boldsymbol{x}_0, \boldsymbol{x}_T)}, \boldsymbol{\Sigma}_{t|(\boldsymbol{x}_0, \boldsymbol{x}_T)})$ *are the mean and covariance of* $p_{t|(\boldsymbol{x}_0, \boldsymbol{x}_T)}$:

$$\boldsymbol{\Sigma}_{t|(\boldsymbol{x}_0, \boldsymbol{x}_T)} = \boldsymbol{\Phi}_t - \boldsymbol{\Phi}_t e^{(T-t)\boldsymbol{A}^\top} \boldsymbol{\Phi}_T^{-1} e^{(T-t)\boldsymbol{A}} \boldsymbol{\Phi}_t =: \boldsymbol{\Omega}_t, \tag{11}$$

$$\boldsymbol{\mu}_{t|(\boldsymbol{x}_0, \boldsymbol{x}_T)} = \boldsymbol{\mu}_t^{\boldsymbol{x}_0} + \boldsymbol{\Phi}_t e^{(T-t)\boldsymbol{A}^\top} \boldsymbol{\Phi}_T^{-1}(\boldsymbol{x}_T - \boldsymbol{\mu}_T^{\boldsymbol{x}_0}). \tag{12}$$

*In the above,* $\boldsymbol{\mu}_t^{\boldsymbol{x}}$ *denotes the mean at time* $t$ *of an* unconditioned *mvOU process started from* $(0, \boldsymbol{x})$, *and the function* $\boldsymbol{\Phi}_t = \int_0^t e^{(t-s)\boldsymbol{A}} \boldsymbol{\sigma}\boldsymbol{\sigma}^\top e^{(t-s)\boldsymbol{A}^\top} \mathrm{d}s$ *is independent of the endpoints* $(\boldsymbol{x}_0, \boldsymbol{x}_T)$. *Consequently,* $\boldsymbol{\Sigma}_{t|(\boldsymbol{x}_0, \boldsymbol{x}_T)}$ *depends only on* $t$, *so we write* $\boldsymbol{\Omega}_t = \boldsymbol{\Sigma}_{t|(\cdot, \cdot)}$ *to make this explicit.*

While all our theoretical results are stated in terms of a generic diffusion matrix $\boldsymbol{\sigma}$, we remark that in practice one can always assume $\boldsymbol{\sigma}$ to be diagonal. This is formalised in the following informal lemma, for which we provide a formal statement and proof in Appendix A.1.

**Lemma 1.** *Up to a orthogonal change of coordinates, any linear SDE with generic drift and diffusion matrix, is equal in its law to another linear SDE with a transformed drift and diagonal diffusion matrix.*

## 3.2 The Gaussian case

We derive explicit formulae for Gaussian Schrödinger bridges with *general* linear reference dynamics. Our results generalise those in [5] to mvOU reference processes, and we verify in the appendix that we recover several results of [5, Theorem 3, Table 1] as special cases. Both derivations use the disintegration property (2) together with analytical expressions for the optimal coupling and bridges.

**Theorem 3** (Characterisation of mvOU-GSB). *Consider a mvOU reference process $\mathbb{Q}$ described by* (1) *and Gaussian initial and terminal marginals at times $t = 0$ and $t = T$ respectively:*

$$\rho_0 = \mathcal{N}(\boldsymbol{a}, \boldsymbol{\mathcal{A}}), \rho_T = \mathcal{N}(\boldsymbol{b}, \boldsymbol{\mathcal{B}}).$$

In what follows, we write $\mathbf{\Sigma}_t$ for the covariance of the unconditioned *process* (1) *started at a point mass. Define the* transformed *means and covariances*

$$\overline{\boldsymbol{a}} = \mathbf{\Sigma}_T^{-1/2}(e^{TA}(\boldsymbol{a} - \boldsymbol{m}) + \boldsymbol{m}), \qquad \overline{\boldsymbol{\mathcal{A}}} = \mathbf{\Sigma}_T^{-1/2}e^{TA}\boldsymbol{\mathcal{A}}e^{TA^\top}\mathbf{\Sigma}_T^{-1/2}$$

$$\overline{\boldsymbol{b}} = \mathbf{\Sigma}_T^{-1/2}\boldsymbol{b}, \qquad\qquad\qquad \overline{\boldsymbol{\mathcal{B}}} = \mathbf{\Sigma}_T^{-1/2}\boldsymbol{\mathcal{B}}\mathbf{\Sigma}_T^{-1/2},$$

*and let* $\overline{\boldsymbol{\mathcal{C}}}$ *be the cross-covariance term of the Gaussian entropic optimal transport plan [21, Theorem 1] between* $\overline{\rho}_0 = \mathcal{N}(\overline{\boldsymbol{a}}, \overline{\boldsymbol{\mathcal{A}}})$ *and* $\overline{\rho}_T = \mathcal{N}(\overline{\boldsymbol{b}}, \overline{\boldsymbol{\mathcal{B}}})$ *with unit diffusivity, i.e.*

$$\overline{\boldsymbol{\mathcal{C}}} = \overline{\boldsymbol{\mathcal{A}}}^{1/2}\left(\overline{\boldsymbol{\mathcal{A}}}^{1/2}\overline{\boldsymbol{\mathcal{B}}}\,\overline{\boldsymbol{\mathcal{A}}}^{1/2} + \tfrac{1}{4}\boldsymbol{I}\right)^{1/2}\overline{\boldsymbol{\mathcal{A}}}^{-1/2} - \tfrac{1}{2}\boldsymbol{I}. \tag{13}$$

*Setting* $\boldsymbol{\Gamma}_t = \boldsymbol{\Phi}_t e^{(T-t)A^\top}\boldsymbol{\Phi}_T^{-1}$ *for brevity, define*

$$\mathfrak{A}_t = \left(e^{-(T-t)A} - \boldsymbol{\Gamma}_t\right)\mathbf{\Sigma}_T^{1/2}, \quad \mathfrak{B}_t = \boldsymbol{\Gamma}_t\mathbf{\Sigma}_T^{1/2}, \quad \mathfrak{c}_t = \left(\boldsymbol{I} - e^{-(T-t)A}\right)\boldsymbol{m} \tag{14}$$

*Then the* $\mathbb{Q}$-GSB *is a Markovian Gaussian process for* $0 \leqslant t \leqslant T$ *with mean and covariance*

$$\boldsymbol{\nu}_t = \mathbb{E}[\boldsymbol{Y}_t] = \mathfrak{A}_t\overline{\boldsymbol{a}} + \mathfrak{B}_t\overline{\boldsymbol{b}} + \mathfrak{c}_t, \tag{15}$$

$$\boldsymbol{\Xi}_{st} = \mathbb{V}[\boldsymbol{Y}_s, \boldsymbol{Y}_t] = \boldsymbol{\Omega}_{st} + \mathfrak{A}_s\overline{\boldsymbol{\mathcal{A}}}\mathfrak{A}_t^\top + \mathfrak{A}_s\overline{\boldsymbol{\mathcal{C}}}\mathfrak{B}_t^\top + \mathfrak{B}_s\overline{\boldsymbol{\mathcal{C}}}^\top\mathfrak{A}_t^\top + \mathfrak{B}_s\overline{\boldsymbol{\mathcal{B}}}\mathfrak{B}_t^\top. \tag{16}$$

*Furthermore, the* $\mathbb{Q}$-GSB (SBP-dyn) *is described by a SDE*

$$\mathrm{d}\boldsymbol{Y}_t = \left(\dot{\boldsymbol{\nu}}_t + \boldsymbol{S}_t^\top\boldsymbol{\Xi}_t^{-1}(\boldsymbol{Y}_t - \boldsymbol{\nu}_t)\right)\mathrm{d}t + \boldsymbol{\sigma}\,\mathrm{d}\boldsymbol{B}_t, \qquad \boldsymbol{Y}_t \sim \mathcal{N}(\boldsymbol{a}, \boldsymbol{\mathcal{A}}). \tag{17}$$

*where* $\boldsymbol{S}_t = (\partial_{t'}\boldsymbol{\Omega}_{t,t'})(t) + \mathfrak{A}_t\overline{\boldsymbol{\mathcal{A}}}\dot{\mathfrak{A}}_t^\top + \mathfrak{A}_t\overline{\boldsymbol{\mathcal{C}}}\dot{\mathfrak{B}}_t^\top + \mathfrak{B}_t\overline{\boldsymbol{\mathcal{C}}}^\top\dot{\mathfrak{A}}_t^\top + \mathfrak{B}_t\overline{\boldsymbol{\mathcal{B}}}\dot{\mathfrak{B}}_t^\top.$

Importantly, we show in the Appendix that we recover from the results of Theorem 3 two prior results of [5] – specifically, we obtain results for a Brownian and scalar OU reference process as special cases. Furthermore, in [5] the authors remark that the quantity $\boldsymbol{S}_t^\top\boldsymbol{\Xi}_t^{-1}$, determining the GSB drift, is *symmetric* although $\boldsymbol{S}_t^\top$ is asymmetric in general. The interpretation of this result is that the GSB is driven by a time-varying potential and is thus of gradient type. By contrast, we observe empirically that the mvOU-GSB drift is *asymmetric* when the underlying reference process $\mathbb{Q}$ is not of gradient type.

### 3.3 The Non-Gaussian case – score and flow matching

When the endpoints $(\rho_0, \rho_T)$ are non-Gaussian, direct analytical characterisation of the solution to (SBP-dyn) is out of reach. Here we show that combining the exact characterisation of the $\mathbb{Q}$-bridges (Section 3.1) with score and flow matching (Section 2.2) yields a simulation-free estimator of the Schrödinger bridge when $\mathbb{Q}$ is a mvOU process (1). Specifically, we propose to first exploit the static problem (SBP-static) which can be efficiently solved via Sinkhorn-Knopp iterations [14] with an analytically tractable cost function. Next, we use the property (2) together with the characterisation of $\mathbb{Q}$-bridges presented in Theorem 2. The approach of [52], which considered a Brownian reference, thus arises as a special case for $\boldsymbol{A} = 0, \boldsymbol{m} = 0, \boldsymbol{\sigma} = \sigma\mathbf{I}$.

**Proposition 1.** *When* $\mathbb{Q}$ *is a mvOU process* (1)*,* (SBP-static) *shares the same minimiser as*

$$\min_{\pi\in\Pi(\rho_0,\rho_T)} \tfrac{1}{2}\int(\boldsymbol{x}_T - \boldsymbol{\mu}_T^{\boldsymbol{x}_0})^\top\mathbf{\Sigma}_T^{-1}(\boldsymbol{x}_T - \boldsymbol{\mu}_T^{\boldsymbol{x}_0})\,\mathrm{d}\pi(\boldsymbol{x}_0, \boldsymbol{x}_T) + \mathrm{H}(\pi|\rho_0 \otimes \rho_T). \tag{18}$$

We remind that $\boldsymbol{x} \mapsto \boldsymbol{\mu}_T^{\boldsymbol{x}_0}$ is affine and computation of $\mathbf{\Sigma}_T$ relies on a one-off time integration that does not depend on the endpoints (Section A.1). Equipped with $\pi$ the solution to (18), we parameterise the unknown probability flow and score function of (SBP-dyn) by $\boldsymbol{u}_t^\theta(\boldsymbol{x})$ and $\boldsymbol{s}_t^\varphi(\boldsymbol{x})$ where $(\theta, \varphi)$ are trainable parameters. We seek to minimise the loss

$$L(\theta, \varphi) = \mathbb{E}_{\substack{t,(x_0,x_T)\sim U[0,T]\otimes\pi \\ z\sim p_{t|(x_0,x_T)}}}\left[\|\boldsymbol{u}_t^\theta(\boldsymbol{z}) - \boldsymbol{u}_{t|(x_0,x_T)}(\boldsymbol{z})\|^2 + \lambda_t\|\boldsymbol{s}_t^\varphi(\boldsymbol{z}) - \boldsymbol{s}_{t|(x_0,x_T)}(\boldsymbol{z})\|^2\right], \tag{19}$$

In the practical case when $(\rho_0, \rho_T)$ are discrete, sampling from $t, (x_0, x_T) \sim U[0, T] \otimes \pi$ is trivial once (18) is solved. Sampling from $p_{t|(\boldsymbol{x}_0,\boldsymbol{x}_T)}$ and evaluating $\boldsymbol{u}_{t|(x_0,x_T)}, \boldsymbol{s}_{t|(x_0,x_T)}$ amounts to invoking the results of Theorem 2. Again, computations of these quantities involve one-off solution of a 1D matrix-valued integral: this can be solved to a desired accuracy and then cached and queried so it poses negligible computational cost, and this is what we do in practice. As in the Brownian case, the following result establishes the connection of the conditional score and flow matching loss (19) to (SBP-dyn), and is a generalisation and restatement of results from [52, Theorem 3.2, Proposition 3.4].

**Algorithm 1** mvOU-OTFM: score and flow matching for mvOU-Schrödinger Bridges

---

**Input:** Samples $\{x_i\}_{i=1}^N$, $\{x_j'\}_j^{N'}$ from source and target distribution at times $t = 0, T$, mvOU reference parameters $(A, m, D = \frac{1}{2}\sigma\sigma^\top)$, batch size $B$.
**Initialise:** Probability flow field $u_t^\theta(x)$, score field $s_t^\varphi(x)$.
$\hat{\rho}_0 \leftarrow N^{-1}\sum_{i=1}^N \delta_{x_i}, \hat{\rho}_1 \leftarrow N'^{-1}\sum_{i=1}^{N'}\delta_{x_i'}$            *Form empirical marginals*
$\pi \leftarrow \mathtt{sinkhorn}(C, \hat{\rho}_0, \hat{\rho}_1, \mathtt{reg} = 1.0)$            *Solve* (18) *with log-kernel cost*
**while** not converged **do**
    $\{(x_i, x_i')\}_{i=1}^B \leftarrow \mathtt{sample}(\pi), \ \{t_i\}_{i=1}^B \leftarrow \mathtt{sample}(U[0,T])$
    **for** $1 \leqslant j \leqslant B$ **do**
        $z \leftarrow \mathcal{N}(\mu_{t_j|(x_j, x_j')}, \Sigma_{t_j|(x_j, x_j')})$            *Sample from mvOU-bridge (Thm. 2)*
        $\ell_j \leftarrow \|u_{t_j}^\theta(z) - u_{t_j|(x_j, x_j')}\|^2 + \lambda_{t_j}\|s_{t_j}^\varphi(z) - s_{t_j|(x_j, x_j')}\|^2$     *(Flow, score)-matching loss*
    **end for**
    $L = B^{-1}\sum_{j=1}^B \ell_j$
    $(\theta, \varphi) \leftarrow \mathtt{Step}(\nabla_\theta L, \nabla_\varphi L).$
**end while**

---

**Theorem 4** (mvOU-OTFM solves (SBP-dyn))**.** *If $p_t(x) > 0$ for all $(t, x)$,* (19) *shares the same gradients as the* unconditional *loss:* $L(\theta, \varphi) = \mathbb{E}_{t \in U[0,T]}\mathbb{E}_{z \sim p_t}[\|u_t^\theta(z) - u_t(z)\|^2 + \lambda_t\|s_t^\varphi(z) - s_t(z)\|^2]$. *Furthermore, if the coupling $\pi$ solves* (SBP-static) *as given in Proposition 1, then for $(u_t^\theta, s_t^\varphi)$ achieving the global minimum of* (19)*, the solution to* (SBP-dyn) *is given by the SDE* $dX_t = (u_t^\theta(X_t) + Ds_t^\varphi(X_t))\, dt + \sigma\, dB_t$.

We stress here that the linearity assumption is only imposed on the reference process $\mathbb{Q}$ in (1), and not on the SBP solution $\mathbb{P}$. In the general non-Gaussian case the mvOU-SBP dynamics will still be nonlinear, much in the same way as for the Brownian SBP.

Further, a non-linear reference dynamics can be connected to linear dynamics of the form (1) by linearisation about a fixed point of the drift. That is, for a smooth reference drift $f(x)$ with $x_0$ a fixed point, $f(x_0) = 0$, one has $f(x) = (\partial_x f)(x_0)(x - x_0) + O(\|x - x_0\|^2)$. This naturally motivates the study of mvOU processes with $A = (\partial_x f)(x_0)$ and $m = x_0$.

## 4   Results

**Gaussian marginals: benchmarking accuracy**    As a first visual illustration of Theorem 3, we show in Fig. 2 the family of marginals solving (SBP-dyn) for the same pair of Gaussian marginals in dimension $d = 10$, where the reference process $\mathbb{Q}$ is taken to be (a)(i) a mvOU process with high-dimensional rotational drift and (b)(i) a standard Brownian motion. While both provide valid interpolations of $\rho_0$ to $\rho_1$, it is immediately clear that the dynamics are completely different. To provide some further insight, we show in (a)(ii) and (b)(ii) the time-dependent vector field generating each bridge. According to Theorem 3, both processes are time-dependent linear SDEs of the form (17). For the mvOU reference the asymmetric nature of the drift matrix is evident, while for the Brownian reference the symmetry of the drift matrix corresponds to potential driven dynamics [5].

In the Gaussian case, our explicit formulae for the OU-GSB allow us to empirically quantify the accuracy of our neural solver. Since existing computational methods for solving the $\mathbb{Q}$-Schrödinger bridge for non-gradient $\mathbb{Q}$ are limited, we compare MVOU-OTFM against Iterative Proportional Maximum Likelihood (IPML) [53] and Neural Lagrangian Schrödinger Bridge (NLSB) [25]. The former is based on a forward-backward SDE characterisation of the $\mathbb{Q}$-SB and alternates between simulation and drift estimation using Gaussian processes, while the latter uses a Neural SDE framework. For $d$ ranging from 2 to 100, we solve (SBP-dyn) with each method and report in Table 1 the average marginal error, measured in the Bures-Wasserstein metric [49] and the average vector field error in $L^2$. In all cases, we find that MVOU-OTFM achieves the highest accuracy among all solvers considered. Among the others, NLSB is generally the most accurate followed by IPML, while BM-OTFM performs the poorest since the reference process is misspecified.

In addition to being more accurate, MVOU-OTFM is extremely fast to train owing to being simulation free, taking approximately 1-2 minutes to train on CPU for $d = 50$, regressing the score and flow networks against (19). NLSB took 15+ minutes to train on GPU requiring backpropagation through

SDE solves, while IPFP requires iterative SDE simulation. On the other hand, our approach directly exploits the exact solution to $\mathbb{Q}$-bridges rather than relying on numerical approximations.

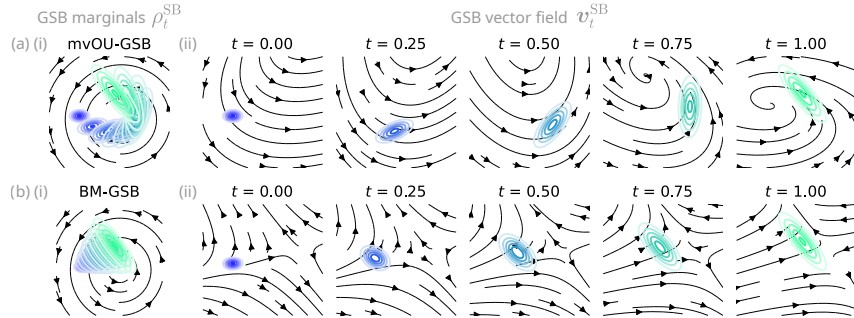

Figure 2: **Gaussian Schrödinger Bridges.** (a) (i) Marginals of the Gaussian Schrödinger bridge ($d = 10$) with mvOU reference (MVOU-GSB) (ii) Time dependent vector field generating the bridge. (b) Same as (a) but for Brownian reference.

| | Marginal error $(\mathrm{BW}_2^2)$ | | | | | Force error $(L^2)$ | | | |
|---|---|---|---|---|---|---|---|---|---|
| $d$ | MVOU-OTFM | BM-OTFM | IPML ($\leftarrow$) | IPML ($\rightarrow$) | NLSB | MVOU-OTFM | BM-OTFM | IPML | NLSB |
| 2 | **0.19**$\pm$**0.17** | 8.40$\pm$0.77 | 5.55$\pm$1.53 | 5.65$\pm$1.41 | 1.21$\pm$0.18 | **3.56**$\pm$**0.25** | 12.23$\pm$0.27 | 10.31$\pm$0.45 | 7.59$\pm$0.33 |
| 5 | **0.23**$\pm$**0.16** | 9.06$\pm$0.66 | 6.34$\pm$5.64 | 3.24$\pm$0.98 | 1.16$\pm$0.26 | **3.66**$\pm$**0.19** | 12.27$\pm$0.12 | 10.08$\pm$0.75 | 7.72$\pm$0.15 |
| 10 | **0.59**$\pm$**0.36** | 8.93$\pm$0.55 | 3.00$\pm$0.73 | 3.00$\pm$0.63 | 1.36$\pm$0.13 | **3.82**$\pm$**0.13** | 12.33$\pm$0.13 | 10.96$\pm$1.11 | 8.15$\pm$0.25 |
| 25 | **0.84**$\pm$**0.22** | 8.81$\pm$1.19 | 6.97$\pm$2.01 | 5.03$\pm$0.60 | 2.97$\pm$1.24 | **4.67**$\pm$**0.16** | 12.54$\pm$0.16 | 12.42$\pm$0.80 | 11.02$\pm$2.41 |
| 50 | **2.21**$\pm$**0.36** | 11.74$\pm$0.37 | 9.03$\pm$0.21 | 8.32$\pm$0.63 | 6.39$\pm$0.13 | **6.25**$\pm$**0.17** | 13.26$\pm$0.15 | 14.14$\pm$1.00 | 14.40$\pm$0.02 |
| 100 | **6.84**$\pm$**0.78** | 15.14$\pm$0.95 | 16.19$\pm$1.87 | 14.38$\pm$0.38 | 17.40$\pm$0.13 | **10.45**$\pm$**0.18** | 15.53$\pm$0.14 | 15.30$\pm$0.39 | 16.49$\pm$0.07 |

Table 1: Marginal error measured in Bures-Wasserstein metric and force error measured in $L^2$ norm for the Schrödinger bridge learned between two Gaussian measures in varying dimension $d$.

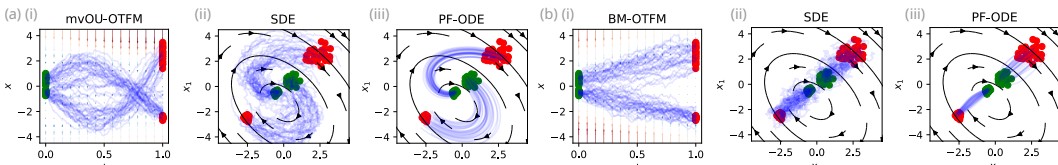

Figure 3: **Non-Gaussian marginals.** (a)(i) Sampled trajectories from the mvOU-SB learned between non-Gaussian marginals in $d = 10$, shown in $(t, x_0)$ coordinates with score field shown in background. (a)(ii, iii) Sampled SDE (stochastic) and PF-ODE (deterministic) trajectories shown in $(x_0, x_1)$ coordinates, and reference drift shown in background. (b) Same as (a) but for Brownian reference.

**Non-Gaussian marginals**  We demonstrate an application of MVOU-OTFM for general non-Gaussian marginals in Fig. 3, where the reference dynamics are the same as those in Fig. 2. Again, the contrast between the mvOU and Brownian reference dynamics is clear, and we show the distinction between sampling of trajectories via the SDE (3) and PF-ODE formulations (4).

**Repressilator dynamics with iterated reference**  As an application to a system of biophysical importance, we consider the repressilator [16], a model of a synthetic gene circuit exhibiting oscillatory, and hence non-equilibrium, dynamics in cells. This system is composed of three genes in a loop, in which each gene represses the activity of the next (Fig. 4(a)). This model, implemented as a SDE (162), was also studied by [44], which introduced a method, "Schrödinger Bridge with Iterative Reference Refinement" (SBIRR) building upon IPFP [53] to solve a series of Schrödinger bridge problems whilst simultaneously learning an improved reference $\mathbb{Q}$ within a parametric family. In particular, they propose to alternate between solving the $\mathbb{Q}$-SBP, using IPFP as a subroutine, and fitting of a parametric global drift describing $\mathbb{Q}$. The same approach can be used with mvOU-OTFM, which we present in Algorithm 2 in the Appendix: we propose to apply Algorithm 1 as a subroutine, and solve a regularised linear regression problem to update the mvOU reference parameters $(\boldsymbol{A}, \boldsymbol{m})$ at each outer step.

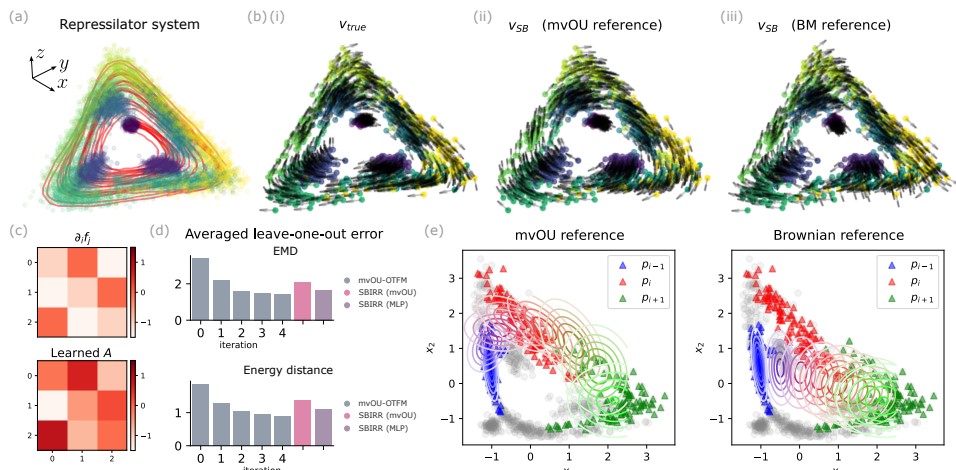

Figure 4: **Repressilator dynamics.** (a) Repressilator population snapshots and trajectories. (b) (i) Ground truth vector field. (ii, iii) Inferred multi-marginal SB vector field with (fitted mvOU, Brownian) references. (c) Ground truth linearisation of system and drift $\boldsymbol{A}$ learned by mvOU-OTFM. (d) Leave-one-out error by iteration. (e) Illustration of leave-one-out interpolation between two example timepoints $p_{i-1}$ (blue), $p_{i+1}$ (green) with learned mvOU reference vs. Brownian reference.

| Error metric | Iterate 0 | Iterate 1 | Iterate 2 | Iterate 3 | Iterate 4 | SBIRR (mvOU) | SBIRR (MLP) |
|---|---|---|---|---|---|---|---|
| EMD | $3.38 \pm 1.52$ | $2.22 \pm 1.12$ | $1.59 \pm 0.66$ | $1.49 \pm 0.64$ | $\mathbf{1.40 \pm 0.57}$ | $2.10 \pm 0.74$ | $1.67 \pm 0.95$ |
| Energy distance | $1.86 \pm 1.06$ | $1.29 \pm 0.86$ | $1.03 \pm 0.65$ | $0.95 \pm 0.58$ | $\mathbf{0.89 \pm 0.55}$ | $1.39 \pm 0.82$ | $1.10 \pm 0.86$ |

Table 2: Repressilator leave-one-out interpolation error for mvOU-OTFM and SBIRR.

We use a similar setup to [44] and sample snapshots for $T = 10$ instants from the repressilator system. We show in Fig. 4 the sampled data along with example trajectories. We run Algorithm 2 for 5 iterations and we compare in Fig. 4(b) the ground truth vector field to the SB vector field, both the fitted mvOU reference (at the final iterate of Alg. 2) and for a Brownian reference (as the first iterate of Alg. 2 or equivalently the output of [52]). As was found by [44, 58], iterated fitting of a global autonomous vector field for the reference process leverages multi-timepoint information and allows reconstruction of dynamics better adapted to the underlying system.

We reason that the "best" mvOU process to describe the dynamics should resemble the linearisation of the repressilator system about its fixed point. Comparing the Jacobian of the repressilator system to the fitted drift matrix $\boldsymbol{A}$ in Fig. 4(c), shows a clear resemblance – in particular the cyclic pattern of activation and inhibition is recovered. For a quantitative assessment of performance, we show in Fig. 4(d) and Table 2 the averaged leave-one-out error for a marginal interpolation task: for each $1 < i < T$, Alg. 2 is run on the $T - 1$ remaining timepoints with timepoint $t_i$ held out. The trained model is evaluated on predicting $t_i$ from $t_{i-1}$. For both the earth mover's distance (EMD) and energy distance [41] we find that prediction accuracy improves with each additional iteration of Alg. 2.

We compare to SBIRR [44] with two choices of reference process: (i) mvOU and (ii) a general reference family parameterised by a feedforward neural network. The former scenario is thus comparable to mvOU-OTFM, whilst the latter considers a wider class of reference processes. We find that both methods offer an improvement on the null reference (iteration 0 of Alg. 2), and as expected SBIRR with a neural reference family outperforms the mvOU reference family. However, mvOU-OTFM achieves a higher accuracy than either of these methods which suggests that, while SBIRR is able to handle general reference processes and is thus more flexible, mvOU-OTFM trains more accurately. Furthermore, mvOU-OTFM is significantly faster, training in 1-2 minutes on CPU. By comparison, training of SBIRR requires at least several minutes with GPU acceleration.

**Dynamics-resolved single cell data** We next apply our framework to a single-cell RNA sequencing (scRNA-seq) dataset [4] of the cell cycle, where partial experimental quantification of the vector field is possible via metabolic labelling, i.e. for each observed cell state $\boldsymbol{x}_i$ we also have a velocity estimate $\hat{\boldsymbol{v}}_i$ [40]. We consider $N = 2793$ cells in $d = 30$ PCA dimensions (Fig. 5(a)(i)). For each cell

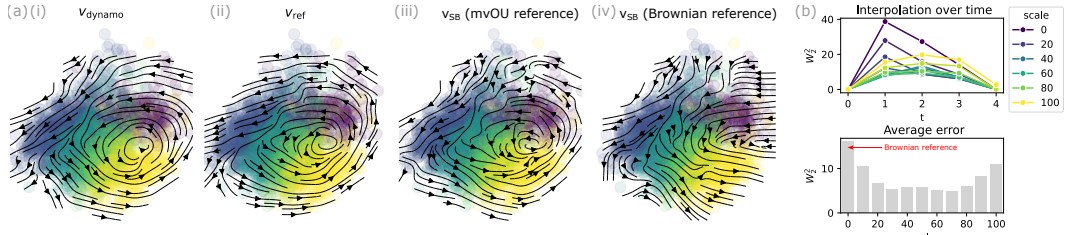

Figure 5: **Cell cycle scRNA-seq.** (a) Streamlines of (i) transcriptomic vector field calculated from metabolic labelling data (ii) Fitted mvOU reference process, and Schrödinger bridge drift $\boldsymbol{v}_{\text{SB}}$ learned by OTFM with (iii) mvOU reference and (iv) Brownian reference. (b) Marginal interpolation error between first and last snapshot as a function of the reference velocity scale parameter, $\gamma$.

an experimental reading of its progression along the cell cycle is available via fluorescence, and cells are binned into $T = 5$ snapshots using this coordinate. From the paired state-velocity data $\{\boldsymbol{x}_i, \hat{\boldsymbol{v}}_i\}$ we fit parameters $(\boldsymbol{A}, \boldsymbol{m})$ of a mvOU process (1) by ridge regression (Fig. 5(a)(ii)). A limitation of "RNA velocity" is that estimation of vector field direction is often significantly more reliable than magnitude [59]. We introduce therefore an additional "scale" parameter $\gamma > 0$ by which to scale the mvOU drift $\gamma\boldsymbol{A}$. We reason that this amounts to a choice of matching the disparate and a priori unknown timescales of the snapshots and velocity data.

In order to select an appropriate scale, for $0 \leqslant \gamma \leqslant 100$ we use the mvOU-GSB to interpolate between the first and last snapshots (see Fig. 7 for an illustration) and calculate the Bures-Wasserstein distance between the interpolant and each intermediate marginal (Fig. 5(b)). From this it is clear that choosing $\gamma$ too small or too large leads to a mvOU-GSB that does not fit well the data, while a broad range of approximately $30 \leqslant \gamma \leqslant 70$ yields good behaviour. We fix $\gamma = 50$ and use mvOU-OTFM to solve a multi-marginal SBP with reference parameters $(\gamma\boldsymbol{A}, \boldsymbol{m})$. From Fig. 5(a)(iii) we see that the expected cyclic behaviour is recovered. On the other hand, using a Brownian reference (equivalently, setting $\gamma = 0$) fails to do so.

Finally, to further demonstrate robustness of our approach to the dimension $d$, we carry out the interpolation analysis for $d = 50, 100$ and provide numerical results in Table 4. These settings of moderately high dimensionality are typical for single cell studies, where PCA projection to 20-50 dimensions is routinely used prior to any downstream analysis. In all cases, we are able to run mvOU-OTFM within minutes on CPU.

## 5 Discussion

Motivated by applications to non-equilibrium dynamics, we study the Schrödinger bridge problem where the reference process is taken to be in a family of multivariate Ornstein-Uhlenbeck (mvOU) processes. These processes are able to capture some key behaviours of non-equilibrium systems whilst still being analytically tractable. Leveraging the tractability, we derive an exact solution of the bridge between Gaussian measures for mvOU reference. We extend our approach to non-Gaussian measures using a score and flow matching approach that avoids the need for costly numerical simulation. We showcase the improvement in both accuracy and speed of our approach in a variety of settings, both low and high-dimensional scenarios.

**Limitations** A key limitation of our method is that our analytical expressions involve matrix inversions and powers of positive definite matrices which scale as $O(d^3)$ with the dimension $d$. The same limitation was identified in [5] as also applying to their approach. In practice, we find that application to problems with $d \leqslant 100$ work well. Scaling our methods to even higher dimensional settings is left to future work, and we expect that tools from the Gaussian process literature will be helpful for this. Regarding flow matching, the link between Alg. 1 and (SBP-dyn) requires the solution to (SBP-static) to be computed exactly. In all our experiments, we achieve this by computing couplings using all available samples. In large-scale settings where the number of data points $N$ is very large or possibly streaming, a common approach is to use *minibatch* couplings [50]. In such settings, the use of minibatches results in learned flows being biased. An in-depth discussion of this is provided in [24], which advocates for the infrequent computation of OT couplings on large batch sizes rather than the commonly used per-iteration minibatch OT.

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

# A  Theoretical results

| Quantity | Description | Introduced in |
|---|---|---|
| $\boldsymbol{A} \in \mathbb{R}^{d \times d}, \boldsymbol{m} \in \mathbb{R}^d$ | Multivariate Ornstein-Uhlenbeck drift parameters | Eq. (1) |
| $\boldsymbol{D} = \frac{1}{2}\boldsymbol{\sigma}\boldsymbol{\sigma}^\top$ | Multivariate Ornstein-Uhlenbeck diffusivity | Eq. (1) |
| $\mathbb{P}$ | Schrödinger bridge path measure | Section 2.1 |
| $\mathbb{Q}$ | Reference process path measure | Section 2.1 |
| $\boldsymbol{\mu}_t^{\boldsymbol{x}_0}, \boldsymbol{\Sigma}_t$ | Unconditional mean & cov. of mvOU process started at $\boldsymbol{x}_0$ | Section 3.1 |
| $\boldsymbol{p}_{t\|(\boldsymbol{x}_0,\boldsymbol{x}_T)}$ | Conditional density of mvOU bridge | Section 3.1 |
| $\boldsymbol{c}_{t\|(\boldsymbol{x}_0,\boldsymbol{x}_T)}$ | mvOU bridge control between $(\boldsymbol{x}_0, \boldsymbol{x}_T)$ | Section 3.1 |
| $\boldsymbol{\mu}_{t\|(\boldsymbol{x}_0,\boldsymbol{x}_T)},$ $\boldsymbol{\Sigma}_{t\|(\boldsymbol{x}_0,\boldsymbol{x}_T)} = \boldsymbol{\Omega}_t$ | Conditional mean and covariance of mvOU bridge | Section 3.1 |
| $\boldsymbol{\Omega}_{st}$ | Conditional covariance process of mvOU bridge | Section 3.1 |
| $\boldsymbol{u}_{t\|(\boldsymbol{x}_0,\boldsymbol{x}_T)}, \boldsymbol{s}_{t\|(\boldsymbol{x}_0,\boldsymbol{x}_T)}$ | Conditional probability flow and score field of mvOU bridge | Section 3.1 |
| $\mathcal{N}(\boldsymbol{a}, \boldsymbol{\mathcal{A}}), \mathcal{N}(\boldsymbol{b}, \boldsymbol{\mathcal{B}})$ | Initial and terminal mvOU-GSB marginals | Section 3.2 |
| $\mathcal{N}(\overline{\boldsymbol{a}}, \overline{\boldsymbol{\mathcal{A}}}), \mathcal{N}(\overline{\boldsymbol{b}}, \overline{\boldsymbol{\mathcal{B}}})$ | Transformed mvOU-GSB marginals | Section 3.2 |
| $\overline{\mathcal{C}}$ | Cross-covariance of entropic transport plan | Section 3.2 |
| $\mathfrak{A}_t, \mathfrak{B}_t, \mathfrak{c}_t$ | Key quantities for the mvOU-GSB | Section 3.2 |
| $\boldsymbol{\nu}_t$ | mvOU-GSB mean process | Section 3.2 |
| $\boldsymbol{\Xi}_t, \boldsymbol{\Xi}_{st}$ | mvOU-GSB variance and covariance process | Section 3.2 |
| $\boldsymbol{S}_t^\top \boldsymbol{\Xi}_t^{-1}$ | SDE drift matrix of mvOU-GSB | Section 3.2. |

Table 3: Glossary of some key notations and quantities used in the statements of our theoretical results.

## A.1  Some calculations on multivariate Ornstein-Uhlenbeck processes

For convenience, we first collect some results about multivariate Ornstein-Uhlenbeck processes of the form (1), a detailed discussion of mvOU processes can be found in e.g. [54]. For a *time-invariant* process with drift matrix $\boldsymbol{A}$ and diffusion $\boldsymbol{\sigma}$, we have that

$$(\boldsymbol{X}_t | \boldsymbol{X}_0 = \boldsymbol{x}_0) \sim \mathcal{N}(\boldsymbol{\mu}_t, \boldsymbol{\Sigma}_t),$$

where

$$\boldsymbol{\mu}_t = e^{t\boldsymbol{A}}\boldsymbol{x}_0, \quad \boldsymbol{\Sigma}_t = \int_0^t e^{(t-\tau)\boldsymbol{A}}\boldsymbol{\sigma}\boldsymbol{\sigma}^\top e^{(t-\tau)\boldsymbol{A}^\top}\,\mathrm{d}\tau. \tag{20}$$

If $\boldsymbol{X}_0 \sim \mathcal{N}(\boldsymbol{\mu}_0, \boldsymbol{\Sigma}_0)$ and writing $\boldsymbol{X}_{0t} = \boldsymbol{X}_0 \oplus \boldsymbol{X}_t$, some tedious but straightforward applications of conditional expectations and the tower property reveal that

$$\boldsymbol{X}_{0t} \sim \mathcal{N}(\boldsymbol{\mu}_{0t}, \boldsymbol{\Sigma}_{0t}), \quad \boldsymbol{\mu}_{0t} = \boldsymbol{\mu}_0 \oplus e^{t\boldsymbol{A}}\boldsymbol{\mu}_0, \quad \boldsymbol{\Sigma}_{0t} = \begin{bmatrix} \boldsymbol{\Sigma}_0 & \boldsymbol{\Sigma}_0 e^{t\boldsymbol{A}^\top} \\ e^{t\boldsymbol{A}}\boldsymbol{\Sigma}_0 & \boldsymbol{\Sigma}_t + e^{t\boldsymbol{A}}\boldsymbol{\Sigma}_0 e^{t\boldsymbol{A}^\top} \end{bmatrix}. \tag{21}$$

More generally, for a process $\mathrm{d}\boldsymbol{X}_t = (\boldsymbol{A}\boldsymbol{X}_t + \boldsymbol{b})\mathrm{d}t + \boldsymbol{\sigma}\mathrm{d}\boldsymbol{B}_t$, one has:

$$\boldsymbol{\mu}_t = e^{t\boldsymbol{A}}\boldsymbol{x}_0 + (e^{t\boldsymbol{A}} - \boldsymbol{I})(\boldsymbol{A}^{-1}\boldsymbol{b}), \quad \boldsymbol{\Sigma}_t = \int_0^t e^{(t-s)\boldsymbol{A}}\boldsymbol{\sigma}\boldsymbol{\sigma}^\top e^{(t-s)\boldsymbol{A}^\top}\,\mathrm{d}s. \tag{22}$$

The expressions for the covariance of the mvOU process can be obtained from the fact that the covariance evolves following a *Lyapunov equation*. For the case of constant coefficients, we state the following result.

**Lyapunov equation solution**  It is easy to verify that $\dot{\boldsymbol{G}}_t = \boldsymbol{A}\boldsymbol{G}_t + \boldsymbol{G}_t\boldsymbol{A}^\top + \boldsymbol{Q}_t$ has solution

$$\boldsymbol{G}_t = \int_0^t e^{(t-s)\boldsymbol{A}}\boldsymbol{Q}_s e^{(t-s)\boldsymbol{A}^\top}\,\mathrm{d}s.$$

## A.2  Bridges of multivariate Ornstein-Uhlenbeck processes

This problem has been studied by Chen et al. in [8], however the material therein is geared towards a control audience and considers a more general case where all coefficients are time-dependent. We will re-derive the results that we will need for processes of the form (1). Additionally, while in practice we typically use $\boldsymbol{\sigma} = \sigma\boldsymbol{I}$, we state some results in the general setting of non-isotropic noise.

**Derivation of the dynamical OU-bridge (Theorem 1)**  Consider a mvOU process

$$\mathrm{d}\boldsymbol{X}_t = (\boldsymbol{A}\boldsymbol{X}_t + \boldsymbol{b})\,\mathrm{d}t + \boldsymbol{\sigma}\,\mathrm{d}\boldsymbol{B}_t. \tag{23}$$

Now form the *controlled* version pinned at $(0, \boldsymbol{x}_0)$ and $(T, \boldsymbol{x}_T)$, where $\boldsymbol{u}_t(\boldsymbol{X}_t)$ is an additional force arising from the conditioning of the process at an endpoint:

$$\mathrm{d}\boldsymbol{X}_t = (A\boldsymbol{X}_t + \boldsymbol{b} + \boldsymbol{u}_t(\boldsymbol{X}_t))\,\mathrm{d}t + \boldsymbol{\sigma}\,\mathrm{d}\boldsymbol{B}_t. \tag{24}$$

The main result that we need is that $\boldsymbol{u}_t(\boldsymbol{x})$ is itself a linear time-dependent field whose coefficients are independent of $(\boldsymbol{x}_0, \boldsymbol{x}_T)$. In the above we consider the simplest case of classical optimal control, where the control is *directly* added to the system drift and there is no unobserved system.

Consider the system started at $(0, \boldsymbol{x}_0)$ and pinned to $(T, \boldsymbol{x}_T)$. For now, we make all statements for time-dependent coefficients and the diffusion is non-isotropic. We then form the Lagrangian:

$$\min_{\boldsymbol{u}} \mathbb{E}\left\{\int_0^T \|\boldsymbol{u}_t\|^2 \mathrm{d}t + \sup_{\lambda} \lambda \|\boldsymbol{X}_T - \boldsymbol{x}_T\|^2\right\} = \sup_{\lambda} \min_{\boldsymbol{u}} \mathbb{E}\left\{\int_0^T \|\boldsymbol{u}_t\|^2 \mathrm{d}t + \lambda \|\boldsymbol{X}_T - \boldsymbol{x}_T\|^2\right\}$$

Fixing some $\lambda > 0$, the resulting problem is known as a *linear-quadratic-Gaussian* problem [29] in the control literature:

$$\min_{\boldsymbol{u}} \mathbb{E}\left\{\frac{1}{2}\int_0^T \|\boldsymbol{u}_t\|^2 \mathrm{d}t + \frac{\lambda}{2}\|\boldsymbol{X}_T - \boldsymbol{x}_T\|^2\right\}, \tag{25}$$

for which we can write a HJB equation:

$$0 = \partial_t V_t(\boldsymbol{x}) + \min_{u}\left\{\widetilde{\mathcal{A}}_t[V_t](\boldsymbol{x}, \boldsymbol{u}_t(\boldsymbol{x})) + \frac{1}{2}\|\boldsymbol{u}_t(\boldsymbol{x})\|^2\right\}, \tag{26}$$

subject to the terminal boundary condition $V_T(\boldsymbol{x}) = \frac{\lambda}{2}\|\boldsymbol{x} - \boldsymbol{x}_T\|^2$, and the operators $\widetilde{\mathcal{A}}, \mathcal{A}$ are defined as

$$\widetilde{\mathcal{A}}_t[f] = \langle \boldsymbol{A}_t\boldsymbol{x} + \boldsymbol{b}_t + \boldsymbol{u}_t, \nabla_x f\rangle + \frac{1}{2}\nabla_x \cdot (\boldsymbol{\sigma}_t\boldsymbol{\sigma}_t^\top \nabla_x f), \tag{27}$$

$$\mathcal{A}_t[f] = \langle \boldsymbol{A}_t\boldsymbol{x} + \boldsymbol{b}_t, \nabla_x f\rangle + \frac{1}{2}\nabla_x \cdot (\boldsymbol{\sigma}_t\boldsymbol{\sigma}_t^\top \nabla_x f), \tag{28}$$

these are the generators of the controlled (24) and uncontrolled (23) SDEs respectively. Substituting all these in, we find that the HJBE is

$$0 = \partial_t V_t(\boldsymbol{x}) + \mathcal{A}_t[V_t](\boldsymbol{x}) + \min_{\boldsymbol{u}}\left[\langle \boldsymbol{u}_t, \nabla_x V_t\rangle + \frac{1}{2}\|\boldsymbol{u}_t\|^2\right]. \tag{29}$$

It follows from the first order condition on the "inner" problem that $\boldsymbol{u}_t = -\nabla_x V_t$, so

$$0 = \partial_t V_t(\boldsymbol{x}) + \mathcal{A}_t[V_t](\boldsymbol{x}) - \frac{1}{2}\|\nabla_x V_t(\boldsymbol{x})\|^2. \tag{30}$$

Use an ansatz that the value function is quadratic:

$$V_t(\boldsymbol{x}) = \frac{1}{2}\boldsymbol{x}^\top \boldsymbol{M}_t\boldsymbol{x} + \boldsymbol{c}_t^\top \boldsymbol{x} + d_t \implies \nabla_x V_t(\boldsymbol{x}) = \boldsymbol{M}_t\boldsymbol{x} + \boldsymbol{c}_t. \tag{31}$$

Then:

$$\mathcal{A}_t[V_t] = \boldsymbol{x}^\top (\boldsymbol{A}_t^\top \boldsymbol{M}_t)\boldsymbol{x} + \langle \boldsymbol{A}_t^\top \boldsymbol{c}_t + \boldsymbol{M}_t\boldsymbol{b}_t, \boldsymbol{x}\rangle + \langle \boldsymbol{b}_t, \boldsymbol{c}_t\rangle + \frac{1}{2}\mathrm{tr}(\boldsymbol{\sigma}_t\boldsymbol{\sigma}_t^\top \boldsymbol{M}_t). \tag{32}$$

Now note that the quadratic form only depends on the *symmetric* part of the matrix, i.e.

$$\boldsymbol{x}^\top \boldsymbol{A}_t^\top \boldsymbol{M}_t\boldsymbol{x} = \frac{1}{2}\boldsymbol{x}^\top \left(\boldsymbol{A}_t^\top \boldsymbol{M}_t + \boldsymbol{M}_t\boldsymbol{A}_t\right)\boldsymbol{x}.$$

The HJBE is therefore

$$0 = \frac{1}{2}\boldsymbol{x}^\top \dot{\boldsymbol{M}}_t\boldsymbol{x} + \dot{\boldsymbol{c}}_t^\top \boldsymbol{x} + \dot{d}_t$$
$$+ \frac{1}{2}\boldsymbol{x}^\top (\boldsymbol{A}_t^\top \boldsymbol{M}_t + \boldsymbol{M}_t\boldsymbol{A}_t)\boldsymbol{x} + (\boldsymbol{A}_t^\top \boldsymbol{c}_t + \boldsymbol{M}_t^\top \boldsymbol{b}_t)^\top \boldsymbol{x} + \boldsymbol{b}_t^\top \boldsymbol{c}_t + \frac{1}{2}\mathrm{tr}(\boldsymbol{\sigma}_t\boldsymbol{\sigma}_t^\top \boldsymbol{M}_t) \tag{33}$$
$$- \frac{1}{2}(\boldsymbol{M}_t\boldsymbol{x} + \boldsymbol{c}_t)^\top (\boldsymbol{M}_t\boldsymbol{x} + \boldsymbol{c}_t)$$

So

$$0 = \dot{M}_t + A_t^\top M_t + M_t A_t - M_t^2, \tag{34}$$

$$0 = \dot{c}_t + A_t^\top c_t + M_t b_t - M_t c_t. \tag{35}$$

with the boundary condition $M_T = \lambda \mathbf{I}$ and $c_t = -\lambda x_T$.

Let $\Lambda_t = M_t^{-1}$, so that $\dot{\Lambda}_t = -M_t^{-1} \dot{M}_t M_t^{-1} = -\Lambda_t \dot{M}_t \Lambda_t$. Substituting, we find that

$$\dot{\Lambda}_t = \Lambda_t A_t^\top + A_t \Lambda_t - \mathbf{I}. \tag{36}$$

Actually we can rewrite the value function in a different way, which is more convenient for us:

$$V_t(x) = \frac{1}{2}(x - k_t)^\top M_t(x - k_t) + const$$

in which case $c_t = -M_t k_t$, and so we have $k_T = x_T$. The corresponding ODE for $k_t$ is

$$\dot{k}_t = A_t k_t + b_t. \tag{37}$$

Let us now specialise to the case of time-invariant coefficients, which allows us to write down explicit expressions for the solutions:

$$\dot{k}_t = A k_t + b \implies k_t = e^{-(T-t)A}(x_T + A^{-1}b) - A^{-1}b. \tag{38}$$

Rewriting the SDE drift to have the form $Ax + b = A(x - m)$ we have that $b = -Am \implies A^{-1}b = -m$, when $A$ is nonsingular. Then:

$$k_t = e^{-(T-t)A}(x_T - m) + m. \tag{39}$$

Now we deal with the quadratic term $\Lambda_t$. Let $G_\tau = \Lambda_{T-\tau}$, so that $\partial_\tau G_\tau = -\dot{\Lambda}_{T-\tau}$. Then $G_\tau$ satisfies

$$\partial_\tau G_\tau = -G_\tau A^\top - A G_\tau + \mathbf{I}, \quad G_0 = 0. \tag{40}$$

So

$$G_\tau = \int_0^\tau e^{-(\tau-s)A} e^{-(\tau-s)A^\top} \mathrm{d}s. \tag{41}$$

Substituting back, we find that

$$\Lambda_t = \int_0^{T-t} e^{-(T-t-s)A} e^{-(T-t-s)A^\top} \mathrm{d}s = \int_0^{T-t} e^{-sA} e^{-sA^\top} \mathrm{d}s. \tag{42}$$

The bridge control is therefore

$$u_{t|(x_0, x_T)} = -\Lambda_t^{-1}(x - k_t) \tag{43}$$

For the general case of non-constant coefficients, we express them as solutions of ODEs with terminal boundary conditions.

$$\dot{k}_t = A_t k_t + b_t, \quad k_T = x_T, \tag{44}$$

$$\dot{\Lambda}_t = \Lambda_t A_t^\top + A_t \Lambda_t - \mathbf{I}, \quad \Lambda_T = 0. \tag{45}$$

In practice, both of these equations can be approximately solved by numerical integration in time.

Finally we remark here that the diffusion component does not play a role in the control, which is classical.

**"Static" Ornstein-Uhlenbeck bridge statistics (Theorem 2)** As was studied by [8] for a more general scenario of time-varying processes, the mvOU process and its conditioned versions are Gaussian processes. Under these assumptions, $(X_s, X_t)$ has joint mean

$$\mu_s \oplus \mu_t, \tag{46}$$

and covariance

$$\begin{bmatrix} \Phi_s & \Phi_s e^{(t-s)A^\top} \\ e^{(t-s)A} \Phi_s & \Phi_t \end{bmatrix}. \tag{47}$$

where $\boldsymbol{\Phi}_t$ is the covariance of the process started from a point mass at time $t$:

$$\boldsymbol{\Phi}_t = \int_0^t e^{(t-s)\boldsymbol{A}}\boldsymbol{\sigma}\boldsymbol{\sigma}^\top e^{(t-s)\boldsymbol{A}^\top}\mathrm{d}s, \quad \boldsymbol{\Phi}_t = \boldsymbol{\Phi}_{t-s} + e^{(t-s)\boldsymbol{A}}\boldsymbol{\Phi}_s e^{(t-s)\boldsymbol{A}^\top}. \tag{48}$$

By taking the Schur complement, we have that $\boldsymbol{X}_s|\boldsymbol{X}_t$ is distributed with variance

$$\mathbb{V}[\boldsymbol{X}_s|\boldsymbol{X}_t] = \boldsymbol{\Phi}_s - \boldsymbol{\Phi}_s e^{(t-s)\boldsymbol{A}^\top}\boldsymbol{\Phi}_t^{-1}e^{(t-s)\boldsymbol{A}}\boldsymbol{\Phi}_s \tag{49}$$

and mean

$$\mathbb{E}[\boldsymbol{X}_s|\boldsymbol{X}_t] = \boldsymbol{\mu}_s + \boldsymbol{\Phi}_s e^{(t-s)\boldsymbol{A}^\top}\boldsymbol{\Phi}_t^{-1}(\boldsymbol{X}_t - \boldsymbol{\mu}_t). \tag{50}$$

More generally, we can consider the three-point correlation $(\boldsymbol{X}_r, \boldsymbol{X}_s, \boldsymbol{X}_t)$. Then the mean is $\boldsymbol{\mu}_r \oplus \boldsymbol{\mu}_s \oplus \boldsymbol{\mu}_t$ and the covariance is

$$\begin{bmatrix} \boldsymbol{\Phi}_r & \boldsymbol{\Phi}_r e^{(s-r)\boldsymbol{A}^\top} & \boldsymbol{\Phi}_r e^{(t-r)\boldsymbol{A}^\top} \\ \cdot & \boldsymbol{\Phi}_s & \boldsymbol{\Phi}_s e^{(t-s)\boldsymbol{A}^\top} \\ \cdot & \cdot & \boldsymbol{\Phi}_t \end{bmatrix} \tag{51}$$

Using the Schur complement again, we have that $(\boldsymbol{X}_r, \boldsymbol{X}_s)|\boldsymbol{X}_t$ has covariance

$$\mathbb{V}[\boldsymbol{X}_r, \boldsymbol{X}_s|\boldsymbol{X}_t] = \boldsymbol{\Phi}_r e^{(s-r)\boldsymbol{A}^\top} - \boldsymbol{\Phi}_r e^{(t-r)\boldsymbol{A}^\top}\boldsymbol{\Phi}_t^{-1}e^{(t-s)\boldsymbol{A}}\boldsymbol{\Phi}_s. \tag{52}$$

For the case of time-varying coefficients, we need to introduce the state transition matrix $\boldsymbol{\Psi}_{ts}$ [29] which describes the deterministic aspects of evolution between $s < t$ under $(\boldsymbol{A}_t)_t$. That is, for a dynamics $\dot{\boldsymbol{x}}_t = \boldsymbol{A}_t\boldsymbol{x}_t$ one has $\boldsymbol{x}_t = \boldsymbol{\Psi}_{t,s}\boldsymbol{x}_s$ and $\boldsymbol{\Psi}(t,s) = \boldsymbol{\Psi}(t,r)\boldsymbol{\Psi}(r,s)$ for $t \geqslant r \geqslant s$. In this case there is not an explicit expression for $\boldsymbol{\Phi}_t$. Instead, let $(\boldsymbol{\Phi}_t)_t$ be the unconditional variance evolutions for such a process started at $t = 0$, obtained by solving

$$\dot{\boldsymbol{\Phi}}_t = \boldsymbol{A}_t\boldsymbol{\Phi}_t + \boldsymbol{\Phi}_t\boldsymbol{A}_t^\top + \boldsymbol{\sigma}_t\boldsymbol{\sigma}_t^\top, \quad \boldsymbol{\Phi}_0 = \boldsymbol{0}. \tag{53}$$

Then the more general result from [8] for the covariance of $(\boldsymbol{X}_r, \boldsymbol{X}_s, \boldsymbol{X}_r)$ is

$$\begin{bmatrix} \boldsymbol{\Phi}_r & \boldsymbol{\Phi}_r\boldsymbol{\Psi}_{sr}^\top & \boldsymbol{\Phi}_r\boldsymbol{\Psi}_{tr}^\top \\ \boldsymbol{\Psi}_{sr}\boldsymbol{\Phi}_r & \boldsymbol{\Phi}_s & \boldsymbol{\Phi}_s\boldsymbol{\Psi}_{ts}^\top \\ \boldsymbol{\Psi}_{tr}\boldsymbol{\Phi}_r & \boldsymbol{\Psi}_{ts}\boldsymbol{\Phi}_s & \boldsymbol{\Phi}_t. \end{bmatrix} \tag{54}$$

From this result, the same Schur complement computation yields expressions for the conditional covariances and mean:

$$\mathbb{V}[\boldsymbol{X}_r, \boldsymbol{X}_s|\boldsymbol{X}_t] = \boldsymbol{\Phi}_r\boldsymbol{\Psi}_{sr}^\top - \boldsymbol{\Phi}_r\boldsymbol{\Psi}_{tr}^\top\boldsymbol{\Phi}_t^{-1}\boldsymbol{\Psi}_{ts}\boldsymbol{\Phi}_s, \tag{55}$$

$$\mathbb{V}[\boldsymbol{X}_s|\boldsymbol{X}_t] = \boldsymbol{\Phi}_s - \boldsymbol{\Phi}_s\boldsymbol{\Psi}_{ts}^\top\boldsymbol{\Phi}_t^{-1}\boldsymbol{\Psi}_{ts}\boldsymbol{\Phi}_s, \tag{56}$$

$$\mathbb{E}[\boldsymbol{X}_s|\boldsymbol{X}_t] = \boldsymbol{\mu}_s + \boldsymbol{\Phi}_s\boldsymbol{\Psi}_{ts}^\top\boldsymbol{\Phi}_t^{-1}(\boldsymbol{X}_t - \boldsymbol{\mu}_t) \tag{57}$$

**Formal statement and proof of Lemma 1**

**Lemma** (Formal statement of Lemma 1). *Let* $\mathrm{d}\boldsymbol{X}_t = \boldsymbol{A}\boldsymbol{X}_t\mathrm{d}t + \boldsymbol{\sigma}\mathrm{d}\boldsymbol{B}_t$ *be a linear SDE with generic drift* $\boldsymbol{A} \in \mathbb{R}^{d\times d}$ *and diffusion* $\boldsymbol{\sigma} \in \mathbb{R}^{d\times d}$. *There exists an orthogonal matrix* $\boldsymbol{U} \in \mathbb{R}^{d\times d}$ *such that* $\boldsymbol{Y}_t = \boldsymbol{U}^\top\boldsymbol{X}_t$ *obeys a linear SDE with transformed drift matrix* $\boldsymbol{U}^\top\boldsymbol{A}\boldsymbol{U}$ *and a diagonal diffusion matrix* $\boldsymbol{\Lambda} = \mathrm{diag}(\boldsymbol{\lambda})$, *where* $\boldsymbol{\lambda} = (\lambda_1, \ldots, \lambda_d) \geqslant 0$.

*Proof.* Let $\boldsymbol{D} = \frac{1}{2}\boldsymbol{\sigma}\boldsymbol{\sigma}^\top$. This is positive semidefinite and therefore can its spectral decomposition can be written as $\boldsymbol{D} = \frac{1}{2}\boldsymbol{U}\boldsymbol{\Lambda}^2\boldsymbol{U}^\top$ where $\boldsymbol{U}$ is orthogonal and $\boldsymbol{\Lambda} = \mathrm{diag}(\boldsymbol{\lambda}) \geq 0$. Consider the SDEs

$$\mathrm{d}\boldsymbol{X}_t = \boldsymbol{A}\boldsymbol{X}_t\mathrm{d}t + \boldsymbol{\sigma}\mathrm{d}\boldsymbol{B}_t \tag{a.}$$

and

$$\mathrm{d}\boldsymbol{X}_t = \boldsymbol{A}\boldsymbol{X}_t\mathrm{d}t + \boldsymbol{U}\boldsymbol{\Lambda}\boldsymbol{U}^\top\mathrm{d}\boldsymbol{B}_t. \tag{b.}$$

Let $f(\boldsymbol{x})$ be a twice differentiable test function. SDE (a.) has generator $(\mathcal{L}f)(\boldsymbol{x}) = \langle\partial_x f(\boldsymbol{x}), \boldsymbol{A}\boldsymbol{x}\rangle + \frac{1}{2}\mathrm{tr}(\boldsymbol{\sigma}^\top\partial_{xx}f(\boldsymbol{x})\boldsymbol{\sigma})$. Computing the generator for SDE (b.), $(\tilde{\mathcal{L}}f)(\boldsymbol{x}) = \langle\partial_x f(\boldsymbol{x}), \boldsymbol{A}\boldsymbol{x}\rangle + \frac{1}{2}\mathrm{tr}(\boldsymbol{U}\boldsymbol{\Lambda}\boldsymbol{U}^\top\partial_{xx}f(\boldsymbol{x})\boldsymbol{U}\boldsymbol{\Lambda}\boldsymbol{U}^\top)$. Note that $\mathrm{tr}(\boldsymbol{U}\boldsymbol{\Lambda}\boldsymbol{U}^\top\partial_{xx}f(\boldsymbol{x})\boldsymbol{U}\boldsymbol{\Lambda}\boldsymbol{U}^\top) = \mathrm{tr}(\partial_{xx}f(\boldsymbol{x})\boldsymbol{U}\boldsymbol{\Lambda}^2\boldsymbol{U}^\top) = \mathrm{tr}(\boldsymbol{\sigma}^\top\partial_{xx}f(\boldsymbol{x})\boldsymbol{\sigma})$. Hence both SDEs share the same generator, and so are equal in law.

Since orthogonal transformations of Brownian increments are also Brownian increments, we have that $\boldsymbol{U}^\top\mathrm{d}\boldsymbol{B}_t = \mathrm{d}\tilde{\boldsymbol{B}}_t$. Substituting into (b.), we have $\mathrm{d}\boldsymbol{X}_t = \boldsymbol{A}\boldsymbol{X}_t\mathrm{d}t + \boldsymbol{U}\boldsymbol{\Lambda}\mathrm{d}\tilde{\boldsymbol{B}}_t$. Then,

$$\mathrm{d}(\boldsymbol{U}^\top\boldsymbol{X}_t) = \boldsymbol{U}^\top\boldsymbol{A}\boldsymbol{X}_t\mathrm{d}t + \boldsymbol{\Lambda}\mathrm{d}\tilde{\boldsymbol{B}}_t \implies \mathrm{d}\boldsymbol{Y}_t = \boldsymbol{U}^\top\boldsymbol{A}\boldsymbol{U}\boldsymbol{Y}_t\mathrm{d}t + \boldsymbol{\Lambda}\mathrm{d}\tilde{\boldsymbol{B}}_t.$$

$\square$

## A.3 Simulation-free Schrödinger bridges with linear reference dynamics

Having characterised the solution of the SBP for general reference processes and now that we have derived the score and flow for the Ornstein-Uhlenbeck bridge, the path is clear towards a simulation-free scheme for learning Schrödinger bridges where the reference dynamics are given by a *linear* SDE.

For solution of the static SBP problem (18) (equivalently, (SBP-static)), the transition kernel of the reference dynamics is given by

$$\frac{\mathrm{d}\boldsymbol{Q}_{0t}}{\mathrm{d}\boldsymbol{x} \otimes \mathrm{d}\boldsymbol{x}}(\boldsymbol{x}_0, \boldsymbol{x}_t) = \frac{1}{Z_t} \exp\left(-\frac{1}{2}(\boldsymbol{x}_t - \boldsymbol{\mu}_t^{\boldsymbol{x}_0})^\top \boldsymbol{\Sigma}_t^{-1}(\boldsymbol{x}_t - \boldsymbol{\mu}_t^{\boldsymbol{x}_0})\right) \frac{\mathrm{d}\boldsymbol{Q}_0}{\mathrm{d}\boldsymbol{x}}(\boldsymbol{x}_0)$$

$$\implies -\log\left(\frac{\mathrm{d}\boldsymbol{Q}_{0t}}{\mathrm{d}\boldsymbol{x} \otimes \mathrm{d}\boldsymbol{x}}\right) = \frac{1}{2}(\boldsymbol{x}_t - \boldsymbol{\mu}_t^{\boldsymbol{x}_0})^\top \boldsymbol{\Sigma}_t^{-1}(\boldsymbol{x}_t - \boldsymbol{\mu}_t^{\boldsymbol{x}_0}) + \log Z_t - \log\left(\frac{\mathrm{d}\boldsymbol{Q}_0}{\mathrm{d}\boldsymbol{x}}(\boldsymbol{x}_0)\right), \quad (58)$$

where $\boldsymbol{\mu}_t^{\boldsymbol{x}_0}$ denotes the mean at time $t$ conditional on $(0, \boldsymbol{x}_0)$. Notably, the last two terms depend only on $\boldsymbol{x}_0$. It is a classical result (and very easy to show) that these kinds of terms do not affect the minimiser of (18) and so they are immaterial. The cost function to use is thus effectively

$$C(\boldsymbol{x}_0, \boldsymbol{x}_t) = \frac{1}{2}(\boldsymbol{x}_t - \boldsymbol{\mu}_t^{\boldsymbol{x}_0})^\top \boldsymbol{\Sigma}_t^{-1}(\boldsymbol{x}_t - \boldsymbol{\mu}_t^{\boldsymbol{x}_0}). \tag{59}$$

Once a solution $\boldsymbol{\pi}$ of the static SBP is on hand, we want to utilise the stochastic regression objective on the conditional score and flow. Sampling $(\boldsymbol{x}_0, \boldsymbol{x}_t)$ from $\boldsymbol{\pi}$, recall that $\boldsymbol{P}^{(x_0, x_1)} = \boldsymbol{Q}^{(x_0, x_1)}$ so we want to sample from the $\boldsymbol{Q}$-bridge using (11), (12):

$$t \sim U[0, 1], \quad \boldsymbol{x}_t^{(x_0, x_1)} \sim \mathcal{N}(\boldsymbol{\mu}_{t|(\boldsymbol{x}_0, \boldsymbol{x}_1)}, \boldsymbol{\Sigma}_{t|(\boldsymbol{x}_0, \boldsymbol{x}_1)}). \tag{60}$$

Equations (9) and (12) give us the score and flow at $\boldsymbol{x}_t$.

## A.4 Characterisation of the $\mathbb{Q}$-GSB (Theorem 3)

We will construct the $\mathbb{Q}$-GSB utilising its characterisation (2). Our approach is similar to that of [5] – we first obtain explicit formulae for the static $\mathbb{Q}$-GSB (SBP-static) and then build towards the dynamical $\mathbb{Q}$-GSB (SBP-dyn) by using the characterisation of $\mathbb{Q}$-bridges.

**Static $\mathbb{Q}$-GSB** The standard Gaussian EOT problem has a well known solution [32, 21, 5, 22]. Here we use the notations of [21] and define

$$\mathrm{OT}_{\sigma^2}^\otimes(\alpha, \beta) := \min_{\pi \in \Pi(\alpha, \beta)} \frac{1}{2} \int \|\boldsymbol{x} - \boldsymbol{y}\|_2^2 \, \mathrm{d}\pi(\boldsymbol{x}, \boldsymbol{y}) + \sigma^2 \, \mathrm{H}(\pi | \alpha \otimes \beta), \tag{61}$$

where $\sigma > 0$ is the regularisation level and $\alpha = \mathcal{N}(\boldsymbol{a}, \boldsymbol{A})$ and $\beta = \mathcal{N}(\boldsymbol{b}, \boldsymbol{B})$. The solution to (61) in the Gaussian case is given by [21, Theorem 1]

$$\mathrm{OT}_{\sigma^2}^\otimes(\alpha, \beta) = \frac{1}{2}\|\boldsymbol{a} - \boldsymbol{b}\|_2^2 + \frac{1}{2}\mathsf{B}_{\sigma^2}^\otimes(\boldsymbol{A}, \boldsymbol{B}) \tag{62}$$

$$\mathsf{B}_{\sigma^2}^\otimes(\boldsymbol{A}, \boldsymbol{B}) = \mathrm{tr}(\boldsymbol{A}) + \mathrm{tr}(\boldsymbol{B}) - 2\mathrm{tr}(\boldsymbol{C}) + \sigma^2 \log \det\left(\frac{1}{\sigma^2}\boldsymbol{C} + \boldsymbol{I}\right), \tag{63}$$

$$\boldsymbol{C} = \boldsymbol{A}^{1/2}\left(\boldsymbol{A}^{1/2}\boldsymbol{B}\boldsymbol{A}^{1/2} + \frac{\sigma^4}{4}\boldsymbol{I}\right)^{1/2}\boldsymbol{A}^{-1/2} - \frac{\sigma^2}{2}\boldsymbol{I}. \tag{64}$$

We are, of course, interested in a slightly different problem, namely, for $\rho_0 = \mathcal{N}(\boldsymbol{a}, \boldsymbol{\mathcal{A}})$ and $\rho_T = \mathcal{N}(\boldsymbol{b}, \boldsymbol{\mathcal{B}})$, we seek to solve $\min_{\pi \in \Pi(\rho_0, \rho_T)} \mathrm{KL}(\pi | \mathbb{Q}_{0,T})$. Expanding the definition of KL and

simplifying, we get

$$\text{(SBP-static)} = \min_{\pi \in \Pi(\rho_0, \rho_T)} \int \mathrm{d}\pi \log \left( \frac{\mathrm{d}\pi}{\mathrm{d}\mathbb{Q}_{0,T}} \right) \tag{65}$$

$$= \min_{\pi \in \Pi(\rho_0, \rho_T)} - \int \mathrm{d}\pi \log \left( \frac{\mathrm{d}\mathbb{Q}_{0,T}}{\mathrm{d}x_0 \otimes \mathrm{d}x_T} \right) + \mathrm{H}(\pi | \rho_0 \otimes \rho_T) + \mathrm{H}(\rho_0 | \mathrm{d}x_0) + \mathrm{H}(\rho_T | \mathrm{d}x_T) \tag{66}$$

$$\simeq \min_{\pi \in \Pi(\rho_0, \rho_T)} - \int \mathrm{d}\pi \log \left( \frac{\mathrm{d}\mathbb{Q}_{0,T}}{\mathrm{d}x_0 \otimes \mathrm{d}x_T} \right) + \mathrm{H}(\pi | \rho_0 \otimes \rho_T) \tag{67}$$

$$\simeq \min_{\pi \in \Pi(\rho_0, \rho_T)} - \int \mathrm{d}\pi(x_0, x_T) \log \left( \frac{\mathrm{d}\mathbb{Q}_{T|0}(x_T | x_0)}{\mathrm{d}x_T} \right) + \mathrm{H}(\pi | \rho_0 \otimes \rho_T) \tag{68}$$

where by $\simeq$ we denote equality of the objective up to additive constants which do not affect the minimiser $\pi$. We note that the last line is exactly (61) with $\alpha, \beta = \rho_0, \rho_T$ and $\sigma = 1$ and a modified cost. Further, recognise $\mathrm{d}\mathbb{Q}_{T|0}(x_T | x_0)/\mathrm{d}x_T$ as the $\mathbb{Q}$-transition density, and $\mathbb{Q}_{T|0} = \mathcal{N}(\boldsymbol{\mu}_T^{x_0}, \boldsymbol{\Sigma}_T)$. Thus,

$$\text{(SBP-static)} \simeq \min_{\pi \in \Pi(\rho_0, \rho_T)} \frac{1}{2} \int (\boldsymbol{x}_T - \boldsymbol{\mu}_T^{x_0})^\top \boldsymbol{\Sigma}_T^{-1} (\boldsymbol{x}_T - \boldsymbol{\mu}_T^{x_0}) \, \mathrm{d}\pi(\boldsymbol{x}_0, \boldsymbol{x}_T) + \mathrm{H}(\pi | \rho_0 \otimes \rho_T) \tag{69}$$

where we remind that $\boldsymbol{\mu}_t^{x_0}$ is the mean at time $t$ conditional on starting at $(0, \boldsymbol{x}_0)$, and $\boldsymbol{\Sigma}_t$ is the covariance at time $t$, started at a point mass, i.e. $\boldsymbol{\Sigma}_0 = 0$.

In particular, we have $\boldsymbol{\Sigma}_t = \boldsymbol{\Phi}_t$, using the notation of Theorem 2. Recall that in the case of constant coefficients, we have the following explicit expressions:

$$\boldsymbol{\mu}_t^{x_0} = e^{t\boldsymbol{A}}(\boldsymbol{x}_0 - \boldsymbol{m}) + \boldsymbol{m}, \tag{70}$$

$$\boldsymbol{\Phi}_t = \int_0^t e^{(t-s)\boldsymbol{A}} \boldsymbol{\sigma}\boldsymbol{\sigma}^\top e^{(t-s)\boldsymbol{A}^\top} \mathrm{d}s. \tag{71}$$

Define the following changes of variable $(\boldsymbol{x}_0, \boldsymbol{x}_T) \mapsto (\overline{\boldsymbol{x}}_0, \overline{\boldsymbol{x}}_T)$, injective whenever $\boldsymbol{\Sigma}_T$ is positive definite:

$$\overline{\boldsymbol{x}}_0 = T_0(\boldsymbol{x}_0) = \boldsymbol{\Sigma}_T^{-1/2} \boldsymbol{\mu}_T^{x_0},$$

$$\overline{\boldsymbol{x}}_T = T_T(\boldsymbol{x}_T) = \boldsymbol{\Sigma}_T^{-1/2} \boldsymbol{x}_T$$

Let $\overline{\rho}_0 = (T_0)_\# \rho_0$ and $\overline{\rho}_T = (T_T)_\# \rho_T$, and $\overline{\pi} = (T_0 \times T_T)_\# \pi$ and note that the relative entropy is invariant to this change of coordinates. So the problem (69) is equivalent to

$$\min_{\overline{\pi} \in \Pi(\overline{\rho}_0, \overline{\rho}_T)} \frac{1}{2} \int \|\overline{\boldsymbol{x}}_T - \overline{\boldsymbol{x}}_0\|_2^2 \, \mathrm{d}\overline{\pi}(\overline{\boldsymbol{x}}_0, \overline{\boldsymbol{x}}_T) + \mathrm{H}(\overline{\pi} | \overline{\rho}_0 \otimes \overline{\rho}_T). \tag{72}$$

This is exactly $\mathrm{OT}_{\sigma^2=1}^\otimes(\overline{\rho}_0, \overline{\rho}_T)$ as per (61) with $\overline{\rho}_0 = \mathcal{N}(\overline{\boldsymbol{a}}, \overline{\boldsymbol{\mathcal{A}}})$ and $\overline{\rho}_T = \mathcal{N}(\overline{\boldsymbol{b}}, \overline{\boldsymbol{\mathcal{B}}})$. The transformed means and covariances are

$$\overline{\boldsymbol{a}} = \boldsymbol{\Sigma}_T^{-1/2}(e^{T\boldsymbol{A}}(\boldsymbol{a} - \boldsymbol{m}) + \boldsymbol{m})$$

$$\overline{\boldsymbol{\mathcal{A}}} = \boldsymbol{\Sigma}_T^{-1/2} e^{T\boldsymbol{A}} \boldsymbol{\mathcal{A}} e^{T\boldsymbol{A}^\top} \boldsymbol{\Sigma}_T^{-1/2}$$

$$\overline{\boldsymbol{b}} = \boldsymbol{\Sigma}_T^{-1/2} \boldsymbol{b}$$

$$\overline{\boldsymbol{\mathcal{B}}} = \boldsymbol{\Sigma}_T^{-1/2} \boldsymbol{\mathcal{B}} \boldsymbol{\Sigma}_T^{-1/2}$$

We remark that for time-dependent coefficients our approach still applies, however computation of the transformation coefficients will be in terms of the general state transition matrix instead of matrix exponentials. In what follows, we focus on results for constant coefficients for simplicity and their practical relevance.

The solution to (72) is therefore

$$(\overline{\boldsymbol{x}}_0, \overline{\boldsymbol{x}}_T) \sim \overline{\pi} = \mathcal{N} \left( \begin{bmatrix} \overline{\boldsymbol{a}} \\ \overline{\boldsymbol{b}} \end{bmatrix}, \begin{bmatrix} \overline{\boldsymbol{\mathcal{A}}} & \overline{\boldsymbol{\mathcal{C}}} \\ \overline{\boldsymbol{\mathcal{C}}}^\top & \overline{\boldsymbol{\mathcal{B}}} \end{bmatrix} \right),$$

where $\overline{\boldsymbol{\mathcal{C}}}$ is the cross-covariance the transport plan, given by (64).

**Dynamic $\mathbb{Q}$-GSB : marginals and covariance**   Let $X_t|(x_0, x_T)$ denote the $\mathbb{Q}$-bridge pinned at $(x_0, x_T)$. Then by Theorem 2,

$$X_t|(x_0, x_T) = \mathcal{N}(\mu_{t|(x_0,x_T)}, \Sigma_{t|(x_0,x_T)}) = \mathcal{N}(\mu_{t|(x_0,x_T)}, \Omega_t).$$

Invoking the inverse mappings $T_0^{-1}, T_T^{-1}$ and substituting the expression for $\mu_{t|(x_0,x_T)}$ from 2, we get

$$\mu_{t|(T_0^{-1}(\overline{x}_0), T_T^{-1}(\overline{x}_T))} = \underbrace{\left( e^{-(T-t)A} - \Gamma_t \right) \Sigma_T^{1/2}}_{\mathfrak{A}_t} \overline{x}_0 + \underbrace{\Gamma_t \Sigma_T^{1/2}}_{\mathfrak{B}_t} \overline{x}_T + \underbrace{\left( I - e^{-(T-t)A} \right) m}_{\mathfrak{c}_t} \quad (73)$$

$$= \mathfrak{A}_t \overline{x}_0 + \mathfrak{B}_t \overline{x}_T + \mathfrak{c}_t. \quad (74)$$

where $\Gamma_t = \Phi_t e^{(T-t)A^\top} \Phi_T^{-1}$. For clarity, we state first a general formula:

**Lemma 2.** *Let* $(X_0, X_1) \sim \mathcal{N}\left( \begin{bmatrix} \mu_0 \\ \mu_1 \end{bmatrix}, \begin{bmatrix} \Sigma_{00} & \Sigma_{01} \\ \Sigma_{10} & \Sigma_{11} \end{bmatrix} \right)$. *Let*

$$Y_{X_0, X_1} \sim \mathcal{N}(AX_0 + BX_1 + c, \Omega).$$

*Then,* $\mathbb{E}[Y] = A\mu_0 + B\mu_1 + c$ *and* $\mathbb{V}[Y] = \Omega + A\Sigma_{00}A^\top + B\Sigma_{11}B^\top + A\Sigma_{01}B^\top + B\Sigma_{01}^\top A^\top$.

Application to (74) gives us the following formula for the variance of the bridge at time $t$:

$$\mathbb{V}[X_t] = \Omega_t + \mathfrak{A}_t \overline{A} \mathfrak{A}_t^\top + \mathfrak{B}_t \overline{B} \mathfrak{B}_t^\top + \mathfrak{A}_t \overline{C} \mathfrak{B}_t^\top + \mathfrak{B}_t \overline{C}^\top \mathfrak{A}_t^\top =: \Xi_t. \quad (75)$$

Substituting and simplifying, we get the following expressions for each of the terms:

$$\mathfrak{A}_t \overline{A} \mathfrak{A}_t^\top = (I - \Gamma_t e^{(T-t)A}) e^{tA} \mathcal{A} e^{tA^\top} (I - \Gamma_t e^{(T-t)A})^\top, \quad (76)$$

$$\mathfrak{B}_t \overline{B} \mathfrak{B}_t^\top = \Gamma_t \mathcal{B} \Gamma_t^\top, \quad (77)$$

$$\mathfrak{A}_t \overline{C} \mathfrak{B}_t^\top = (e^{-(T-t)A} - \Gamma_t) \Sigma_T^{1/2} \overline{C} \Sigma_T^{1/2} \Gamma_t^\top. \quad (78)$$

Similarly, the mean can be computed as

$$\mathbb{E}[X_t] = \mathfrak{A}_t \overline{a} + \mathfrak{B}_t \overline{b} + \mathfrak{c}_t \quad (79)$$

$$= \left( e^{-(T-t)A} - \Gamma_t \right) e^{TA}(a - m) + \Gamma_t(b - m) + m =: \nu_t. \quad (80)$$

To calculate the covariance of the SB process, we use again the disintegration property (2). We have from (52) for the $\mathbb{Q}$-bridges and $0 < s < t < T$:

$$\mathbb{V}[X_s, X_t|X_0, X_T] = \Phi_s e^{(t-s)A^\top} - \Phi_s e^{(T-s)A^\top} \Phi_T^{-1} e^{(T-t)A} \Phi_t =: \Omega_{s,t}. \quad (81)$$

Now,

$$\mathbb{V}[X_s, X_t] = \mathbb{E}[(X_s - \nu_s)(X_t - \nu_t)^\top] \quad (82)$$

$$= \mathbb{E}[X_s X_t^\top] - \nu_s \nu_t^\top \quad (83)$$

$$= \mathbb{E}[\mathbb{E}[X_s X_t^\top | X_0, X_T]] - \nu_s \nu_t^\top. \quad (84)$$

and

$$\mathbb{E}[X_s X_t^\top | X_0, X_T] = \mathbb{V}[X_s, X_t | X_0, X_T] + \mu_{s|(X_0, X_T)} \mu_{t|(X_0, X_T)}^\top, \quad (85)$$

in which the first (variance) term doesn't actually depend on $(X_0, X_T)$. So,

$$\mathbb{E}[X_s X_t^\top] = \Omega_{st} + \mathbb{E}_{(X_0, X_T)}[\mu_{s|(X_0, X_T)} \mu_{t|(X_0, X_T)}^\top]. \quad (86)$$

In fact let's switch to the "mapped" coordinates $\overline{X}_0, \overline{X}_T$ for the endpoints. We abuse notation and omit the inverse map in what follows. Then, from previously,

$$\mu_{t|(\overline{X}_0, \overline{X}_T)} = \mathfrak{A}_t \overline{X}_0 + \mathfrak{B}_t \overline{X}_T + \mathfrak{c}_t. \quad (87)$$

Expanding, collecting and cancelling terms, we have that

$$\mathbb{E}[\mu_{s|(\overline{X}_0, \overline{X}_T)} \mu_{t|(\overline{X}_0, \overline{X}_T)}^\top] - \nu_s \nu_t^\top \quad (88)$$

$$= \mathfrak{A}_s \mathbb{V}[\overline{X}_0] \mathfrak{A}_t^\top + \mathfrak{A}_s \mathbb{V}[\overline{X}_0, \overline{X}_T] \mathfrak{B}_t^\top + \mathfrak{B}_s \mathbb{V}[\overline{X}_T, \overline{X}_0] \mathfrak{A}_t^\top + \mathfrak{B}_s \mathbb{V}[\overline{X}_T] \mathfrak{B}_t^\top \quad (89)$$

$$= \mathfrak{A}_s \overline{A} \mathfrak{A}_t^\top + \mathfrak{A}_s \overline{C} \mathfrak{B}_t^\top + \mathfrak{B}_s \overline{C}^\top \mathfrak{A}_t^\top + \mathfrak{B}_s \overline{B} \mathfrak{B}_t^\top. \quad (90)$$

Putting everything together, we get

$$\mathbb{V}[X_s, X_t] = \Omega_{st} + \mathfrak{A}_s \overline{A} \mathfrak{A}_t^\top + \mathfrak{A}_s \overline{C} \mathfrak{B}_t^\top + \mathfrak{B}_s \overline{C}^\top \mathfrak{A}_t^\top + \mathfrak{B}_s \overline{B} \mathfrak{B}_t^\top =: \Xi_{s,t}. \quad (91)$$

**Dynamic $\mathbb{Q}$-GSB : SDE representation**   We proceed via the generator route also used by [5]. Expanding the covariance (91), we find that

$$\mathbb{V}[\boldsymbol{X}_t, \boldsymbol{X}_{t+h}] = \boldsymbol{\Xi}_{t,t+h} = \boldsymbol{\Omega}_{t,t+h} + \mathfrak{A}_t\overline{\boldsymbol{\mathcal{A}}}\mathfrak{A}_{t+h}^\top + \mathfrak{A}_t\overline{\boldsymbol{\mathcal{C}}}\mathfrak{B}_{t+h}^\top + \mathfrak{B}_t\overline{\boldsymbol{\mathcal{C}}}^\top\mathfrak{A}_{t+h}^\top + \mathfrak{B}_t\overline{\boldsymbol{\mathcal{B}}}\mathfrak{B}_{t+h}^\top \quad (92)$$

$$= \boldsymbol{\Xi}_t + h\left\{(\partial_{t'}\boldsymbol{\Omega}_{t,t'})(t) + \mathfrak{A}_t\overline{\boldsymbol{\mathcal{A}}}\dot{\mathfrak{A}}_t^\top + \mathfrak{A}_t\overline{\boldsymbol{\mathcal{C}}}\dot{\mathfrak{B}}_t^\top + \mathfrak{B}_t\overline{\boldsymbol{\mathcal{C}}}^\top\dot{\mathfrak{A}}_t^\top + \mathfrak{B}_t\overline{\boldsymbol{\mathcal{B}}}\dot{\mathfrak{B}}_t^\top\right\} + \ldots \quad (93)$$

$$= \boldsymbol{\Xi}_t + h\boldsymbol{S}_t + \ldots \quad (94)$$

where we have set the key quantity

$$\boldsymbol{S}_t = (\partial_{t'}\boldsymbol{\Omega}_{t,t'})(t) + \mathfrak{A}_t\overline{\boldsymbol{\mathcal{A}}}\dot{\mathfrak{A}}_t^\top + \mathfrak{A}_t\overline{\boldsymbol{\mathcal{C}}}\dot{\mathfrak{B}}_t^\top + \mathfrak{B}_t\overline{\boldsymbol{\mathcal{C}}}^\top\dot{\mathfrak{A}}_t^\top + \mathfrak{B}_t\overline{\boldsymbol{\mathcal{B}}}\dot{\mathfrak{B}}_t^\top \quad (95)$$

Now, $(\boldsymbol{X}_{t+h}, \boldsymbol{X}_t)$ are jointly Gaussian with mean and covariance

$$\begin{bmatrix}\boldsymbol{\nu}_{t+h} \\ \boldsymbol{\nu}_t\end{bmatrix}, \begin{bmatrix}\boldsymbol{\Xi}_{t+h} & \boldsymbol{\Xi}_{t+h,t} \\ \boldsymbol{\Xi}_{t,t+h} & \boldsymbol{\Xi}_t\end{bmatrix}. \quad (96)$$

So

$$\mathbb{E}[\boldsymbol{X}_{t+h}|\boldsymbol{X}_t] = \boldsymbol{\nu}_{t+h} + \boldsymbol{\Xi}_{t,t+h}^\top\boldsymbol{\Xi}_t^{-1}(\boldsymbol{X}_t - \boldsymbol{\nu}_t) \quad (97)$$

$$= \boldsymbol{X}_t + h\left[\dot{\boldsymbol{\nu}}_t + \boldsymbol{S}_t^\top\boldsymbol{\Xi}_t^{-1}(\boldsymbol{X}_t - \boldsymbol{\nu}_t)\right] \quad (98)$$

$$\mathbb{V}[\boldsymbol{X}_{t+h}|\boldsymbol{X}_t] = \boldsymbol{\Xi}_{t+h} - \boldsymbol{\Xi}_{t,t+h}^\top\boldsymbol{\Xi}_t^{-1}\boldsymbol{\Xi}_{t,t+h} \quad (99)$$

$$= h(\dot{\boldsymbol{\Sigma}}_t - (\boldsymbol{S}_t + \boldsymbol{S}_t^\top)). \quad (100)$$

Let $\boldsymbol{x} \mapsto u(\boldsymbol{x})$ be a twice differentiable test function. Then:

$$\mathbb{E}\left[u(\boldsymbol{X}_{t+h})|\boldsymbol{X}_t = \boldsymbol{x}\right] \quad (101)$$

$$= \mathbb{E}\Big[u(\boldsymbol{X}_t) + (\partial_x u(\boldsymbol{X}_t))^\top(\boldsymbol{X}_{t+h} - \boldsymbol{X}_t)+$$

$$\frac{1}{2}(\boldsymbol{X}_{t+h} - \boldsymbol{X}_t)^\top(\partial_{xx}^2 u(\boldsymbol{X}_t))(\boldsymbol{X}_{t+h} - \boldsymbol{X}_t)\Big|\boldsymbol{X}_t = \boldsymbol{x}\Big] \quad (102)$$

$$= u(\boldsymbol{x}) + h(\partial_x u(\boldsymbol{x}))^\top\left(\dot{\boldsymbol{\nu}}_t + \boldsymbol{S}_t^\top\boldsymbol{\Xi}_t^{-1}(\boldsymbol{x} - \boldsymbol{\nu}_t)\right) +$$

$$\frac{1}{2}\mathbb{E}\left[(\boldsymbol{X}_{t+h} - \boldsymbol{X}_t)^\top(\partial_{xx}^2 u(\boldsymbol{X}_t))(\boldsymbol{X}_{t+h} - \boldsymbol{X}_t)\Big|\boldsymbol{X}_t = \boldsymbol{x}\right] \quad (103)$$

$$= u(\boldsymbol{x}) + h(\partial_x u(\boldsymbol{x}))^\top\left(\dot{\boldsymbol{\nu}}_t + \boldsymbol{S}_t^\top\boldsymbol{\Xi}_t^{-1}(\boldsymbol{x} - \boldsymbol{\nu}_t)\right) + \frac{h}{2}\text{tr}[(\partial_{xx}^2 u(\boldsymbol{x}))(\dot{\boldsymbol{\Xi}}_t - (\boldsymbol{S}_t + \boldsymbol{S}_t^\top))] + \ldots \quad (104)$$

Subtracting $u(\boldsymbol{x})$ and taking the limit $h \to 0$, it is clear from the definition of the generator that the SDE drift is

$$\dot{\boldsymbol{\nu}}_t + \boldsymbol{S}_t^\top\boldsymbol{\Xi}_t^{-1}(\boldsymbol{x} - \boldsymbol{\nu}_t). \quad (105)$$

This takes the same form as the equation found in [5], however our formulae for $\boldsymbol{S}_t$ allow us to apply it to any linear reference SDE, not necessarily ones with scalar drift. Additionally, we empirically verify that for asymmetric $\boldsymbol{A}$ the matrix $\boldsymbol{S}_t$ is asymmetric. This contrasts with the symmetric nature of the drift for the gradient-type setting.

Now we want to work out what each of these terms are in practice. Effectively we need to compute $\dot{\mathfrak{A}}_t$, $\dot{\mathfrak{B}}_t$ and $(\partial_{t'}\boldsymbol{\Omega}_{t,t'})(t)$:

$$\dot{\mathfrak{A}}_t = (\boldsymbol{A}e^{-(T-t)\boldsymbol{A}} - \dot{\boldsymbol{\Gamma}}_t)\boldsymbol{\Sigma}_T^{1/2}, \quad (106)$$

$$\dot{\mathfrak{B}}_t = \dot{\boldsymbol{\Gamma}}_t\boldsymbol{\Sigma}_T^{1/2}, \quad (107)$$

$$\dot{\boldsymbol{\Gamma}}_t = \left(e^{t\boldsymbol{A}}\boldsymbol{\sigma}\boldsymbol{\sigma}^\top e^{T\boldsymbol{A}^\top} - \boldsymbol{\Phi}_t\boldsymbol{A}^\top e^{(T-t)\boldsymbol{A}^\top}\right)\boldsymbol{\Phi}_T^{-1}, \quad (108)$$

$$(\partial_{t'}\boldsymbol{\Omega}_{t,t'})(t) = \boldsymbol{\Phi}_t\left[\boldsymbol{A}^\top - e^{(T-t)\boldsymbol{A}^\top}\boldsymbol{\Phi}_T^{-1}e^{(T-t)\boldsymbol{A}}\left(-\boldsymbol{A}\boldsymbol{\Phi}_t + e^{t\boldsymbol{A}}\boldsymbol{\sigma}\boldsymbol{\sigma}^\top e^{t\boldsymbol{A}^\top}\right)\right]. \quad (109)$$

**Remark on the case of time-varying coefficients**  In this case we can still assume without loss of generality that we still work on the time interval $[0, T]$, by shifting the time coordinate if necessary. Let $\boldsymbol{\Psi}(t, s)$ be the state transition matrix associated with $\boldsymbol{A}_t$ and $\boldsymbol{\Phi}_t$ be the solution at time $t$ to (53). Then we still have $\boldsymbol{\Sigma}_T = \boldsymbol{\Phi}_T$ for the transition kernel covariance in the cost, i.e. the covariance started from a point mass. For the mean of the reference process started from $\boldsymbol{x}_0$, we have the generalised formula

$$\boldsymbol{\mu}_t(\boldsymbol{x}_0) = \boldsymbol{\Psi}(t, 0)\boldsymbol{x}_0 - \int_0^t \boldsymbol{\Psi}(t, s)\boldsymbol{A}_s\boldsymbol{m}_s \, \mathrm{d}s, \tag{110}$$

and we remark that its inverse $\boldsymbol{\mu}_t^{-1}$ always exists since $\boldsymbol{\Psi}$ is never singular:

$$\boldsymbol{\mu}_t^{-1}(\boldsymbol{y}) = \boldsymbol{\Psi}(t, 0)^{-1}\left(\boldsymbol{y} + \int_0^t \boldsymbol{\Psi}(t, s)\boldsymbol{A}_s\boldsymbol{m}_s \, \mathrm{d}s\right). \tag{111}$$

Then

$$\overline{\boldsymbol{a}} = \boldsymbol{\Sigma}_T^{-1/2}\boldsymbol{\mu}_T^a, \tag{112}$$

$$\overline{\boldsymbol{A}} = \boldsymbol{\Sigma}_T^{-1/2}\boldsymbol{\Psi}(t, 0)\boldsymbol{\mathcal{A}}\boldsymbol{\Psi}(t, 0)^\top\boldsymbol{\Sigma}_T^{-1/2}, \tag{113}$$

$$\overline{\boldsymbol{b}} = \boldsymbol{\Sigma}_T^{-1/2}\boldsymbol{b}, \tag{114}$$

$$\overline{\boldsymbol{B}} = \boldsymbol{\Sigma}_T^{-1/2}\boldsymbol{\mathcal{B}}\boldsymbol{\Sigma}_T^{-1/2}. \tag{115}$$

Recall that the general expression for the $\mathbb{Q}$-bridge conditioned on $(0, \boldsymbol{x}_0), (T, \boldsymbol{x}_T)$ is

$$\boldsymbol{\mu}_{t|(\boldsymbol{x}_0, \boldsymbol{x}_1)} = \boldsymbol{\mu}_t(\boldsymbol{x}_0) + \boldsymbol{\Phi}_t\boldsymbol{\Psi}(T, t)^\top\boldsymbol{\Phi}_T^{-1}(\boldsymbol{x}_T - \boldsymbol{\mu}_T(\boldsymbol{x}_0)). \tag{116}$$

Letting $\boldsymbol{x}_0 = \boldsymbol{\mu}_T^{-1}(\boldsymbol{\Sigma}_T^{1/2}\overline{\boldsymbol{x}}_0)$ and $\boldsymbol{x}_T = \boldsymbol{\Sigma}_T^{1/2}\overline{\boldsymbol{x}}_T$ and substituting the expression for $\boldsymbol{\mu}_t^{-1}$ we have

$$\begin{aligned}\boldsymbol{\mu}_{t|(T_0^{-1}(\overline{\boldsymbol{x}}_0), T_T^{-1}(\overline{\boldsymbol{x}}_T))} &= \left(\boldsymbol{\Psi}(t, T) - \boldsymbol{\Phi}_t\boldsymbol{\Psi}(T, t)^\top\boldsymbol{\Phi}_T^{-1}\right)\boldsymbol{\Sigma}_T^{1/2}\overline{\boldsymbol{x}}_0 \\ &\quad + \boldsymbol{\Phi}_t\boldsymbol{\Psi}(T, t)^\top\boldsymbol{\Phi}_T^{-1}\boldsymbol{\Sigma}_T^{1/2}\overline{\boldsymbol{x}}_T \\ &\quad + \int_t^T \boldsymbol{\Psi}(t, s)\boldsymbol{A}_s\boldsymbol{m}_s \mathrm{d}s.\end{aligned} \tag{117}$$

Set $\boldsymbol{\Gamma}_t = \boldsymbol{\Phi}_t\boldsymbol{\Psi}(T, t)^\top\boldsymbol{\Phi}_T^{-1}$, then

$$\mathfrak{A}_t = \left(\boldsymbol{\Psi}(t, T) - \boldsymbol{\Gamma}_t\right)\boldsymbol{\Sigma}_T^{1/2}, \tag{118}$$

$$\mathfrak{B}_t = \boldsymbol{\Gamma}_t\boldsymbol{\Sigma}_T^{1/2}, \tag{119}$$

$$\mathfrak{c}_t = \int_t^T \boldsymbol{\Psi}(t, s)\boldsymbol{A}_s\boldsymbol{m}_s \mathrm{d}s. \tag{120}$$

**Case: Brownian motion**  Let's check this against the results of Bunne and Hsieh [5], who consider a class of reference processes

$$\mathrm{d}\boldsymbol{Y}_t = (c_t\boldsymbol{Y}_t + \alpha_t)\,\mathrm{d}t + g_t\,\mathrm{d}\boldsymbol{B}_t.$$

This corresponds to a special case of ours, where the drift is scalar-valued. Setting $g_t = \omega, \alpha_t = c_t = 0$, their results give us the marginal parameters of the GSB connecting $\mathcal{N}(\boldsymbol{\mu}_0, \boldsymbol{\Sigma}_0)$ and $\mathcal{N}(\boldsymbol{\mu}_1, \boldsymbol{\Sigma}_1)$. In what follows, using their results and notations of [5, Table 1], we have that $r_t = t, \overline{r}_t = 1 - t, \zeta = 0, \kappa(t, t') = \omega^2 t, \rho_t = t$. Then:

$$\boldsymbol{\mu}_t = \overline{r}_t\boldsymbol{\mu}_0 + r_t\boldsymbol{\mu}_1 + \zeta(t) - r_t\zeta(1), \tag{121}$$

$$= (1 - t)\boldsymbol{\mu}_0 + t\boldsymbol{\mu}_1, \tag{122}$$

$$\boldsymbol{\Sigma}_t = \overline{r}_t^2\boldsymbol{\Sigma}_0 + r_t^2\boldsymbol{\Sigma}_1 + r_t\overline{r}_t(\boldsymbol{C}_\sigma + \boldsymbol{C}_\sigma^\top) + \kappa(t, t')(1 - \rho_t)\mathbf{I} \tag{123}$$

$$= (1 - t)^2\boldsymbol{\Sigma}_0 + t^2\boldsymbol{\Sigma}_1 + t(1 - t)(\boldsymbol{C}_\sigma + \boldsymbol{C}_\sigma^\top) + \omega^2 t(1 - t)\mathbf{I}. \tag{124}$$

Let us compute the same quantities using our formulae. We set $\boldsymbol{A} = 0, \boldsymbol{m} = 0$ and $T = 1$. Then, $\boldsymbol{\Phi}_t = t\omega^2\mathbf{I}, \boldsymbol{\Gamma}_t = t\mathbf{I}, \boldsymbol{\Sigma}_T^{1/2} = \omega\mathbf{I}$. Then,

$$\overline{\boldsymbol{A}} = \omega^{-2}\boldsymbol{\mathcal{A}}, \qquad \overline{\boldsymbol{B}} = \omega^{-2}\boldsymbol{\mathcal{B}}.$$

Also from the underbraced expressions, we have
$$\mathfrak{A}_t = \omega(1-t)\mathbf{I}, \qquad \mathfrak{B}_t = \omega t\mathbf{I}.$$

Assembling our results, we get
$$\mathbb{E}[\boldsymbol{X}_t] = (\mathbf{I} - \boldsymbol{\Gamma}_t)\boldsymbol{a} + \boldsymbol{\Gamma}_t\boldsymbol{b}$$
$$= (1-t)\boldsymbol{a} + t\boldsymbol{b}$$
$$\mathbb{V}[\boldsymbol{X}_t] = \boldsymbol{\Omega}_t + \mathfrak{A}_t\overline{\boldsymbol{\mathcal{A}}}\mathfrak{A}_t^\top + \mathfrak{B}_t\overline{\boldsymbol{\mathcal{B}}}\mathfrak{B}_t^\top + \mathfrak{A}_t\overline{\boldsymbol{\mathcal{C}}}\mathfrak{B}_t^\top + \mathfrak{B}_t\overline{\boldsymbol{\mathcal{C}}}\mathfrak{A}_t^\top$$
$$= \boldsymbol{\Omega}_t + (1-t)^2\boldsymbol{\mathcal{A}} + t^2\boldsymbol{\mathcal{B}} + \omega^2 t(1-t)(\overline{\boldsymbol{\mathcal{C}}} + \overline{\boldsymbol{\mathcal{C}}}^\top).$$

Now $\overline{\boldsymbol{\mathcal{C}}}$ is the transport plan obtained from the *scaled* input Gaussians, of variance $\overline{\boldsymbol{\mathcal{A}}} = \boldsymbol{\mathcal{A}}/\omega^2, \overline{\boldsymbol{\mathcal{B}}} = \boldsymbol{\mathcal{B}}/\omega^2$ (in the case of Brownian reference process, there is no linear mapping via the flow map). For the scaled measures, EOT is calculated with $\varepsilon = \sigma^2 = 1$. It stands to reason that once we scale everything back, we should get $\overline{\boldsymbol{\mathcal{C}}} = \boldsymbol{\mathcal{C}}/\omega^2$. To verify this, recall that for the untransformed problem
$$\arg\min_{\pi\in\Pi(\alpha,\beta)} \frac{1}{2}\int \frac{1}{\omega^2}\|x - y\|^2 \mathrm{d}\pi + \mathrm{H}(\pi|\alpha\otimes\beta) = \arg\min_{\pi\in\Pi(\alpha,\beta)} \frac{1}{2}\int \|x - y\|^2 \mathrm{d}\pi + \omega^2\,\mathrm{H}(\pi|\alpha\otimes\beta).$$

Then for the RHS problem, we have that
$$\boldsymbol{\mathcal{C}} = \boldsymbol{\mathcal{A}}^{1/2}(\boldsymbol{\mathcal{A}}^{1/2}\boldsymbol{\mathcal{B}}\boldsymbol{\mathcal{A}}^{1/2} - \frac{\omega^4}{4}\mathbf{I})^{1/2}\boldsymbol{\mathcal{A}}^{-1/2} - \frac{\omega^2}{2}\mathbf{I}$$
$$= \omega^2(\overline{\boldsymbol{\mathcal{A}}}^{1/2}(\overline{\boldsymbol{\mathcal{A}}}^{1/2}\overline{\boldsymbol{\mathcal{B}}}\,\overline{\boldsymbol{\mathcal{A}}}^{1/2} + \frac{1}{4}\mathbf{I})^{1/2}\overline{\boldsymbol{\mathcal{A}}}^{-1/2} - \frac{1}{2}\mathbf{I})$$
$$= \omega^2\overline{\boldsymbol{\mathcal{C}}}.$$

Also it is easy to verify that $\boldsymbol{\Omega}_t = \boldsymbol{\Sigma}_{t|(x_0,x_1)} = \omega^2 t(1-t)\mathbf{I}$. So,
$$\mathbb{V}[\boldsymbol{X}_t] = \omega^2 t(1-t)\mathbf{I} + (1-t)^2\boldsymbol{\mathcal{A}} + t^2\boldsymbol{\mathcal{B}} + t(1-t)(\boldsymbol{\mathcal{C}} + \boldsymbol{\mathcal{C}}^\top), \tag{125}$$
where $\boldsymbol{\mathcal{C}}$ is the EOT plan between the unscaled measures $\mathcal{N}(\boldsymbol{a}, \boldsymbol{\mathcal{A}}), \mathcal{N}(\boldsymbol{b}, \boldsymbol{\mathcal{B}})$ with $\varepsilon = \omega^2$. This is exactly the same result as the one derived in [5].

**Case: Centered scalar OU process $A = -\lambda\mathbf{I}, m = 0$** From [5, Table 1], we have that $\overline{r}_t = \sinh(\lambda t)/\sinh(\lambda), \overline{r}_t = \sinh(\lambda t)\coth(\lambda t) - \sinh(\lambda t)\coth(\lambda), \zeta = 0$. The mean is then
$$\boldsymbol{\mu}_t = \overline{r}_t\boldsymbol{\mu}_0 + r_t\boldsymbol{\mu}_1. \tag{126}$$
With $\kappa(t, t') = \omega^2 e^{-\lambda t}\sinh(\lambda t)/\lambda$ and $\rho_t = e^{-\lambda(1-t)}\sinh(\lambda t)/\sinh(\lambda)$, the variance is
$$\boldsymbol{\Sigma}_t = \overline{r}_t^2\boldsymbol{\Sigma}_0 + r_t\boldsymbol{\Sigma}_1 + r_t\overline{r}_t(\boldsymbol{C}_{\sigma^\star} + \boldsymbol{C}_{\sigma^\star}^\top) + \kappa(t,t)(1 - \rho_t)\mathbf{I}, \tag{127}$$
where $\sigma^{\star 2} = \omega^2\sinh(\lambda)/\lambda$. For convenience, we will check things term by term and using Mathematica.

Check first the mean. We have:
$$\boldsymbol{\Phi}_t = \omega^2\left(\frac{1 - e^{-2\lambda t}}{2\lambda}\right)\mathbf{I}, \qquad \boldsymbol{\Gamma}_t = e^{-(1-t)\lambda}\left(\frac{1 - e^{-2\lambda t}}{1 - e^{-2\lambda}}\right)\mathbf{I} \tag{128}$$

Then, plugging into our formula,
$$\mathbb{E}[\boldsymbol{X}_t] = \underbrace{e^{-\lambda}(e^{(1-t)\lambda}\mathbf{I} - \boldsymbol{\Gamma}_t)}_{=\overline{r}_t}\,\boldsymbol{a} + \underbrace{\boldsymbol{\Gamma}_t}_{=r_t}\,\boldsymbol{b}. \tag{129}$$

Now the variance. We have that
$$\boldsymbol{\Sigma}_1 = \omega^2\left(\frac{e^{-\lambda}\sinh(\lambda)}{\lambda}\right)\mathbf{I}, \tag{130}$$
$$\overline{\boldsymbol{\mathcal{A}}} = \frac{\lambda e^{-\lambda}}{\omega^2\sinh(\lambda)}\boldsymbol{\mathcal{A}} = \alpha\boldsymbol{\mathcal{A}}, \tag{131}$$
$$\overline{\boldsymbol{\mathcal{B}}} = \frac{\lambda}{\omega^2 e^{-\lambda}\sinh(\lambda)}\boldsymbol{\mathcal{B}} = \beta\boldsymbol{\mathcal{B}}, \tag{132}$$
$$\mathfrak{A}_t = \left(e^{(1-t)\lambda}\mathbf{I} - \boldsymbol{\Gamma}_t\right)\boldsymbol{\Sigma}_1^{1/2}, \tag{133}$$
$$\mathfrak{B}_t = \boldsymbol{\Gamma}_t\boldsymbol{\Sigma}_1^{1/2}. \tag{134}$$

From here it is straightforward to verify that

$$\mathfrak{A}_t \overline{\mathcal{A}} \mathfrak{A}_t^\top = \overline{r}_t^2 \mathcal{A}, \quad \mathfrak{B}_t \overline{\mathcal{B}} \mathfrak{B}_t^\top = r_t^2 \mathcal{B}. \tag{135}$$

Now note that

$$\overline{\mathcal{C}} = \overline{\mathcal{A}}^{1/2} \left( \overline{\mathcal{A}}^{1/2} \overline{\mathcal{B}} \overline{\mathcal{A}}^{1/2} + \frac{1}{4}\mathbf{I} \right)^{1/2} \overline{\mathcal{A}}^{-1/2} - \frac{1}{2}\mathbf{I} \tag{136}$$

$$= \sqrt{\alpha\beta} \left[ \mathcal{A}^{1/2} \left( \mathcal{A}^{1/2} \mathcal{B} \mathcal{A}^{1/2} + \frac{(\alpha\beta)^{-1}}{4}\mathbf{I} \right)^{1/2} \mathcal{A}^{-1/2} - \frac{(\alpha\beta)^{-1/2}}{2}\mathbf{I} \right] \tag{137}$$

$$= \sqrt{\alpha\beta}\,\mathcal{C}_{1/\sqrt{\alpha\beta}}, \tag{138}$$

where $\mathcal{C}_{1/\sqrt{\alpha\beta}}$ denotes the EOT covariance for entropic regularisation level $1/\sqrt{\alpha\beta}$. Then:

$$\mathfrak{A}_t \overline{\mathcal{C}} \mathfrak{B}_t^\top + \mathfrak{B}_t \overline{\mathcal{C}}^\top \mathfrak{A}_t^\top = \underbrace{(e^{\lambda(1-t)}\mathbf{I} - \Gamma_t)\Gamma_t \Sigma_1 \sqrt{\alpha\beta}}_{r_t \overline{r}_t} (\mathcal{C}_{1/\sqrt{\alpha\beta}} + \mathcal{C}_{1/\sqrt{\alpha\beta}}^\top) \tag{139}$$

It is also easy to check that

$$\frac{1}{\sqrt{\alpha\beta}} = \frac{\omega^2 \sinh(\lambda)}{\lambda}.$$

Finally,

$$\Omega_t = \Sigma_{t|(x_0, x_1)} = \Phi_t(\mathbf{I} - e^{-2\lambda(1-t)}\Phi_t \Phi_1^{-1}) = \kappa(t, t)(1 - \rho_t)\mathbf{I}. \tag{140}$$

We have verified that all the terms in the expression for the variance agree.

## A.5 Proof of Theorem 4

*Proof.* We follow the arguments of [52] with application to the generalised Schrödinger bridge – not much changes for $\mathbb{Q}$ as the reference and we reproduce in detail the arguments as follows. For the equality of gradients, it suffices to show this for the flow matching component of the loss, since the score matching component can be handled using the exact same arguments [52]. Let $\boldsymbol{u}_{t|(\boldsymbol{x}_0, \boldsymbol{x}_T)}$ be the conditional flow between $(\boldsymbol{x}_0, \boldsymbol{x}_T)$ and $\boldsymbol{u}_t$ be the SB flow defined by

$$\boldsymbol{u}_t(\boldsymbol{x}) = \mathbb{E}_{(\boldsymbol{x}_0, \boldsymbol{x}_T) \sim \pi} \left[ \frac{p_{t|(\boldsymbol{x}_0, \boldsymbol{x}_T)}(\boldsymbol{x})}{p_t(\boldsymbol{x})} \boldsymbol{u}_{t|(x_0, x_T)}(\boldsymbol{x}) \right], \quad p_t(\boldsymbol{x}) = \mathbb{E}_{(\boldsymbol{x}_0, \boldsymbol{x}_T) \sim \pi}\, p_{t|(\boldsymbol{x}_0, \boldsymbol{x}_T)}(\boldsymbol{x}).$$

Let $\boldsymbol{u}_t^\theta$ be the neural flow approximation with parameter $\theta$. Assuming that $p_t(\boldsymbol{x}) > 0$ for all $(t, \boldsymbol{x})$ we then have:

$$\nabla_\theta \mathbb{E}_{(\boldsymbol{x}_0, \boldsymbol{x}_T) \sim \pi} \mathbb{E}_{\boldsymbol{x} \sim p_{t|(\boldsymbol{x}_0, \boldsymbol{x}_T)}} \|\boldsymbol{u}_t^\theta(\boldsymbol{x}) - \boldsymbol{u}_{t|(\boldsymbol{x}_0, \boldsymbol{x}_T)}(\boldsymbol{x})\|^2 - \nabla_\theta \mathbb{E}_{\boldsymbol{x} \sim p_t(\boldsymbol{x})} \|\boldsymbol{u}_t^\theta(\boldsymbol{x}) - \boldsymbol{u}_t(\boldsymbol{x})\|^2 \tag{141}$$

$$= \nabla_\theta \mathbb{E}_{(\boldsymbol{x}_0, \boldsymbol{x}_T) \sim \pi} \mathbb{E}_{\boldsymbol{x} \sim p_{t|(\boldsymbol{x}_0, \boldsymbol{x}_T)}} \left[ \|\boldsymbol{u}_t^\theta(\boldsymbol{x})\|^2 + \|\boldsymbol{u}_{t|(\boldsymbol{x}_0, \boldsymbol{x}_T)}(\boldsymbol{x})\|^2 - 2\langle \boldsymbol{u}_t^\theta(\boldsymbol{x}), \boldsymbol{u}_{t|(\boldsymbol{x}_0, \boldsymbol{x}_T)}(\boldsymbol{x}) \rangle \right] \tag{142}$$

$$\quad - \nabla_\theta \mathbb{E}_{\boldsymbol{x} \sim p_t(\boldsymbol{x})} \left[ \|\boldsymbol{u}_t^\theta(\boldsymbol{x})\|^2 + \|\boldsymbol{u}_t(\boldsymbol{x})\|^2 - 2\langle \boldsymbol{u}_t^\theta(\boldsymbol{x}), \boldsymbol{u}_t(\boldsymbol{x}) \rangle \right] \tag{143}$$

$$= 2\nabla_\theta \mathbb{E}_{(\boldsymbol{x}_0, \boldsymbol{x}_T) \sim \pi} \mathbb{E}_{\boldsymbol{x} \sim p_{t|(\boldsymbol{x}_0, \boldsymbol{x}_T)}} \langle \boldsymbol{u}_t^\theta(\boldsymbol{x}), \boldsymbol{u}_t(\boldsymbol{x}) \rangle - \langle \boldsymbol{u}_t^\theta(\boldsymbol{x}), \boldsymbol{u}_{t|(\boldsymbol{x}_0, \boldsymbol{x}_T)}(\boldsymbol{x}) \rangle, \tag{144}$$

where in the last line terms not depending on $\theta$ are zero and we use the fact that $\mathbb{E}_{(\boldsymbol{x}_0, \boldsymbol{x}_T)} \mathbb{E}_{\boldsymbol{x}|(\boldsymbol{x}_0, \boldsymbol{x}_T)} f(\boldsymbol{x}) = \mathbb{E}_{\boldsymbol{x}} f(\boldsymbol{x})$. Now, using the relation between $\boldsymbol{u}_t$ and $\boldsymbol{u}_{t|(\boldsymbol{x}_0, \boldsymbol{x}_T)}$,

$$\mathbb{E}_{\boldsymbol{x} \sim p_t(\boldsymbol{x})} \langle \boldsymbol{u}_t^\theta(\boldsymbol{x}), \boldsymbol{u}_t(\boldsymbol{x}) \rangle = \int \mathrm{d}p_t(\boldsymbol{x}) \langle \boldsymbol{u}_t^\theta(\boldsymbol{x}), \boldsymbol{u}_t(\boldsymbol{x}) \rangle \tag{145}$$

$$= \int \mathrm{d}p_t(\boldsymbol{x}) \left\langle \boldsymbol{u}_t^\theta(\boldsymbol{x}), \int \mathrm{d}\pi(\boldsymbol{x}_0, \boldsymbol{x}_T) \frac{p_{t|(\boldsymbol{x}_0, \boldsymbol{x}_T)}(\boldsymbol{x})}{p_t(\boldsymbol{x})} \boldsymbol{u}_{t|(\boldsymbol{x}_0, \boldsymbol{x}_T)}(\boldsymbol{x}) \right\rangle \tag{146}$$

$$= \int \mathrm{d}\pi(\boldsymbol{x}_0, \boldsymbol{x}_T) \int \mathrm{d}\boldsymbol{x}\, p_t(\boldsymbol{x}) \left\langle \boldsymbol{u}_t^\theta(\boldsymbol{x}), \frac{p_{t|(\boldsymbol{x}_0, \boldsymbol{x}_T)}(\boldsymbol{x})}{p_t(\boldsymbol{x})} \boldsymbol{u}_{t|(\boldsymbol{x}_0, \boldsymbol{x}_T)}(\boldsymbol{x}) \right\rangle \tag{147}$$

$$= \int \mathrm{d}\pi(\boldsymbol{x}_0, \boldsymbol{x}_T) \int \mathrm{d}p_{t|(\boldsymbol{x}_0, \boldsymbol{x}_T)}(\boldsymbol{x}) \left\langle \boldsymbol{u}_t^\theta(\boldsymbol{x}), \boldsymbol{u}_{t|(\boldsymbol{x}_0, \boldsymbol{x}_T)}(\boldsymbol{x}) \right\rangle \tag{148}$$

$$= \mathbb{E}_{(\boldsymbol{x}_0, \boldsymbol{x}_T) \sim \pi} \mathbb{E}_{\boldsymbol{x} \sim p_{t|(\boldsymbol{x}_0, \boldsymbol{x}_T)}} \langle \boldsymbol{u}_t^\theta(\boldsymbol{x}), \boldsymbol{u}_{t|(\boldsymbol{x}_0, \boldsymbol{x}_T)}(\boldsymbol{x}) \rangle. \tag{149}$$

So we conclude that the two gradients are equal. Clearly, the *unconditional* loss can be rewritten over candidate flow and score fields $(\hat{\boldsymbol{u}}, \hat{\boldsymbol{s}})$ as

$$(\hat{\boldsymbol{u}}, \hat{\boldsymbol{s}}) \mapsto \|\hat{\boldsymbol{u}} - \boldsymbol{u}\|^2_{L^2(\mathrm{d}p_t(\boldsymbol{x})\mathrm{d}t)} + \|\lambda_t(\hat{\boldsymbol{s}} - \boldsymbol{s})\|^2_{L^2(\mathrm{d}p_t(\boldsymbol{x})\mathrm{d}t)},$$

where we denote by $\|h_t(\boldsymbol{x})\|^2_{L^2(\mathrm{d}p_t(\boldsymbol{x})\mathrm{d}t)} = \int_0^1 \mathrm{d}t \int \mathrm{d}p_t(\boldsymbol{x}) \|h_t(\boldsymbol{x})\|^2_{L^2(\mathbb{R}^d)}$ for a test function $h_t :$ $[0,1] \times \mathbb{R}^d \to \mathbb{R}^d$. This loss is zero iff $\hat{\boldsymbol{u}} = \boldsymbol{u}$ and $\hat{\boldsymbol{s}} = \boldsymbol{s}$ $(\mathrm{d}p_t \times \mathrm{d}t)$-almost everywhere.

Let $\mathbb{P}$ denote the law of the Schrödinger bridge as per (SBP-dyn) and write $p_t(\boldsymbol{x}) = \mathbb{P}_t$ to mean its marginal at time $t$. Then in (2) and for $\mathbb{Q}$ a mvOU process (1), identifying:

- $\mathrm{d}\mathbb{P}^\star_{0T} = \pi$ where $\pi$ is prescribed in Proposition 1, and

- $\mathbb{Q}^{x_0, x_T}_t = p_{t|(\boldsymbol{x}_0, \boldsymbol{x}_T)}(\boldsymbol{x})$ where $p_{t|(\boldsymbol{x}_0, \boldsymbol{x}_T)}$ is defined as in Theorem 2,

substituting all these into (2) one has

$$p_t(\boldsymbol{x}) = \int \mathrm{d}\pi(\boldsymbol{x}_0, \boldsymbol{x}_T) p_{t|(\boldsymbol{x}_0, \boldsymbol{x}_T)}.$$

Consequently, there exists $\boldsymbol{u}_t(\boldsymbol{x})$ such that $\partial_t p_t(\boldsymbol{x}) = -\nabla \cdot (p_t(\boldsymbol{x}) u_t(\boldsymbol{x}))$. Writing $\boldsymbol{s}_t(\boldsymbol{x}) = \nabla_x \log p_t(\boldsymbol{x})$ to be the score, recognising terms from the probability flow ODE, it follows that that the SDE

$$\mathrm{d}\boldsymbol{X}_t = (\boldsymbol{u}_t(\boldsymbol{X}_t) + \boldsymbol{D}\boldsymbol{s}_t(\boldsymbol{X}_t))\mathrm{d}t + \boldsymbol{\sigma}\mathrm{d}\boldsymbol{B}_t \tag{150}$$

generates the marginals $p_t(\boldsymbol{x})$ of the Schrödinger bridge. Since $\mathbb{P}$ is characterised as a mixture of $\mathbb{Q}$-bridges, it follows that $\boldsymbol{X}_t$ defined by the SDE (150) generates the Markovianisation of $\mathbb{P}$ (see Appendix B of [52]). Moreover, $\mathbb{P}$ is the *unique* process that is both Markov and a mixture of $\mathbb{Q}$-bridges [28, 45]. □

## A.6 Additional background

**Hamilton-Jacobi-Bellman equation**   Consider the *deterministic* control problem on $[0, T]$:

$$V_0(x_0) = \min_u \int_0^T C(x_t, u_t)\,\mathrm{d}t + D(x_T). \tag{151}$$

subject to a dynamics $\dot{x}_t = F(x_t, u_t)$. $V_t(x)$ is the *value* function for the problem starting at $(t, x)$ up to the final time $T$. The function $x_T \mapsto D(x_T)$ specifies a cost on the final value. The corresponding Hamilton-Jacobi-Bellman (HJB) equation [23, Section 3.11] is

$$\partial_t V_t(x) + \min_{u_t} \{\langle \nabla_x V_t(x), F(x, u_t(x)) \rangle + C(x, u_t(x))\} = 0, \tag{152}$$

subject to the final boundary condition $V_T(x) = D(x)$. A heuristic derivation is as follows. Considering a time interval $(t, t + \delta t)$ and a path $x_t$, it's clear that

$$V_t(x_t) = \min_u \left\{ \int_t^{t+\delta t} C(x_s, u_s)\mathrm{d}s + V_{t+\delta t}(x_{t+\delta t}) \right\}. \tag{153}$$

Using Taylor expansion and the constraint $\dot{x}_t = F(x_t, u_t)$ we have to leading order that

$$V_{t+\delta t}(x_{t+\delta t}) = V_t(x_t) + (\partial_t V_t(x_t) + \langle \nabla_x V_t(x_t), F(x_t, u_t) \rangle)\,\delta t + O(\delta t^2). \tag{154}$$

Substituting back and also approximating the integral, we find that

$$V_t(x_t) = \min_u\ C(x_t, u_t)\delta t + V_t(x_t) + (\partial_t V_t(x_t) + \langle \nabla_x V_t(x_t), F(x_t, u_t) \rangle)\,\delta t \tag{155}$$

Cancelling terms, rearranging and taking a limit $\delta t \downarrow 0$, we get the desired result.

**Stochastic Hamilton-Jacobi-Bellman** Now we consider the *stochastic* variant of the HJB, in which case $X_t$ is driven by an SDE of the form

$$\mathrm{d}X_t = F(X_t, u_t)\,\mathrm{d}t + \sigma_t\,\mathrm{d}B_t. \tag{156}$$

The value function is thus in expectation:

$$V_0(x_0) = \min_u \mathbb{E}\left\{\int_0^T C(X_t, u_t)\,\mathrm{d}t + D(X_T)\right\}. \tag{157}$$

Carrying out the same expansion as before, we have

$$V_t(x_t) = \min_u \mathbb{E}\left\{\int_t^{t+\delta t} C(X_s, u_s)\mathrm{d}s + V_{t+\delta t}(X_{t+\delta t})\right\}. \tag{158}$$

Note that in the above, $(X_{t+\delta t}|X_t = x_t)$ is a random variable and hence so is $V_{t+\delta t}(X_{t+\delta t})$ which is why it appears in the expectation. Using Itô's formula to expand this, we get

$$V_{t+\delta t}(X_{t+\delta t}) = V_t(X_t) + \left[\partial_t V_t(X_t) + \frac{1}{2}\nabla \cdot (\sigma_t \sigma_t^\top \nabla_x V_t(X_t))\right]\mathrm{d}t + \langle \nabla_x V_t(X_t), \mathrm{d}X_t\rangle$$

$$= V_t(X_t) + \left[\partial_t V_t(X_t) + \frac{1}{2}\nabla \cdot (\sigma_t \sigma_t^\top \nabla_x V_t(X_t)) + \langle \nabla_x V_t(X_t), F(X_t, u_t)\rangle\right]\mathrm{d}t$$

$$+ \langle \nabla_x V_t(X_t), \sigma_t \mathrm{d}B_t\rangle.$$

Plugging this in, cancelling terms, and noting that the final term has zero expectation, we find that

$$0 = \partial_t V_t(X_t) + \min_u \left\{\mathcal{A}[V_t](X_t, u_t) + C(X_t, u_t)\right\} \tag{159}$$

where $\mathcal{A}$ is the generator of the SDE governing $X_t$, i.e.

$$\mathcal{A}[f](x, u) = \langle F(x, u), \nabla_x f(x)\rangle + \frac{1}{2}\nabla \cdot (\sigma_t \sigma_t^\top \nabla_x f). \tag{160}$$

The HJB equation for stochastic control problem is therefore

$$0 = \partial_t V_t(x) + \min_u \left\{\mathcal{A}[V_t](x, u_t(x)) + C(x, u_t(x))\right\}, \tag{161}$$

i.e. this is the same as the HJB for the deterministic case, except with a diffusive term arising from the stochasticity.

# B   Experiment details

For (mvOU, BM)-OTFM and IPFP, all computations were carried out by CPU (8x Intel Xeon Gold 6254). NLSB and SBIRR computations were accelerated using a single NVIDIA L40S GPU. Code is available at `https://github.com/zsteve/mvOU_SBP`.

## B.1   Gaussian benchmarking

We consider settings of dimension $d \in \{2, 5, 10, 25, 50\}$ with data consisting of an initial and terminal Gaussian distribution and a mvOU reference process constructed as follows. We sample a $d \times 2$ submatrix $\boldsymbol{U}$ from a random $d \times d$ orthogonal matrix. Then for the reference process, $\boldsymbol{A}$ is constructed as the $d \times d$ matrix

$$\boldsymbol{A} = \boldsymbol{U}\begin{bmatrix} 0 & 1 \\ -2.5 & 0 \end{bmatrix}\boldsymbol{U}^\top.$$

Similarly, we let $\boldsymbol{m} = \boldsymbol{U}\begin{bmatrix} 1 & -1 \end{bmatrix}^\top$. For the marginals, we take $\mathcal{N}(\boldsymbol{\mu}_0, \boldsymbol{\Sigma}_0)$ and $\mathcal{N}(\boldsymbol{\mu}_1, \boldsymbol{\Sigma}_1)$ where

$$\boldsymbol{\mu}_0 = \boldsymbol{U}\begin{bmatrix} -2.5 & -0.5 \end{bmatrix}^\top, \qquad \boldsymbol{\Sigma}_0 = \boldsymbol{U}\begin{bmatrix} 0.1 & 0.005 \\ 0.005 & 0.1 \end{bmatrix}\boldsymbol{U}^\top + 0.1\mathbf{I}.$$

$$\boldsymbol{\mu}_1 = \boldsymbol{U}\begin{bmatrix} 0.5 & 2.5 \end{bmatrix}^\top, \qquad \boldsymbol{\Sigma}_1 = \boldsymbol{U}\begin{bmatrix} 1.1 & -2 \\ -2 & 1.1 \end{bmatrix}\boldsymbol{U}^\top + 0.1\mathbf{I}.$$

For each $d$, the initial and final marginals are approximated by $N = 128$ samples $\{\boldsymbol{x}_0^i\}_{i=1}^N \sim \mathcal{N}(\boldsymbol{\mu}_0, \boldsymbol{\Sigma}_0)$, $\{\boldsymbol{x}_1^i\}_{i=1}^N \sim \mathcal{N}(\boldsymbol{\mu}_1, \boldsymbol{\Sigma}_1)$. We fix $\boldsymbol{\sigma} = \boldsymbol{I}$ and compute the exact marginal parameters $(\boldsymbol{\mu}_t, \boldsymbol{\Sigma}_t)_{t\in[0,1]}$ using the formulas of Theorem 3 and reference parameters $(\boldsymbol{A}, \boldsymbol{m})$. Additionally, we compute the SDE drift $\boldsymbol{v}_{\mathrm{SB}}(t, \boldsymbol{x})$ of the mvOU-GSB using (17).

**(mvOU, BM)-OTFM**   We apply mvOU-OTFM as per Algorithm 1 to learn a conditional flow matching approximation to the Schrödinger bridge. We choose to parameterise the probability flow and score fields using two feed-forward neural networks $\boldsymbol{u}_\theta(t, \boldsymbol{x}) = \text{NN}_\theta(d + 1, d)(t, \boldsymbol{x})$ and $\boldsymbol{s}_\varphi(t, \boldsymbol{x}) = \text{NN}_\varphi(d + 1, d)(t, \boldsymbol{x})$, each with $[64, 64, 64]$ hidden dimensions and ReLU activations. We use a batch size of 64 and learning rate $10^{-2}$ for 2,500 iterations using the AdamW optimiser. For BM-OTFM, the same training procedure was used except the reference process was taken to be Brownian motion with unit diffusivity.

**IPFP**   We use the IPFP implementation provided by the authors of [53], specifically using the variant of their algorithm based on Gaussian process approximations to the Schrödinger bridge drift [53, Algorithm 2]. We provide the mvOU reference process as the prior drift function, i.e. $\boldsymbol{x} \mapsto \boldsymbol{A}(\boldsymbol{x} - \boldsymbol{m})$ and employ an exponential kernel for the Gaussian process vector field approximation. We run the IPFP algorithm for 10 iterations, which is double that used in the original publication. Since IPFP explicitly constructs forward and reverse processes, at each time $0 \leqslant t \leqslant 1$, IPFP outputs *two* estimates of the Schrödinger bridge marginal, one from each process. We consider both outputs and distinguish between them using the $(\rightarrow, \leftarrow)$ symbols in Table 1. For the vector field estimate we employ only the forward drift.

**NLSB**   We use the NLSB implementation provided by the authors of [25]. Following [12, Section 4.6], for a mvOU reference process $\mathrm{d}\boldsymbol{X}_t = \boldsymbol{f}_t\, \mathrm{d}t + \sigma\, \mathrm{d}\boldsymbol{B}_t$, the generalised SB problem (SBP-dyn) can be rewritten as a stochastic control problem

$$\min_{\boldsymbol{u}_t} \mathbb{E}\left[\int_0^1 \frac{1}{2}\|\boldsymbol{u}_t\|_2^2\, \mathrm{d}t\right], \text{ subject to } \mathrm{d}\boldsymbol{X}_t = (\boldsymbol{f}_t + \boldsymbol{u}_t)\, \mathrm{d}t + \sigma\, \mathrm{d}\boldsymbol{B}_t.$$

On the other hand, the Lagrangian SB problem [25, Definition 3.1] is

$$\min_{\boldsymbol{v}_t} \mathbb{E}\left[\int_0^1 L(t, \boldsymbol{x}, \boldsymbol{v}_t(\boldsymbol{x}))\mathrm{d}t\right], \text{ subject to } \mathrm{d}\boldsymbol{X}_t = \boldsymbol{v}_t\, \mathrm{d}t + \sigma\, \mathrm{d}\boldsymbol{B}_t.$$

This shows that the appropriate Lagrangian to use is $L(t, \boldsymbol{x}, \boldsymbol{v}_t(\boldsymbol{x})) = \frac{1}{2}\|\boldsymbol{v}_t(\boldsymbol{x}) - \boldsymbol{f}_t(\boldsymbol{x})\|_2^2$. We apply this in the mvOU setting by taking $\boldsymbol{f}_t(\boldsymbol{x}) = \boldsymbol{A}(\boldsymbol{x} - \boldsymbol{m})$. The NLSB approach backpropagate through solution of the SDE to directly learn a neural approximation of the SB drift $\boldsymbol{v}_t$. We train NLSB the same hyperparameters as used in the original publication, using the Adam optimiser with learning rate $10^{-3}$ for a total of 2,500 epochs.

**Metrics**   We measure both the approximation error of Gaussian Schrödinger bridge marginals as well as the error in vector field estimation. For the marginal reconstruction, for each method and for each $0 \leqslant t \leqslant 1$ we sample points integrated forward in time and compute the mean $\hat{\boldsymbol{\mu}}_t$ and variance $\hat{\boldsymbol{\Sigma}}_t$ from samples. We then compare to the ground truth marginals $(\boldsymbol{\mu}_t, \boldsymbol{\Sigma}_t)$ computed from exact formulas using the Bures-Wasserstein metric (62). For the vector field, for each method and for each time $0 \leqslant t \leqslant 1$ we sample $M = 1024$ points from the ground truth SB marginal $\boldsymbol{N}(\boldsymbol{\mu}_t, \boldsymbol{\Sigma}_t)$ and empirically estimate $\|\hat{\boldsymbol{v}}_{\text{SB}} - \boldsymbol{v}_{\text{SB}}\|_{\mathcal{N}(\boldsymbol{\mu}_t, \boldsymbol{\Sigma}_t)}$ from samples. All experiments were repeated over 5 independent runs and summary statistics for metrics are shown in Table 1.

### B.2   Gaussian mixture example

We consider initial and terminal marginals sampled from Gaussian mixtures. At time $t = 0$, we sample from

$$\mathcal{N}\left(\boldsymbol{U}\begin{bmatrix}-0.5 & -0.5\end{bmatrix}^\top, 0.01\boldsymbol{U}\boldsymbol{U}^\top\right), \qquad \mathcal{N}\left(\boldsymbol{U}\begin{bmatrix}0.5 & 0.5\end{bmatrix}^\top, 0.0625\boldsymbol{U}\boldsymbol{U}^\top\right)$$

in a 1:1 ratio, and at time $t = 1$ we sample in the same fashion from

$$\mathcal{N}\left(\boldsymbol{U}\begin{bmatrix}-2.5 & -2.5\end{bmatrix}^\top, 0.01\boldsymbol{U}\boldsymbol{U}^\top\right), \qquad \mathcal{N}\left(\boldsymbol{U}\begin{bmatrix}2.5 & 2.5\end{bmatrix}^\top, 0.25\boldsymbol{U}\boldsymbol{U}^\top\right).$$

Here, the matrix $\boldsymbol{U}$ and reference process parameters $(\boldsymbol{A}, \boldsymbol{m})$ are the same as used for the previous Gaussian example for $d = 10$. All other training details are the same as for the Gaussian example.

## B.3 Repressilator example

Based on the system studied in [44], we simulate stochastic trajectories from the system

$$
\begin{aligned}
\frac{\mathrm{d}x_1}{\mathrm{d}t} &= \left( \frac{\beta}{1 + (x_3/k)^n} - \gamma x_1 \right) \mathrm{d}t + \sigma \, \mathrm{d}B_t^{(1)} \\
\frac{\mathrm{d}x_2}{\mathrm{d}t} &= \left( \frac{\beta}{1 + (x_1/k)^n} - \gamma x_2 \right) \mathrm{d}t + \sigma \, \mathrm{d}B_t^{(2)} , \qquad \begin{bmatrix} \beta \\ n \\ k \\ \gamma \\ \sigma \end{bmatrix} = \begin{bmatrix} 10 \\ 3 \\ 1 \\ 1 \\ 0.1 \end{bmatrix} , \qquad (162) \\
\frac{\mathrm{d}x_3}{\mathrm{d}t} &= \left( \frac{\beta}{1 + (x_2/k)^n} - \gamma x_3 \right) \mathrm{d}t + \sigma \, \mathrm{d}B_t^{(3)} ,
\end{aligned}
$$

with initial condition $\mathcal{N}([1, 1, 2], 0.01\mathbf{I})$ and simulated for the time interval $t \in [0, 10]$ using the Euler-Maruyama discretisation. Snapshots were sampled at $T = 10$ time points evenly spaced on $[0, 10]$, each comprising of 100 samples.

We employ Algorithm 2 to this data, running for 5 iterations starting from an initial Brownian reference process $\boldsymbol{A} = \boldsymbol{0}, \boldsymbol{m} = \boldsymbol{0}$. For the inner loop running mvOU-OTFM (Algorithm 1), we parameterise the probability flow and score as in the Gaussian example, with feed-forward networks of hidden dimension $[64, 64, 64]$ and ReLU activations. We run mvOU-OTFM for 1,000 iterations using the AdamW optimiser, a batch size of 64 and a learning rate of $10^{-2}$. At each step of the outer loop, we employ ridge regression to fit the updated mvOU reference parameters $(\boldsymbol{A}, \boldsymbol{m})$. We do this using the standard `RidgeCV` method implemented in the `scikit-learn` package, which automatically selects the regularisation parameter.

We also carry out hold-one-out runs where for $2 \leqslant i \leqslant 9$ (i.e. all time-points except for the very first and last), Algorithm 2 is applied to $T - 1$ snapshots with the snapshot at $t_i$ held out. Once the mvOU reference parameters are learned, forward integration of the learned mvOU-SB is used to predict the marginal at $t_i$. We report the reconstruction error in terms of the earth-mover distance (EMD) and energy distance [41]. Table 2 shows results averaged over held-out timepoints, and full results (split by timepoint) are shown in Table 5.

Since the system marginals are unimodal we reason that they can be reasonably well approximated by Gaussians. For each $2 \leqslant i \leqslant 9$ we fit multivariate Gaussians to the snapshots at $t_{i-1}, t_i, t_{i+1}$ and use the results of Theorem 3 with the fitted mvOU reference output by Algorithm 2 to solve the mvOU-GSB between $p_{t_{i-1}}, p_{t_{i+1}}$. This is illustrated in Figure 4 for $i = 4$, and we show the full results for all timepoints in Figure 6 in comparison with the standard Brownian GSB.

**SBIRR** We apply SBIRR [44] using the implementation provided with the original publication, which notably includes an improved implementation of IPFP [53] that utilises GPU acceleration. We provide the reference vector field $\boldsymbol{x} \mapsto \boldsymbol{A}(\boldsymbol{x} - \boldsymbol{m})$ and seek to learn a reference process in one of two families: (1) mvOU processes, i.e. we consider the family of reference drifts $\hat{\boldsymbol{A}}(\boldsymbol{x} - \hat{\boldsymbol{m}})$ where $\hat{\boldsymbol{A}}, \hat{\boldsymbol{m}}$ are to be fit, or (2) general drifts, i.e. we parameterise the drift using a feed-forward neural network with hidden dimensions $[64, 64, 64]$. For each choice of reference family, we run Algorithm 1 of [44] for 5 outer iterations and 10 inner IPFP iterations, as was also done in the original paper.

## B.4 Cell cycle scRNA-seq

The metabolic labelled cell cycle dataset of [4] is obtained and preprocessed following the tutorial available with the Dynamo [40] package. This gives a dataset of $N = 2,793$ cells, embedded in 30 PCA dimensions. In addition to transcriptional state $\{\boldsymbol{x}_i\}_{i=1}^N$, Dynamo uses metabolic labelling data to predict the transcriptional velocity $\{\hat{\boldsymbol{v}}_i\}_{i=1}^N$ for each cell.

To fit the reference process parameters $(\boldsymbol{A}, \boldsymbol{m})$, we use again ridge regression via the `RidgeCV` method in `scikit-learn`. We train mvOU-OTFM using Algorithm 1 with $\sigma = 0.3$, parameterising the probability flow and score as previously using feed-forward networks of hidden dimensions $[64, 64, 64]$ and train with a batch size of 64, learning rate of $10^{-2}$ for a total of 1,000 iterations.

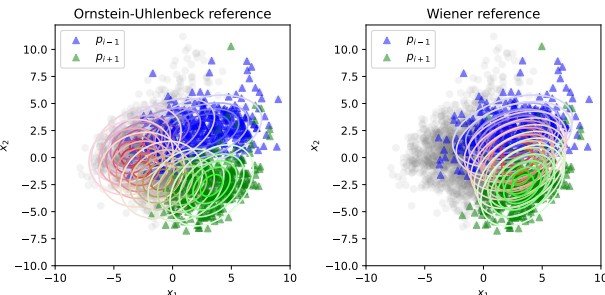

Figure 6: **Repressilator mvOU-GSB interpolation.** Using the learned mvOU reference process, we interpolate between $p_{i-1}$ (blue) and $p_{i+1}$ (green). Middle timepoint $p_i$ is shown in red.

Figure 7: **Cell cycle mvOU-GSB interpolation.** Using the learned mvOU reference process and scale factor $\gamma = 50$, we interpolate between the first snapshot $p_1$ (blue) and last snapshot $p_T$ (green). All computations are done in $d = 30$ and shown in leading 2 PCs.

| | | | | | Scale factor $\gamma$ | | | | | | |
|---|---|---|---|---|---|---|---|---|---|---|---|---|
| | $t$ | 0.0 | 10.0 | 20.0 | 30.0 | 40.0 | 50.0 | 60.0 | 70.0 | 80.0 | 90.0 | 100.0 |
| | 0.25 | 40.18 | 30.8 | 22.55 | 16.34 | 12.77 | 11.07 | 10.26 | 10.11 | 10.69 | 11.93 | 13.65 |
| $d = 50$ | 0.5 | 28.59 | 18.47 | 11.53 | 9.62 | 12.36 | 15.32 | 15 | 12.52 | 10.64 | 11.02 | 13.96 |
| | 0.75 | 15.31 | 10.31 | 7.47 | 7.06 | 8.18 | 8.9 | 8.5 | 7.9 | 8.09 | 9.42 | 11.92 |
| | 0.25 | 44.49 | 36.82 | 29.97 | 24.19 | 19.99 | 17.48 | 16.13 | 15.29 | 14.78 | 14.7 | 15.18 |
| $d = 100$ | 0.5 | 32.48 | 23.75 | 17.12 | 13.55 | 13.79 | 16.84 | 19.78 | 20.21 | 18.25 | 15.7 | 14.22 |
| | 0.75 | 19.33 | 14.95 | 12 | 10.74 | 11.19 | 12.68 | 13.98 | 14.36 | 14.03 | 13.69 | 13.82 |

Table 4: Single cell interpolation results for $d = 50, 100$.

---

**Algorithm 2** Iterated reference fitting with mvOU-OTFM

---

**Input:** Samples $\{x_i^{t_j}\}_{i=1}^{N_j}$ from multiple snapshots at times $\{t_j\}_{j=1}^{T}$, initial mvOU reference parameters $(A, m)$, diffusivity $D = \frac{1}{2}\sigma\sigma^{\top}$
**Initialise:** Probability flow field $u_t^{\theta}(x)$, score field $s_t^{\varphi}(x)$.
**Define:** $\texttt{FitReference}(X, V) := \arg\min_{A,m} \|V - A(X - m)\|_2^2 + \lambda\|A\|_F^2 + \gamma\|m\|^2$
$\hat{\rho}_j \leftarrow N^{-1}\sum_{i=1}^{N}\delta_{x_i^{t_j}}, 1 \leqslant j \leqslant T$          *Form empirical marginals*
**while** not converged **do**
    $(u_t^{\theta}, s_t^{\theta}) \leftarrow \texttt{fitOTFM}_{(A,m,D)}(\hat{\rho}_1, \ldots, \hat{\rho}_T)$      *Fit flow and score with reference parameters*
    $v_i^{t_j} \leftarrow (u_{t_j}^{\theta} + Ds_{t_j}^{\varphi})(x_i^{t_j}), \quad 1 \leqslant i \leqslant N_j, 1 \leqslant j \leqslant T$      *Get SDE drift*
    $A, m \leftarrow \texttt{FitReference}(\{(v_i^{t_j})_{i=1}^{N_i}\}_{j=1}^{T}, \{(x_i^{t_j})_{i=1}^{N_i}\}_{j=1}^{T})$      *Update reference parameters*
**end while**

---

| | | | | | Leave-one-out marginal interpolation error | | | |
|---|---|---|---|---|---|---|---|---|
| Error metric | $t$ | Iterate 0 | Iterate 1 | Iterate 2 | Iterate 3 | Iterate 4 | SBIRR (mvOU) | SBIRR (MLP) |
| | 1 | $3.59 \pm 0.17$ | $3.14 \pm 0.12$ | $2.28 \pm 0.15$ | $2.08 \pm 0.11$ | $\mathbf{2.02 \pm 0.11}$ | $2.24 \pm 0.28$ | $2.71 \pm 0.41$ |
| | 2 | $5.20 \pm 0.47$ | $2.59 \pm 0.29$ | $1.62 \pm 0.28$ | $1.27 \pm 0.25$ | $\mathbf{1.13 \pm 0.16}$ | $3.13 \pm 0.62$ | $2.29 \pm 0.92$ |
| | 3 | $3.23 \pm 0.24$ | $1.42 \pm 0.18$ | $1.10 \pm 0.14$ | $0.86 \pm 0.08$ | $\mathbf{0.83 \pm 0.10}$ | $2.67 \pm 0.85$ | $1.39 \pm 0.55$ |
| EMD | 4 | $1.48 \pm 0.20$ | $0.52 \pm 0.05$ | $0.47 \pm 0.05$ | $\mathbf{0.47 \pm 0.06}$ | $0.48 \pm 0.06$ | $1.38 \pm 0.46$ | $0.94 \pm 0.28$ |
| | 5 | $2.50 \pm 0.40$ | $1.43 \pm 0.65$ | $\mathbf{1.12 \pm 0.32}$ | $1.29 \pm 0.11$ | $1.21 \pm 0.13$ | $1.63 \pm 0.17$ | $1.40 \pm 0.71$ |
| | 6 | $6.18 \pm 0.41$ | $3.42 \pm 0.50$ | $2.18 \pm 0.28$ | $1.91 \pm 0.38$ | $\mathbf{1.75 \pm 0.17}$ | $2.40 \pm 0.32$ | $1.96 \pm 1.41$ |
| | 7 | $2.56 \pm 0.25$ | $3.09 \pm 1.53$ | $2.13 \pm 0.46$ | $2.23 \pm 0.55$ | $1.93 \pm 0.50$ | $1.45 \pm 0.20$ | $\mathbf{0.51 \pm 0.13}$ |
| | 8 | $2.29 \pm 0.26$ | $2.12 \pm 0.09$ | $1.82 \pm 0.33$ | $1.81 \pm 0.31$ | $\mathbf{1.81 \pm 0.16}$ | $1.93 \pm 0.61$ | $2.14 \pm 0.40$ |
| | 1 | $3.00 \pm 0.09$ | $2.75 \pm 0.07$ | $2.24 \pm 0.10$ | $2.10 \pm 0.07$ | $\mathbf{2.05 \pm 0.08}$ | $2.20 \pm 0.18$ | $2.56 \pm 0.26$ |
| | 2 | $3.53 \pm 0.21$ | $2.27 \pm 0.19$ | $1.66 \pm 0.20$ | $1.38 \pm 0.21$ | $\mathbf{1.26 \pm 0.13}$ | $2.61 \pm 0.33$ | $2.05 \pm 0.61$ |
| | 3 | $2.29 \pm 0.17$ | $1.31 \pm 0.16$ | $1.00 \pm 0.10$ | $0.80 \pm 0.06$ | $\mathbf{0.78 \pm 0.07}$ | $2.10 \pm 0.52$ | $1.25 \pm 0.43$ |
| Energy | 4 | $0.93 \pm 0.14$ | $0.25 \pm 0.06$ | $0.17 \pm 0.04$ | $0.15 \pm 0.02$ | $\mathbf{0.14 \pm 0.01}$ | $0.99 \pm 0.38$ | $0.84 \pm 0.25$ |
| | 5 | $1.10 \pm 0.18$ | $0.53 \pm 0.27$ | $\mathbf{0.45 \pm 0.13}$ | $0.55 \pm 0.04$ | $0.51 \pm 0.07$ | $0.64 \pm 0.10$ | $0.66 \pm 0.37$ |
| | 6 | $2.49 \pm 0.12$ | $1.22 \pm 0.10$ | $1.13 \pm 0.15$ | $1.01 \pm 0.22$ | $0.91 \pm 0.11$ | $1.43 \pm 0.17$ | $\mathbf{0.90 \pm 0.64}$ |
| | 7 | $0.97 \pm 0.13$ | $1.39 \pm 0.65$ | $1.02 \pm 0.21$ | $1.05 \pm 0.25$ | $0.91 \pm 0.25$ | $0.65 \pm 0.09$ | $\mathbf{0.17 \pm 0.08}$ |
| | 8 | $0.54 \pm 0.11$ | $0.57 \pm 0.03$ | $0.56 \pm 0.11$ | $0.55 \pm 0.10$ | $0.55 \pm 0.04$ | $0.48 \pm 0.08$ | $\mathbf{0.36 \pm 0.06}$ |

Table 5: Full results for repressilator example

