# OpenReview forum: "Learning non-equilibrium diffusions with Schrödinger bridges: from exactly solvable to simulation-free"
_NeurIPS.cc/2025/Conference — NeurIPS 2025 poster_

### Official Review · Reviewer_uH9r · 2025-06-30

**Clarity:** 3
**Significance:** 3
**Originality:** 3
**Rating:** 4
**Confidence:** 4

**Summary:**

The study frames the Schrödinger bridge (SB) problem around a multivariate Ornstein–Uhlenbeck reference process, keeping the math tractable while covering non-equilibrium systems. It provides a closed-form solution for Gaussian marginals (mvOU-GSB) and, for arbitrary marginals, introduces MVOU-OTFM, a simulation-free method that learns flow and score via conditional flow matching rooted in exact mvOU-bridge formulas. Across several benchmarks, the proposed approach trains quickly and achieves the lowest error rates among all baselines.

**Questions:**

* The reference process ought to be treated as a prior that is chosen before any data are seen. In SBIRR the prior dynamics remain fixed and are not directly inferred from the observations. Such an experiment in single cell-data, If we instead fit the OU parameters directly to the data, how is that fundamentally different from simply augmenting SBP-Dyn with additional parameters?

* The experiments are limited to low-dimensional settings with small sample sizes, such as Repressilator or cell-dynamics data. Can the proposed algorithm still produce an optimal transport plan for larger-scale datasets?

**Ethical Concerns:**

["NO or VERY MINOR ethics concerns only"]

**Final Justification:**

Although the mvOU process is well known and its derivations already exist, I acknowledge that a closed-form matching objective for the dynamic-SB problem with an mvOU prior has not been thoroughly treated in the literature. This work applies mvOU to the SB problem and demonstrates its effectiveness on multi-marginal data. The main limitation is the reliance on a pre-computed OT coupling, which can bias the coupling in high-dimensional settings. The manuscript should explicitly discuss this approximation error.

**Limitations:**

yes

**Quality:**

3

**Strengths And Weaknesses:**

Strengths:

* The paper delivers the first closed-form Schrödinger bridge solution for Gaussian marginals under a fully multivariate, possibly asymmetric OU reference, broadening the scope of Gaussian SB theory.

* Leveraging these formulas, MVOU-OTFM eliminates SDE simulations by directly regressing flow and score fields, yielding order-of-magnitude speed-up.

* From synthetic mixtures to 30-dimensional single-cell RNA-seq, MVOU-OTFM consistently achieves the lowest error.

Weaknesses:

* The paper’s claim that extending a scalar OU SDE to a multivariate one is a major contribution is not persuasive. Multivariate OU processes and their closed-form solutions are already well known. A stronger advance would be to use a full non-diagonal diffusion matrix and to show a simulation-free bridge algorithm. Instead, the authors keep a diagonal diffusion matrix in their experiments, so the theorem feels too limited to count as a significant contribution on its own.

* The central aim of any Schrödinger Bridge algorithm is to learn dynamics that generate the optimal coupling, since that coupling is notoriously hard to identify for real data. Section 3.3 claims that a single Sinkhorn solve achieves this goal, yet the claim is fragile. Sinkhorn tackles an entropically regularised transport problem, its cost and memory scale poorly with dimension and sample size, and the regularisation introduces a persistent bias unless the regularisation strength tends to zero. Under these conditions the resulting dynamics cannot be considered optimal SB solution. Established SB methods improve accuracy through iterative proportional fitting (IPF) [1,2] or iterative Markovian fitting (IMF) [3,4]. Please add either empirical results or a theoretical argument that compares the proposed single Sinkhorn step to these iterative procedures. Otherwise the use of “SB: exactly solvable” in the title feels misleading.

----
    [1] Diffusion schrödinger bridge with applications to score-based generative modeling.
    [2] Likelihood Training of Schrödinger Bridge using FBSDEs Theory.
    [3] Diffusion Scrödinger Bridge Matching.
    [4] Diffusion Bridge Mixture Transports, Schrödinger Bridge Problems and Generative Modeling

---

> ### Author Rebuttal · Authors · 2025-07-30
>
> Thank you for your time to read our paper and we appreciate your detailed feedback.
>
> > ... extending a scalar OU SDE to a multivariate one ... is not persuasive. Multivariate OU processes ... are already well known.
>
> We **agree** with the reviewer that multivariate OU (mvOU) processes are well-known – their analytical tractability is **exactly what we exploit** to develop our theoretical and computational results.
>
> As the reviewer acknowledges, our characterisation of the the Schrodinger bridge with mvOU reference are **novel** and we consistently achieves the **lowest error** with much less computation compared to approaches that are more general at the expense of relying on simulation.
> We attribute these improvements to the **use of closed form expressions for the flow and score**.
>
> We point out that the gap between scalar OU SDEs (as in generative modelling literature) and their generic multivariate counterparts is **non-trivial** – in fact, a scalar OU process is _always_ gradient-driven. Thus, while sufficient for generative modelling tasks, scalar OU dynamics are **not appropriate for modelling non-equilibrium systems in physical systems**.
>
> Instead, mvOU processes allow for a **generic drift matrix**. The motivation for mvOU dynamics arises naturally -- for a nonlinear system (such as the oscillator in our Figure 4), a **canonical analysis technique** [1, 2] in the physics literature is to study its linearization -- this is a **linear SDE** where the **drift matrix is exactly the Jacobian of $f(x)$**.
>
> Concretely, an SDE $dX_t = f(X_t) dt + \sigma dB_t$ with a fixed point at $f(0) = 0$, can be linearized: $d\tilde{X}_t = A \tilde{X}_t dt + \sigma dB_t$ and $A = (\partial_x f)(0)$.
>
> In the case of gradient dynamics $f(x) = \nabla_x U(x)$, note that $A = (\partial_{xx} U)(0)$ is **symmetric**. The relevance of considering an **asymmetric drift matrix** is to model non-gradient systems.
>
> > A stronger advance would be to use a full non-diagonal diffusion matrix and to show ... algorithm
>
> Thank you for the suggestion. We acknowledge that our examples use diagonal diffusion. As we explain below, we provide theoretical grounds that considering diagonal diffusion and generic drift is sufficient.
>
> - **(i).** **All** of our theoretical results (Thms. 1-4) and our algorithms (Algs. 1-2) are already for a **generic diffusion matrix** $\bm{\sigma}$ (see e.g. line 164, 213)
> - **(ii).** **Any** Linear SDE with a non-diagonal diffusion matrix is **equivalent** to one with a diagonal diffusion matrix under a change of basis.
>
> For **(i)**, while we typeset vector quantities in bold, we will make this convention explicit.
>
> For **(ii)**, note that **any diffusion matrix** $D = \tfrac{1}{2} \sigma \sigma^\top$ can be diagonalized by a change of basis, since it is PSD.
>
> Therefore we can **generically work with the diagonal diffusion setting**, since given any non-diagonal diffusion we can choose an orthogonal basis of $\mathbb{R}^d$ under which it becomes diagonal. **On the other hand**, asymmetry in the drift component **cannot** be eliminated by the same trick.
>
> We formalise this as follows:
>
> **[Lemma.] Up to a change of basis, any linear SDE with generic drift and diffusion matrix is equal in its law to a linear SDE with diagonal diffusion.**
>
> Let $d X_t = A X_t dt + \sigma d B_t$ be a linear SDE with generic drift $A \in \mathbb{R}^{d \times d}$ and diffusion $\sigma \in \mathbb{R}^{d \times d}$. There exists an orthogonal basis $U \in \mathbb{R}^{d \times d}$ such that $Y_t = U^\top X_t$ obeys a linear SDE with drift matrix $U^\top A U$ and a diagonal diffusion matrix $\Lambda = \mathrm{diag}(\lambda)$, $\lambda = (\lambda_1, \ldots, \lambda_d) \ge 0$
>
> _Proof_ Let $D = \frac{1}{2} \sigma \sigma^\top$, this is PSD and can be written $D = \frac{1}{2} U \Lambda^2 U^\top$ where $U$ is orthogonal and $\Lambda = \mathrm{diag}(\lambda) \succeq 0$. Consider the SDEs
> (a.) $$ d X_t = A X_t dt + \sigma d B_t$$ and
> (b.) $$ d X_t = A X_t dt + U \Lambda U^\top d B_t.$$
> (a.) has generator $(\mathcal{L} f)(x) = \langle \partial_x f(x), Ax \rangle + \frac{1}{2} tr(\sigma^\top \partial_{xx} f(x) \sigma)$.
>
> Computing the generator for (b): $(\tilde{\mathcal{L}} f)(x) = \langle \partial_x f(x), Ax \rangle + \frac{1}{2} tr( U \Lambda U^\top \partial_{xx} f(x) U \Lambda U^\top)$.
> Note that $tr( U \Lambda U^\top \partial_{xx} f U \Lambda U^\top) = tr(\partial_{xx} f U \Lambda^2 U^\top) = tr(\sigma^\top \partial_{xx} f \sigma)$.
> Hence both SDEs share the same generator, and so are equal in law.
>
> Since orthogonal transformations of Brownian increments are also Brownian increments, we have that $U^\top dB_t = d\tilde{B}_t$. Substituting into (b.), we have $d X_t = A X_t dt + U \Lambda d \tilde{B}_t$. Then,
> $$ d(U^\top X_t) = U^\top A X_t dt + \Lambda d \tilde{B}_t \Longrightarrow d Y_t = U^\top A U Y_t dt + \Lambda d \tilde{B}_t.$$
> $□$
>
> > Section 3.3 claims that a single Sinkhorn solve achieves this goal, yet the claim is fragile...
>
> We aim to solve the SBP with **discrete** initial and final data, in the form of **empirical distributions**. This is different from the setting of estimating the **continuum SBP from samples** where the initial and final _densities_ must be estimated as well as the law of the bridge (e.g. as in [3]). The **SBP between empirical distributions**, the problem we consider, is well-defined as-is (see e.g. Section 6 of [4]), and that is the viewpoint we take.
>
> Put differently, for source and target $(\mu, \nu)$ supported on $\{ x_1, \ldots, x_M \}$ and $\{ y_1, \ldots, y_N \}$ respectively, the $\mathbb{Q}$-SB takes the form $\mathbb{P} = \sum_{i, j} \pi_{ij} \mathbb{Q}^{(x_i, y_j)}$, where $\{ \pi : \sum_{ij} \pi_{ij} = 1 \}$ is the solution to an **entropic OT problem**.
>
> That this is the case can be read from Equations 1.1-1.2 of [4]; alternatively Proposition D.1 of [5] or Prop 2.1 of [6]. This is exactly what we discuss in Section 2.1 of our paper, and the SB between empirical distributions is **precisely** what we aim to learn.
>
> Regarding regularisation, we point out that it is a standard fact in SBP literature that the entropic term in the "static" EOT problem (see Eq.(SBP-static) in our paper, also Prop. 2.1 of [6]) is **tied to the noise level of the system**, and therefore **must not be taken to zero**. This is in contrast to Monge map estimation, in which any $\varepsilon > 0$ introduces a bias.
>
> We acknowledge that these points deserve an extended discussion, and we will add this to the revised paper.
>
> > Established SB methods improve accuracy through (IPF) [1,2] or iterative Markovian fitting (IMF) [3,4]
>
> We gently point out that **we currently compare to IPFP**: [7, 8] are based on IPF with various drift families. We show our approach is more accurate and faster.
>
> In sum, the use of a single Sinkhorn step for learning the SB between two empirical distributions is justified both from a theoretical point of view (see our answer to the question above and [4, 5, 6]), as well as empirical observations. For the Gaussian mixture benchmarks (Table 1 of our paper),
>
> **Note that we have now increased the number of dimensions from 50 to 100 (see Table 2 in our response to `xnYH`)**
>
> > Otherwise the use of “SB: exactly solvable” in the title feels misleading.
>
> To clarify, the term “exactly solvable” in our title refers to our **analytical results for the Gaussian case**, owing to the existence of matrix-analytic expressions for quantities involving Gaussian marginals and mvOU dynamics.
>
> **In this setting, our results provide an exact solution to the SBP**. We see that ‘exactly solvable’ may confuse readers, and we will make the abstract and introduction more explicit to avoid risking ambiguity.
>
> > The reference process ought to be treated as a prior that is chosen before any data are seen. In SBIRR the prior dynamics remain fixed and are not directly inferred from the observations. Such an experiment in single cell-data, If we instead fit the OU parameters directly to the data, how is that fundamentally different from simply augmenting SBP-Dyn with additional parameters?
>
> To clarify, in SBIRR prior dynamics are **not fixed but updated at each step** [8, Alg. 1], thus using information across observations to update the prior
>
> In our cell cycle example, OU params are fit to RNA velocity data [9] -- we construct the prior using information that is **external** to that used for fitting SB.
>
> > The experiments are limited to low-dimensional settings with small sample sizes... Can the proposed algorithm still produce an optimal transport plan for larger-scale datasets?
>
> **Scaling in $N$**: [Alg. 1] samples **batches** at each iteration so it can be scaled to **arbitrarily large datasets**. Theoretically one should solve EOT over the full dataset, but minibatch OT is widely used to scale to large, even streaming data [6]
>
> **Scaling in $d$**: **In Table 1,2 of our response to `xnYH`**, we report new results for single cell $d = 50, 100$ and the synthetic Gaussian for $d = 100$. Considering most single cell studies use $d < 100$ PCs, our method can scale to the $d$ used in practice
>
> [1] Godrèche C, Luck JM. Characterising the nonequilibrium stationary states of Ornstein–Uhlenbeck processes
>
> [2] Lax M. Fluctuations from the nonequilibrium steady state
>
> [3] Pavon M, Trigila G, Tabak EG. The data‐driven schrödinger bridge
>
> [4] Léonard C. A survey of the schrodinger problem and some of its connections with optimal transport.
>
> [5] Lavenant H, Zhang S, Kim YH, Schiebinger G. Towards a mathematical theory of trajectory inference
>
> [6] Tong A, Malkin N, Fatras K, et al Simulation-free schrodinger bridges via score and flow matching
>
> [7] Vargas F, Thodoroff P, Lamacraft A, Lawrence N. Solving schrödinger bridges via maximum likelihood
>
> [8] Shen Y, Berlinghieri R, Broderick T Multi-marginal schrodinger bridges with iterative reference refinement
>
> [9] Qiu X, Zhang Y, et al Mapping transcriptomic vector fields of single cells

---

> > ### Comment · Reviewer_uH9r · 2025-08-06
> >
> > Thank you for the comprehensive rebuttal and for conducting higher-dimensional experiments. Most of my concerns are addressed and your explanation clarifies the novelty of employing mvOU reference dynamics and the resulting closed-form matching objective. Now I agree with that although mvOU is well known, its integration with the SB formulation has not been fully explored in the literature, so this work maintains its originality. Hence, I will raise the score 2->4 accordingly.
> >
> > However, I still have concerns about relying on a pre-computed OT coupling. The authors of [1] warn that minibatch OT can introduce noticeable approximation error in high-dimensional settings. This limitation should be stated clearly in the paper.

---

> > > ### Author Response · Authors · 2025-08-06
> > >
> > > Thank you for the response and we are very glad to hear most of the concerns were addressed by our rebuttal.
> > > Regarding the inherent bias incurred by using mini-batch OT, we absolutely agree. We should clarify that in our experiments, we avoid this by conducting entropic OT on the **full dataset** of available samples, thereby avoiding this bias. In practice, when scaling up to large datasets, we agree that minibatch OT is the typical practice for scaling up OT flow matching and this incurs an error which may be significant, depending on the batch size and dimension. We will add a discussion of this point in the final paper.

---

> ### Comment · Area_Chair_7ecr · 2025-08-05
>
> Dear reviewer,
> If not already, please take a look at the authors' rebuttal, and discuss if necessary.
> Thanks,
> -AC

---

### Official Review · Reviewer_xh1L · 2025-07-01

**Clarity:** 3
**Significance:** 2
**Originality:** 2
**Rating:** 5
**Confidence:** 3

**Summary:**

This paper studies Schrodinger bridge problem with the reference processes being a multivariate OU process with generic drift. The authors derived a explicit form of the SDE which the bridges should satisfy, as well as a score function and probability flow. With Gaussian marginals, the process can be solved explicitly. In the non-gaussian case, the authors proposed mv-OU-OTFM and show it solves the dynamic. Numerical experiments are done to confirm their claims.

**Questions:**

See above.

**Ethical Concerns:**

["NO or VERY MINOR ethics concerns only"]

**Final Justification:**

My main concern was the mathematical contribution can be considered weak, but the authors explained to me how their work is different from the existing ones, and I am convinced that this paper has enough theoretical justification with good experimental results. Hence I am updating my rating.

**Limitations:**

yes

**Quality:**

3

**Strengths And Weaknesses:**

1. The authors give some introduction in Schrodinger bridges, flow matching, probability flows, etc., which might help readers from a broader background.

2. Although Theorem 1 is adapted from an existing work [1], stating it in this context is still helpful. Similar situation for theorem 2. But I think the authors can be more clear about these results were proved somewhere else, for example, by adding the citation following the name of the theorems.

3. Theorem 3 gives an explicit expression of the Gaussian bridge. The proof is build on the one in [2]. While it generalizes results in [2], one could argue the proof technique is not novel from a mathematic perspective. Note that S should be in upper case in the reference.

4. The authors proposed mvOU-OTFM as the score and flow matching algorithm for OU Schrodinger bridges. The proof idea in Theorem 4 is adapted from [3].

5. Authors done thorough experiments showing mvOU-OTFM outperforms IPML, NLSB, etc. Impressively, with 4 iteration, mvOU-OTFM achieves better results than SBIRR considering it runs much faster. In page 8, I think by algorithm 2, the authors mean algorithm 1? Table 2 is presented but never referred to.


[1] Chen et al. Stochastic bridges of linear systems. 2015.

[2] Bunne et al. The Schrödinger bridge between gaussian measures has a closed form. 2023

[3] Tong et al. Simulation-free Schrodinger bridges via score and flow matching. 2023.

---

> ### Author Rebuttal · Authors · 2025-07-30
>
> Thank you for the time taken to review our work and for your suggestions.
>
> > Although Theorem 1 is adapted from an existing work [1], stating it in this context is still helpful. Similar situation for theorem 2. But I think the authors can be more clear about these results were proved somewhere else, for example, by adding the citation following the name of the theorems.
>
> Thank you for these comments and we agree that the motivation for stating these results in this way is to **ensure cohesion and flow** in the text.
> We agree that the results of Theorem 1 can be read implicitly in [1] and **we will add “adapted from Chen et al.” in the theorem title** in the revised text.
> On the other hand, the results of Theorem 2 as stated are **not readily available** from any papers that we are aware of. As such, we do not put a citation in the theorem statement. We will add some discussion on the closest related results in the literature, and point out that the result is built upon Theorem 1.
>
> > Theorem 3 gives an explicit expression of the Gaussian bridge. The proof is build on the one in [2]. While it generalizes results in [2], one could argue the proof technique is not novel from a mathematic perspective.
>
> We agree that parts of the proof technique for Thm. 3 overlap with the details of [2]. Indeed, both the approaches of [2] and ours proceed via the **well-known and classical characterisation of the SBP as a reciprocal process** (see e.g. Eq. 2 of our submission, also [4, Eq. 1.1]) and then utilising the characterisation of $\mathbb{Q}$-bridges. **In our view, this is the “canonical” path towards building the SBP where it is analytically tractable.**
> We realise that the proof approach is a helpful discussion and context to provide in the main text, and we will add a few sentences about this and the relation to the approach of [2] in the revised paper.
>
> The step of the proof that draws from [2] is the derivation of the drift of the SBP SDE via **calculating the SDE’s generator**, and we acknowledge this in the proof (see Supplementary Material, _"Dynamic $\mathbb{Q}$-GSB : SDE representation"_).
> We argue that our derivation **streamlines** the argument compared to [2], making the derivations easier to follow while being more general. In particular, [2] utilises "heavy machinery" such as the "central identity of QFT" (see [2], Preliminaries for the Proof of Theorem 3) to compute the generator. In our analysis, we find that this is in fact unnecessary and standard arguments suffice.
>
> > Note that S should be in upper case in the reference.
>
> Thank you, indeed the name should be capitalised. We will rectify this in the revised paper.
>
> > Authors done thorough experiments showing mvOU-OTFM outperforms IPML, NLSB, etc.
>
> **We point out that we have now added additional experiments, scaling up to $d = 100$ for the Gaussian example and $d = 50, 100$ for the cell cycle example (see Tables 1, 2 of our response to `xnYH`)**
>
> > In page 8, I think by algorithm 2, the authors mean algorithm 1?
>
> Algorithm 2 refers to the "iterated reference fitting" approach that uses Alg. 1 as an inner subroutine. Due to issues with space, the pseudocode for Alg. 2 is contained in the appendix (see Supplementary Material). Thank you for bringing it to our attention that this may be confusing -- in the revised paper, we will either include it in the main text (using the extra 10th page for camera ready), or put an explicit reference to appendix.
>
> >   Table 2 is presented but never referred to.
>
> Thank you for pointing this out, this has been rectified.
>
> [1] Chen et al. Stochastic bridges of linear systems. 2015.
>
> [2] Bunne et al. The Schrödinger bridge between gaussian measures has a closed form. 2023
>
> [3] Tong et al. Simulation-free Schrodinger bridges via score and flow matching. 2023.
>
> [4] Léonard C. A survey of the schr\" odinger problem and some of its connections with optimal transport. arXiv preprint arXiv:1308.0215. 2013 Aug 1.

---

> > ### Comment · Reviewer_xh1L · 2025-08-06
> >
> > Thank the authors for addressing my concerns and questions. I updated the rating to 5.

---

> ### Comment · Area_Chair_7ecr · 2025-08-05
>
> Dear reviewer,
> If not already, please take a look at the authors' rebuttal, and discuss if necessary.
> Thanks,
> -AC

---

### Official Review · Reviewer_kfao · 2025-07-04

**Clarity:** 4
**Significance:** 3
**Originality:** 3
**Rating:** 5
**Confidence:** 3

**Summary:**

The paper studies the Schrödinger bridge problem with a multivariate Ornstein-Uhlenbeck process as a reference measure. By allowing for a potentially asymmetric drift matrix, this model can be used to study systems exhibiting non-gradient out-of-equilibrium dynamics which are relevant, as stated by the authors, in the contexts of biological and social systems. The paper develops both exact solutions for Gaussian marginals and a simulation-free learning method for general marginals. The approach is based on flow and score matching to avoid expensive numerical simulations. This results in a scalable process for recovering stochastic dynamics from initial and final conditions which is compared to previous methods in different settings.

**Questions:**

I think the case for the paper would be stronger if the authors included a more thorough discussion on the dynamical meaning of assuming a linear drift and state-independent diffusion coefficients. As it is now it just seems to be for theoretical convenience as it allows for the explicit closed-form expressions that their methods are based on. But there is no methodological discussion on when could these assumptions be reasonable and when not and if, in the future, any of these could be removed or softened.

**Ethical Concerns:**

["NO or VERY MINOR ethics concerns only"]

**Limitations:**

This is discussed on the previous section.

**Paper Formatting Concerns:**

In the appendices, many formula fall outside of the margins of the documents. Some reformatting will be needed to void this.

**Quality:**

4

**Strengths And Weaknesses:**

The paper is well written and structured and discusses a topic which has generated attention lately. The main theoretical contributions (Theorems 2 and 3) are clean and interesting. These are complemented with comprehensive simulations that show a clear improvement with respect to previous methods. All this makes the paper a good contribution to the venue. However, it appears to me that the main drawback of the paper is that it does not include a strong discussion of the limitations of the strong dynamical assumptions of the model.

---

> ### Author Rebuttal · Authors · 2025-07-30
>
> Thank you for your time taken to evaluate our submission and for the positive assessment of our work.
>
> > it appears to me that the main drawback of the paper is that it does not include a strong discussion of the limitations of the strong dynamical assumptions of the model.
>
> **We agree that a more in-depth discussion of the limitations of the multivariate Ornstein-Uhlenbeck (mvOU) reference process is warranted, and we will add in the revised paper a discussion of this.**
>
> By considering a multivariate Ornstein-Uhlenbeck reference described by the SDE $dX_t = A (X_t - m) dt + \sigma dB_t$, our framework can describe a family of SBP processes that are **strictly larger than the Brownian SBP**. In other words, we can recover the standard SBP problem (such as the ones studied in [1, 2]) by taking $A, m = 0$.
>
> We want to emphasize that the linearity assumption applies **only** to the **reference** dynamics $\mathbb{Q}$, and not the SB dynamics $\mathbb{P}$. By modelling the SB flow field using neural networks, the mvOU-OTFM algorithm (Alg. 1) can deal with **general non-linear dynamics** of $\mathbb{P}$ in the case of general, non-Gaussian marginals.
>
> Indeed, as the reviewer points out, our framework is still not fully general since it does not allow for reference dynamics with non-linear drift, such as those that can be modelled by NLSB [3] and IPFP [4]. Our work **highlights the tradeoff** between (i) analytically tractable but more restrictive models and (ii) more expressive non-parametric models, which are much less tractable to learn, both in terms of computational cost and statistical complexity.
>
> > I think the case for the paper would be stronger if the authors included a more thorough discussion on the dynamical meaning of assuming a linear drift and state-independent diffusion coefficients. As it is now it just seems to be for theoretical convenience as it allows for the explicit closed-form expressions that their methods are based on. But there is no methodological discussion on when could these assumptions be reasonable and when not and if, in the future, any of these could be removed or softened.
>
> **Thank you for pointing this out and we agree that further discussion of this is desirable. We will add discussion of all of these points in the revised paper. Also see above for our response about the assumption of a linear drift.**
>
> * For **application to non-linear systems**, the linear reference dynamics can be **justified by local linearization** of the nonlinear system about a fixed point, assuming that the observed data arise in the vicinity of the fixed point. This is a **canonical analysis technique** for studying nonlinear dynamics in the physics literature [5].
>
> * While we hinted at this link to linearization about a fixed point (line 263), we will make this connection more explicit: **linearization of a non-linear stochastic dynamical system about a stable point yields a multivariate OU process** where the drift matrix is the Jacobian of the nonlinear system: for a nonlinear SDE $dX_t = f(X_t) dt + \sigma dB_t$ with a fixed point at $f(0) = 0$, one can consider a linearized dynamics $d\tilde{X}_t = A \tilde{X}_t dt + \sigma dB_t$ where $A = (\partial_x f)(0)$.
>
> * We would like to emphasize that the benefit of tractability is not only theoretical but also **methodological/computational** – we attribute availability of closed form expressions to the speed and accuracy of our method in our experiments (see Tables 1 and 2 in initial submission), relative to competing methods which rely upon numerical integration for more expressive but expensive solvers.
>
> Regarding scope for assumptions to be relaxed, we note that concurrent work [6] proposes CurlyFM, which trains a neural approximation to the reference process instead of our exact computations. While this may be a direction along which limitations could be relaxed, we leave a detailed comparison to future work.
>
> > In the appendices, many formula fall outside of the margins of the documents. Some reformatting will be needed to void this.
>
> Thank you, this has been rectified.
>
> [1] Tong A, Malkin N, Fatras K, Atanackovic L, Zhang Y, Huguet G, Wolf G, Bengio Y. Simulation-free schr\" odinger bridges via score and flow matching. arXiv preprint arXiv:2307.03672. 2023 Jul 7.
>
> [2] Bunne C, Hsieh YP, Cuturi M, Krause A. The schrödinger bridge between gaussian measures has a closed form. InInternational Conference on Artificial Intelligence and Statistics 2023 Apr 11 (pp. 5802-5833). PMLR.
>
> [3] Koshizuka T, Sato I. Neural lagrangian schr\" odinger bridge: Diffusion modeling for population dynamics. arXiv preprint arXiv:2204.04853. 2022 Apr 11.
>
> [4] Vargas F, Thodoroff P, Lamacraft A, Lawrence N. Solving schrödinger bridges via maximum likelihood. Entropy. 2021 Aug 31;23(9):1134.
>
> [5] Godrèche C, Luck JM. Characterising the nonequilibrium stationary states of Ornstein–Uhlenbeck processes. Journal of Physics A: Mathematical and Theoretical. 2018 Dec 18;52(3):035002.
>
> [6] Petrović K, Atanackovic L, Kapusniak K, Bronstein MM, Bose J, Tong A. Curly flow matching for learning non-gradient field dynamics. InICLR 2025 Workshop on Machine Learning for Genomics Explorations 2025.

---

> ### Comment · Area_Chair_7ecr · 2025-08-05
>
> Dear reviewer,
> If not already, please take a look at the authors' rebuttal, and discuss if necessary.
> Thanks,
> -AC

---

> ### Comment · Reviewer_kfao · 2025-08-06
>
> I thank the authors for addressing my comments. These have all been properly answered so I keep my rating of 5 (accept).

---

### Official Review · Reviewer_xnYH · 2025-07-07

**Clarity:** 3
**Significance:** 2
**Originality:** 3
**Rating:** 5
**Confidence:** 2

**Summary:**

The paper considers a special case of Schroedinger's bridge where marginals are Gaussian and the dynamics are an OU process instead of Brownian motion. The method is developed and demonstrated on some example data sets.

**Questions:**

I appreciate this is standard in the SBP literature, but I'm always a bit confused as to why it's useful to look at just over 2000 genes, with their expression compressed to 30 PCs. I suspect it's more likely about getting scaling on the data set to work. It would be great to see a larger example that feels a little more plausible.

I didn't quite understand why a larger data set wasn't run at the end. I appreciate the comparisons are with methods that are computationally expensive. But can't you go much larger with this approach? If not why not? Would be good to clarify. I wold have gone higher for significance if we'd been able to see such an example

**Ethical Concerns:**

["NO or VERY MINOR ethics concerns only"]

**Limitations:**

yes

**Quality:**

4

**Strengths And Weaknesses:**

The paper does a good job of connecting to the physics literature, and mapping the ideas onto language that is easily understood by machine learning researchers.

In a relatively compressed space the paper does a good job of covering the related work and how the proposed algorithm fits into the wider landscape.

Again, given the challenges of a technical exposition, the paper does a good job of giving confidence in the method in the main text while providing additional resources in appendices and references.

The experiments show some illustrative examples and also a larger example on gene expression data.

---

> ### Author Rebuttal · Authors · 2025-07-30
>
> Thank you for taking the time to assess our work and for your positive evaluation and suggestions!
>
> > The paper considers a special case of Schroedinger's bridge where marginals are Gaussian and the dynamics are an OU process instead of Brownian motion. The method is developed and demonstrated on some example data sets.
>
> We would like to clarify that the **classical** Schrodinger’s bridge (with Brownian reference) is in fact a **special case** of the mvOU-SBP we consider (setting zero drift and isotropic diffusion), rather than the other way around.
> We would also like to clarify that the case of Gaussian marginals is **only one part of our contribution**, for which we provide thorough analytical results. For the non-analytically tractable (but much more application-relevant) case of **non-Gaussian** initial and final measurements, we introduce an algorithm based on **score and flow matching** to solve the mvOU-SBP.
>
> > I appreciate this is standard in the SBP literature, but I'm always a bit confused as to why it's useful to look at just over 2000 genes, with their expression compressed to 30 PCs. I suspect it's more likely about getting scaling on the data set to work. It would be great to see a larger example that feels a little more plausible.
>
> Thank you for raising the point of dimensionality reduction and pre-processing in the single cell data example. Indeed, as you point out, related SBP works like [2] and, more broadly, applications on single-cell RNA sequencing data use these kinds of preprocessing pipelines as the norm. The reviewer points out two choices:
>
>  - (i.) **Filtering** to ~2000 highly variable genes
>  - (ii.) **PCA** to 30 components.
>
> We provide additional results scaling up to 100 PCs (see below, Table 1), and explain the necessity of these preprocessing steps in what follows.
>
> In our case, these preprocessing steps were the same ones used for preparing the dataset in the original publication, see [4, STAR Methods]. In the present work, our main focus is on **theoretical and methodological development** of the SBP with multivariate OU reference, so we did not re-process the data in the initial submission.
>
> **(i. Filtering)**: Many genes are non-informative for the biological process of interest (in this case, the cell cycle) and consist only of technical artifacts or zeros. Filtering by variance is therefore almost **universally used as best practice**, see e.g. “Feature selection” section of [1]: typically "1,000 and 5,000 HVGs are selected for downstream analysis".
>
> **(ii. PCA)**: This is again a **standard pre-processing practice** to select a “natural” set of coordinates for downstream analysis and helps with the extremely noisy nature of single cell data. Additionally, the theory of SBP has an underlying Gaussian assumption (e.g. that the reference process is driven by a white noise). PCA performs a kind of whitening transformation to the data, making it more favourable for modelling.
>
> > I didn't quite understand why a larger data set wasn't run at the end. I appreciate the comparisons are with methods that are computationally expensive. But can't you go much larger with this approach? If not why not? Would be good to clarify. I wold have gone higher for significance if we'd been able to see such an example
>
> To provide some context, for applications of Schrodinger bridges to single cell data
>
> * [2]: 20-30 dim. PCA
> * [3, 5]: 5 dim. PCA
>
> One limitation which also applies to any method leveraging Gaussian OT [2, 6] is that linear solves involving PSD matrices of dimension $d \times d$ are required, with a intrinsic computational complexity of $O(d^3)$. We realize that this limitation could be more clearly stated, and we will do so in the revised paper. We point out that there may be ways to alleviate this, e.g. leveraging low-rank approximations of covariance matrices, and this is a potential direction for future work.
>
> **Motivated by reviewer `xnYH`'s feedback, we reprocessed the cell cycle data and scaled up the number of PCs to 50 and 100 dimensions.**
>
> ### Table 1 - cell cycle interpolation (equivalent to Figure. 5b)
> Marginal interpolation error between first and last snapshot as a function of the reference velocity scale parameter, $\gamma$.
> #### (a) 50 PCs
> |t|0.0|10.0|20.0|30.0|40.0|50.0|60.0|70.0|80.0|90.0|100.0|
> |----:|------:|-------:|-------:|-------:|-------:|-------:|-------:|-------:|-------:|-------:|--------:|
> |0|0|0|0|0|0|0|0|0|0|0|0|
> |1|40.18|30.8|22.55|16.34|12.77|11.07|10.26|10.11|10.69|11.93|13.65|
> |2|28.59|18.47|11.53|9.62|12.36|15.32|15|12.52|10.64|11.02|13.96|
> |3|15.31|10.31|7.47|7.06|8.18|8.9|8.5|7.9|8.09|9.42|11.92|
> |4|0|0|0|0|0|0|0|0|0|0|0.01|
>
> #### (b) 100 PCs
> t|0.0|10.0|20.0|30.0|40.0|50.0|60.0|70.0|80.0|90.0|100.0|
> |----:|------:|-------:|-------:|-------:|-------:|-------:|-------:|-------:|-------:|-------:|--------:|
> |0|0|0|0|0|0|0|0|0|0|0|0|
> |1|44.49|36.82|29.97|24.19|19.99|17.48|16.13|15.29|14.78|14.7|15.18|
> |2|32.48|23.75|17.12|13.55|13.79|16.84|19.78|20.21|18.25|15.7|14.22|
> |3|19.33|14.95|12|10.74|11.19|12.68|13.98|14.36|14.03|13.69|13.82|
> |4|0|0|0|0|0|0|0|0|0|0|0|
>
> We apply also mvOU-OTFM (Alg.1) for different number of PCs, as done in Figure 5(a) of our paper. Since this year there is no possibility to provide images in the rebuttal, we are unable to provide the visualisation of results for $d = 50, 100$.
> However, regarding the point of scalability, we report the following runtimes for mvOU-OTFM **on CPU**:
>
> ### Runtimes for mvOU-OTFM (Alg. 1) for cell cycle example
> * d = 30, 1m13s
> * d = 50, 1m49s
> * d = 100, 5m20s
>
> We would like to point out that these runtimes are **significantly faster**, even **without using GPU**, compared to methods relying on numerical integration such as NLSB and SBIRR.
>
> To further demonstrate scaling our method, we extend the Gaussian results (Table 1 of our paper) from 50 to 100 dimensions.
>
> ### Table 2 - Gaussian SB results, $d = 100$
> #### (a) Marginal error (Bures-Wasserstein dist.)
> |   dim | OU-GSB              | BM-GSB           | IPML(←)          | IPML(→)          | NLSB             |
> |------:|:--------------------|:-----------------|:-----------------|:-----------------|:-----------------|
> |   100 | **6.84 $\pm$ 0.78** | 15.14 $\pm$ 0.95 | 16.19 $\pm$ 1.87 | 14.38 $\pm$ 0.38 | 17.40 $\pm$ 0.13 |
> #### (b) Force error (L2)
>
> |   dim | OU-GSB               | BM-GSB           | IPML             | NLSB             |
> |------:|:---------------------|:-----------------|:-----------------|:-----------------|
> |   100 | **10.45 $\pm$ 0.18** | 15.53 $\pm$ 0.14 | 15.30 $\pm$ 0.39 | 16.49 $\pm$ 0.07 |
>
>
>
> [1] Luecken MD, Theis FJ. Current best practices in single‐cell RNA‐seq analysis: a tutorial. Molecular systems biology. 2019 Jun;15(6):e8746.
>
> [2] Bunne C, Hsieh YP, Cuturi M, Krause A. The schrödinger bridge between gaussian measures has a closed form. InInternational Conference on Artificial Intelligence and Statistics 2023 Apr 11 (pp. 5802-5833). PMLR.
>
> [3] Shen Y, Berlinghieri R, Broderick T. Multi-marginal schr\" odinger bridges with iterative reference refinement. arXiv preprint arXiv:2408.06277. 2024 Aug 12.
>
> [4] Qiu X, Zhang Y, Martin-Rufino JD, Weng C, Hosseinzadeh S, Yang D, Pogson AN, Hein MY, Min KH, Wang L, Grody EI. Mapping transcriptomic vector fields of single cells. Cell. 2022 Feb 17;185(4):690-711.
>
> [5] Vargas F, Thodoroff P, Lamacraft A, Lawrence N. Solving schrödinger bridges via maximum likelihood. Entropy. 2021 Aug 31;23(9):1134.
>
> [6] Thornton J, Cuturi M. Rethinking initialization of the sinkhorn algorithm. InInternational Conference on Artificial Intelligence and Statistics 2023 Apr 11 (pp. 8682-8698). PMLR.

---

> ### Comment · Area_Chair_7ecr · 2025-08-05
>
> Dear reviewer,
> If not already, please take a look at the authors' rebuttal, and discuss if necessary.
> Thanks,
> -AC

---

### Note · Authors · 2025-08-13

We thank all reviewers for their detailed and thoughtful feedback on our work, and the AC and SAC for handling our submission.
Thanks to reviewer feedback, we have been able to strengthen our work in several directions.
These additions and clarifications from our rebuttal have been well received, as evidenced by two reviewers raising their scores and the others maintaining their original positive score.

We will incorporate all of the material from our rebuttal into the final version of our paper, including:

* Additional background in the introduction, related work discussion/context for our contribution. [reviewers xnYH, xh1L, uH9r]
* Additional discussion on the dynamical assumptions of the model [reviewer kfao]
* Additional discussion and results on runtime, scaling considerations with dimension $d$ [reviewers xnYH, uH9r]
* New results for cell cycle data in $d = 50, 100$ PCs [reviewers xnYH, uH9r]
* New results for Gaussian SB benchmark in $d = 100$ [reviewers xnYH, uH9r]
* Discussion on limitations of minibatch OT, diagonal or non-diagonal drift [reviewer uH9r]

We have also fixed various issues with referencing and formatting, including those pointed out by reviewers.

Once again, thank you all for your engagement and input in this process thus far.

The Authors

---

### Decision · Program_Chairs · 2025-09-17

**Decision:**

Accept (poster)

**Comment:**

The paper generalizes the classical Schr\"odinger's bridge to a case where the reference path measure is induced by an OU process with asymmetric linear coefficient matrix. Analytical solution is worked out for a special case where both marginals are Gaussian, which is not surprising, but a simulation-free algorithm is also proposed for general marginals. The method is then demonstrated on relatively low-dimensional data sets. Both reviewers and I found the idea interesting, and we appreciated that details are worked out. For the benefit of the authors, however, I'd like to point out that asymmetric A means the process is irreversible, but not non-equilibrium (OU is still geometric ergodic in your context where A is negative definite, which means it converges to an equilibrium distribution). I'd also avoid saying asymmetric A means non-conservative forces (as in the abstract), because dissipative A is never conservative,while on the other hand A=[0 1; -1 0] for example gives perfectly conservative dynamics. However, the correction/removal of these statements does not really affect the machine learning component of this paper, and based on reviewers' consensus, I'm happy to recommend acceptance. The authors are encouraged to take all the discussions into consideration when preparing for a revised/final version.